# Learning a Single Neuron with Bias Using Gradient Descent

**Gal Vardi**[*]
Weizmann Institute of Science
gal.vardi@weizmann.ac.il

**Gilad Yehudai**[*]
Weizmann Institute of Science
gilad.yehudai@weizmann.ac.il

**Ohad Shamir**
Weizmann Institute of Science
ohad.shamir@weizmann.ac.il

## Abstract

We theoretically study the fundamental problem of learning a single neuron with a bias term ($\mathbf{x} \mapsto \sigma(\langle \mathbf{w}, \mathbf{x} \rangle + b)$) in the realizable setting with the ReLU activation, using gradient descent. Perhaps surprisingly, we show that this is a significantly different and more challenging problem than the bias-less case (which was the focus of previous works on single neurons), both in terms of the optimization geometry as well as the ability of gradient methods to succeed in some scenarios. We provide a detailed study of this problem, characterizing the critical points of the objective, demonstrating failure cases, and providing positive convergence guarantees under different sets of assumptions. To prove our results, we develop some tools which may be of independent interest, and improve previous results on learning single neurons.

## 1 Introduction

Learning a single ReLU neuron with gradient descent is a fundamental primitive in the theory of deep learning, and has been extensively studied in recent years. Indeed, in order to understand the success of gradient descent on complicated neural networks, it seems reasonable to expect a satisfying analysis of convergence on a single neuron. Although many previous works studied the problem of learning a single neuron with gradient descent, none of them considered this problem with an explicit bias term.

In this work, we study the common setting of learning a single neuron with respect to the squared loss, using gradient descent. We focus on the realizable setting, where the inputs are drawn from a distribution $\mathcal{D}$ on $\mathbb{R}^{d+1}$, and are labeled by a single target neuron of the form $\mathbf{x} \mapsto \sigma(\langle \mathbf{v}, \mathbf{x} \rangle)$, where $\sigma : \mathbb{R} \to \mathbb{R}$ is some non-linear activation function. To capture the bias term, we assume that the distribution $\mathcal{D}$ is such that its first $d$ components are drawn from some distribution $\tilde{\mathcal{D}}$ on $\mathbb{R}^d$, and the last component is a constant 1. Thus, the input $\mathbf{x}$ can be decomposed as $(\tilde{\mathbf{x}}, 1)$ with $\tilde{\mathbf{x}} \sim \tilde{\mathcal{D}}$, the vector $\mathbf{v}$ can be decomposed as $(\tilde{\mathbf{v}}, b_\mathbf{v})$, where $\tilde{\mathbf{v}} \in \mathbb{R}^d$ and $b_\mathbf{v} \in \mathbb{R}$, and the target neuron computes a function of the form $\mathbf{x} \mapsto \sigma(\langle \tilde{\mathbf{v}}, \tilde{\mathbf{x}} \rangle + b_\mathbf{v})$. Similarly, we can define the learned neuron as $\mathbf{x} \mapsto \sigma(\langle \tilde{\mathbf{w}}, \tilde{\mathbf{x}} \rangle + b_\mathbf{w})$, where $\mathbf{w} = (\tilde{\mathbf{w}}, b_\mathbf{w})$. Overall, we can write the objective function we wish to

---

[*]Equal contribution

optimize as follows:

$$F(\mathbf{w}) := \underset{\mathbf{x} \sim \mathcal{D}}{\mathbb{E}} \left[ \frac{1}{2} \left( \sigma(\mathbf{w}^\top \mathbf{x}) - \sigma(\mathbf{v}^\top \mathbf{x}) \right)^2 \right] \tag{1}$$

$$= \underset{\tilde{\mathbf{x}} \sim \tilde{\mathcal{D}}}{\mathbb{E}} \left[ \frac{1}{2} \left( \sigma(\tilde{\mathbf{w}}^\top \tilde{\mathbf{x}} + b_\mathbf{w}) - \sigma(\tilde{\mathbf{v}}^\top \tilde{\mathbf{x}} + b_\mathbf{v}) \right)^2 \right]. \tag{2}$$

Throughout the paper we consider the commonly used ReLU activation function: $\sigma(x) = \max\{0, x\}$.

Although the problem of learning a single neuron is well studied (e.g. Soltanolkotabi [2017], Yehudai and Shamir [2020], Frei et al. [2020], Du et al. [2017], Kalan et al. [2019], Tan and Vershynin [2019], Mei et al. [2018], Oymak and Soltanolkotabi [2018]), none of the previous works considered the problem with an additional bias term. Moreover, previous works on learning a single neuron with gradient methods have certain assumptions on the input distribution $\mathcal{D}$, which do not apply when dealing with a bias term (for example, a certain "spread" in all directions, which does not apply when $\mathcal{D}$ is supported on $\{1\}$ in the last coordinate).

Since neural networks with bias terms are the common practice, it is natural to ask how adding a bias term affects the optimization landscape and the convergence of gradient descent. Although one might conjecture that this is just a small modification to the problem, we in fact show that the effect of adding a bias term is very significant, both in terms of the optimization landscape and in terms of which gradient descent strategies can or cannot work. Our main contributions are as follows:

- We start in Section 3 with some negative results, which demonstrate how adding a bias term makes the problem more difficult. In particular, we show that with a bias term, gradient descent or gradient flow[2] can sometimes fail with probability close to half over the initialization, even when the input distribution is uniform over a ball. In contrast, Yehudai and Shamir [2020] show that without a bias term, for the same input distribution, gradient flow converges to the global minimum with probability $1$.

- In Section 4 we give a full characterization of the critical points of the loss function. We show that adding a bias term changes the optimization landscape significantly: In previous works (cf. Yehudai and Shamir [2020]) it has been shown that under mild assumptions on the input distribution, the only critical points are $\mathbf{w} = \mathbf{v}$ (i.e., the global minimum) and $\mathbf{w} = \mathbf{0}$. We prove that when we have a bias term, the set of critical points has a positive measure, and that there is a cone of local minima where the loss function is flat.

- In Sections 5 and 6 we show that gradient descent converges to the global minimum at a linear rate, under some assumptions on the input distribution and on the initialization. We give two positive convergence results, where each result is under different assumptions, and thus the results complement each other. We also use different techniques for proving each of the results: The analysis in Section 6 follows from some geometric arguments and extends the technique from Yehudai and Shamir [2020], Frei et al. [2020]. The analysis in Section 5 introduces a novel technique, not used in previous works on learning a single neuron, and has a more algebraic nature. Moreover, that analysis implies that under mild assumptions, gradient descent with random initialization converges to the global minimum with probability $1 - e^{\Omega(d)}$.

- The best known result for learning a single neuron without bias using gradient descent for an input distribution that is not spherically symmetric, establishes convergence to the global minimum with probability close to $\frac{1}{2}$ over the random initialization [Yehudai and Shamir, 2020, Frei et al., 2020]. With our novel proof technique presented in Section 5 this result can be improved to probability at least $1 - e^{-\Omega(d)}$ (see Remark 5.7).

## Related work

Although there are no previous works on learning a single neuron with an explicit bias term, there are many works that consider the problem of a single neuron under different settings and assumptions.

Several papers showed that the problem of learning a single neuron can be solved under minimal assumptions using algorithms which are not gradient-based (such as gradient descent or SGD). These

---

[2]I.e., gradient descent with infinitesimal step size.

algorithms include the Isotron proposed by Kalai and Sastry [2009] and the GLMtron proposed by Kakade et al. [2011]. The GLMtron algorithm is also analyzed in Diakonikolas et al. [2020]. These algorithms allow learning a single neuron with bias. We note that these are non-standard algorithms, whereas we focus on the standard gradient descent algorithm. An efficient algorithm for learning a single neuron with error parameter $\epsilon = \Omega(1/\log(d))$ was also obtained in Goel et al. [2017].

In Mei et al. [2018] the authors study the empirical risk of the single neuron problem. However, their analysis does not include the ReLU activation, or adding a bias term. A related analysis is also given in Oymak and Soltanolkotabi [2018], where the ReLU activation is not considered.

Several papers showed convergence guarantees for the single neuron problem with ReLU activation under certain distributional assumptions, although none of these assumptions allows for a bias term. Notably, Tian [2017], Soltanolkotabi [2017], Kalan et al. [2019], Brutzkus and Globerson [2017] showed convergence guarantees for gradient methods when the inputs have a standard Gaussian distribution, without a bias term. Du et al. [2017] showed that under a certain subspace eigenvalue assumption a single neuron can be learned with SGD, although this assumption does not allow adding a bias term. Yehudai and Shamir [2020], Frei et al. [2020] use an assumption about the input distribution being sufficiently "spread" in all directions, which does not allow for a bias term (since that requires an input distribution supported on $\{1\}$ in the last coordinate). Yehudai and Shamir [2020] showed a convergence result under the realizable setting, while Frei et al. [2020] considered the agnostic and noisy settings. In Tan and Vershynin [2019] convergence guarantees are given for the absolute value activation, and a specific distribution which does not allow a bias term.

Less directly related, Vardi and Shamir [2020] studied the problem of implicit regularization in the single neuron setting. In Yehudai and Shamir [2019], Kamath et al. [2020] it is shown that approximating a single neuron using random features (or kernel methods) is not tractable in high dimensions. We note that these results explicitly require that the single neuron which is being approximated will have a bias term. Thus, our work complements these works by showing that the problem of learning a single neuron with bias is also learnable using gradient descent (under certain assumptions). Agnostically learning a single neuron with non gradient-based algorithms and hardness of a agnostically learning a single neuron were studied in Diakonikolas et al. [2020], Goel et al. [2019, 2020].

## 2 Preliminaries

**Notations.** We use bold-faced letters to denote vectors, e.g., $\mathbf{x} = (x_1, \ldots, x_d)$. For $\mathbf{u} \in \mathbb{R}^d$ we denote by $\|\mathbf{u}\|$ the Euclidean norm. We denote $\bar{\mathbf{u}} = \frac{\mathbf{u}}{\|\mathbf{u}\|}$, namely, the unit vector in the direction of $\mathbf{u}$. For $1 \leq i \leq j \leq d$ we denote $\mathbf{u}_{i:j} = (u_i, \ldots, u_j) \in \mathbb{R}^{j-i+1}$. We denote by $\mathbb{1}(\cdot)$ the indicator function, for example $\mathbb{1}(t \geq 5)$ equals 1 if $t \geq 5$ and 0 otherwise. We denote by $U([-r, r])$ the uniform distribution over the interval $[-r, r]$ in $\mathbb{R}$, and by $\mathcal{N}(\mathbf{0}, \Sigma)$ the multivariate normal distribution with mean $\mathbf{0}$ and covariance matrix $\Sigma$. Given two vectors $\mathbf{w}, \mathbf{v}$ we let $\theta(\mathbf{w}, \mathbf{v}) = \arccos\left(\frac{\langle \mathbf{w}, \mathbf{v} \rangle}{\|\mathbf{w}\|\|\mathbf{v}\|}\right) = \arccos(\langle \bar{\mathbf{w}}, \bar{\mathbf{v}} \rangle) \in [0, \pi]$. For a vector $\mathbf{u} \in \mathbb{R}^{d+1}$ we often denote by $\tilde{\mathbf{u}} \in \mathbb{R}^d$ the first $d$ components of $\mathbf{u}$, and denote by $b_{\mathbf{u}} \in \mathbb{R}$ its last component.

**Gradient methods.** In this paper we focus on the following two standard gradient methods for optimizing our objective $F(\mathbf{w})$ from Eq. (2):

- **Gradient descent:** We initialize at some $\mathbf{w}_0 \in \mathbb{R}^{d+1}$, and set a fixed learning rate $\eta > 0$. At each iteration $t \geq 0$ we have: $\mathbf{w}_{t+1} = \mathbf{w}_t - \eta \nabla F(\mathbf{w}_t)$.

- **Gradient Flow:** We initialize at some $\mathbf{w}(0) \in \mathbb{R}^{d+1}$, and for every time $t \geq 0$, we set $\mathbf{w}(t)$ to be the solution of the differential equation $\dot{\mathbf{w}} = -\nabla F(\mathbf{w}(t))$. This can be thought of as a continuous form of gradient descent, where the learning rate is infinitesimally small.

The gradient of the objective in Eq. (1) is:

$$\nabla F(\mathbf{w}) = \mathop{\mathbb{E}}_{\mathbf{x} \sim \mathcal{D}} \left[ \left( \sigma(\mathbf{w}^\top \mathbf{x}) - \sigma(\mathbf{v}^\top \mathbf{x}) \right) \cdot \sigma'(\mathbf{w}^\top \mathbf{x}) \mathbf{x} \right]. \tag{3}$$

Since $\sigma$ is the ReLU function, it is differentiable everywhere except for 0. Practical implementations of gradient methods define $\sigma'(0)$ to be some constant in $[0, 1]$. Following this convention, the gradient

used by these methods still correspond to Eq. (3). We note that the exact value of $\sigma'(0)$ has no effect on our results.

## 3 Negative results

In this section we demonstrate that adding bias to the problem of learning a single neuron with gradient descent can make the problem significantly harder.

First, on an intuitive level, previous results (e.g., Yehudai and Shamir [2020], Frei et al. [2020], Soltanolkotabi [2017], Du et al. [2017], Tan and Vershynin [2019]) considered assumptions on the input distribution, which require enough "spread" in all directions (for example, a strictly positive density in some neighborhood around the origin). Adding a bias term, even if the first $d$ coordinates of the distribution satisfy a "spread" assumption, will give rise to a direction without "spread", since in this direction the distribution is concentrated on 1, hence the previous results do not apply.

Next, we show two negative results where the input distribution is uniform on a ball around the origin. We note that due to Theorem 6.4 from Yehudai and Shamir [2020], we know that gradient flow on a single neuron without bias will converge to the global minimum with probability 1 over the random initialization. The only case where it will fail to converge is when $\mathbf{w}_0$ is initialized in the exact direction $-\mathbf{v}$, which happens with probability 0 with standard random initializations.

### 3.1 Initialization in a flat region

If we initialize the bias term in the same manner as the other coordinates, then we can show that gradient descent will fail with probability close to half, even if the input distribution is uniform over a (certain) origin-centered ball:

**Theorem 3.1.** *Suppose we initialize each coordinate of $\mathbf{w}_0$ (including the bias) according to $U([-1,1])$. Let $\epsilon > 0$ and let $\tilde{\mathcal{D}}$ be the uniform distribution supported on a ball around the origin in $\mathbb{R}^d$ of radius $\epsilon$. Then, w.p $> 1/2 - \epsilon\sqrt{d}$, gradient descent on the objective in Eq. (2) satisfies $\mathbf{w}_t = \mathbf{w}_0$ for all $t$ (namely, it gets stuck at its initial point $\mathbf{w}_0$).*

Note that by Theorem 6.4 in Yehudai and Shamir [2020], if there is no bias term in the objective, then gradient descent will converge to the global minimum w.p 1 using this random initialization scheme and this input distribution. The intuition for the proof is that with constant probability over the initialization, $b_{\mathbf{w}}$ is small enough so that $\sigma(\tilde{\mathbf{w}}^\top \tilde{\mathbf{x}} + b_{\mathbf{w}}) = 0$ almost surely. If this happens, then the gradient will be $\mathbf{0}$ and gradient descent will never move. The full proof can be found in Appendix B.1. We note that Theorem 3.1 is applicable when $\epsilon$ is sufficiently small, e.g. $\epsilon \ll 1/\sqrt{d}$.

### 3.2 Targets with negative bias

Theorem 3.1 shows a difference between learning with and without the bias term. A main caveat of this example is the requirement that the bias is initialized in the same manner as the other parameters. Standard deep learning libraries (e.g. Pytorch [Paszke et al., 2019]) often initialize the bias term to zero by default, while using random initialization schemes for the other parameters.

Alas, we now show that even if we initialize the bias term to be exactly zero, and the input distribution is uniform over an arbitrary origin-centered ball, we might fail to converge to the global minimum for certain target neurons:

**Theorem 3.2.** *Let $\tilde{\mathcal{D}}$ be the uniform distribution on $\mathcal{B} = \{\tilde{\mathbf{x}} \in \mathbb{R}^d : \|\tilde{\mathbf{x}}\| \leq r\}$ for some $r > 0$. Let $\mathbf{v} \in \mathbb{R}^{d+1}$ such that $\tilde{\mathbf{v}} = (1, 0, \ldots, 0)^\top$ and $b_{\mathbf{v}} = -\left(r - \frac{r}{2d^2}\right)$. Let $\mathbf{w}_0 \in \mathbb{R}^{d+1}$ such that $b_{\mathbf{w}_0} = 0$ and $\tilde{\mathbf{w}}_0$ is drawn from the uniform distribution on a sphere of radius $\rho > 0$. Then, with probability at least $\frac{1}{2} - o_d(1)$ over the choice of $\mathbf{w}_0$, gradient flow does not converge to the global minimum.*

We prove the theorem in Appendix B.2. The intuition behind the proof is the following: The target neuron has a large negative bias, so that only a small (but positive) measure of input points are labelled as non-zero. By randomly initializing $\tilde{\mathbf{w}}$, with probability close to $\frac{1}{2}$ there are no inputs that both $\mathbf{v}$ and $\mathbf{w}$ label positively. Since the gradient is affected only by inputs that $\mathbf{w}$ labels positively, then during the optimization process the gradient will be independent of the direction of $\mathbf{v}$, and $\mathbf{w}$ will not converge to the global minimum.

**Remark 3.3.** *Theorem 3.2 shows that gradient flow is not guaranteed to converge to a global minimum when $b_{\mathbf{v}}$ is negative, instead it converges to a local minimum with a loss of $F(\mathbf{0})$. However, the loss $F(\mathbf{0})$ is determined by the input distribution. Take $\mathbf{v} = \left(1, 0, \ldots, 0, -\left(r - \frac{r}{2d^2}\right)\right)^{\top}$ considered in the theorem. On one hand, for a uniform distribution on a ball of radius $r$ as in the theorem we have:*

$$
\begin{aligned}
F(\mathbf{0}) &= \frac{1}{2} \cdot \mathbb{E}_{\mathbf{x}}\left[\left(\sigma(\mathbf{v}^{\top}\mathbf{x})\right)^2\right] = \frac{1}{2} \cdot \mathbb{E}_{\mathbf{x}}\left[\mathbb{1}(\mathbf{v}^{\top}\mathbf{x} \geq 0)\left(\mathbf{v}^{\top}\mathbf{x}\right)^2\right] \\
&= \frac{1}{2} \cdot \mathbb{E}_{\mathbf{x}}\left[\mathbb{1}\left(x_1 \geq r - \frac{r}{2d^2}\right)\left(x_1 - \left(r - \frac{r}{2d^2}\right)\right)^2\right] \\
&\leq \frac{1}{2} \cdot \frac{r^2}{4d^4} \cdot \Pr_{\mathbf{x}}\left[x_1 \geq r\left(1 - \frac{1}{2d^2}\right)\right] \leq r^2 e^{-\Omega(d)}.
\end{aligned}
$$

*Thus, for any reasonable $r$, a local minimum with loss $F(\mathbf{0})$ is almost as good as the global minimum. On the other hand, take a distribution $\tilde{\mathcal{D}}$ with a support bounded in a ball of radius $r$, such that half of its mass is uniformly distributed in $A := \left\{\tilde{\mathbf{x}} \in \mathbb{R}^d : x_1 > r - \frac{r}{4d^2}\right\}$, and the other half is uniformly distributed in $\mathcal{B} \setminus A$. In this case, it is not hard to see that the same proof as in Theorem 3.2 works, and gradient flow will converge to a local minimum with loss $F(\mathbf{0}) = \Omega\left(\frac{r}{d^2}\right)$, which is arbitrarily large if $r$ is large enough.*

Although in the example given in Theorem 3.2 the objective at $\mathbf{w} = \mathbf{0}$ is almost as good as the objective at $\mathbf{w} = \mathbf{v}$, we emphasize that w.p almost $\frac{1}{2}$ gradient flow cannot reach the global minimum even asymptotically. On the other hand, in the bias-less case by Theorem 6.4 in Yehudai and Shamir [2020] gradient flow on the same input distribution will reach the global minimum w.p 1. Also note that the scale of the initialization of $\mathbf{w}_0$ has no effect on the result.

## 4 Characterization of the critical points

In the previous section we have shown two examples where gradient methods on the problem of a single neuron with bias will either get stuck in a flat region, or converge to a local minimum. In this section we delve deeper into the examples presented in the previous section, and give a full characterization of the critical points of the objective. We will use the following assumption on the input distribution:

**Assumption 4.1.** *The distribution $\tilde{\mathcal{D}}$ on $\mathbb{R}^d$ has a density function $p(\tilde{\mathbf{x}})$, and there are $\beta, c > 0$, such that $\tilde{\mathcal{D}}$ is supported on $\{\tilde{\mathbf{x}} : \|\tilde{\mathbf{x}}\| \leq c\}$, and for every $\tilde{\mathbf{x}}$ in the support we have $p(\tilde{\mathbf{x}}) \geq \beta$.*

The assumption essentially states that the distribution over the first $d$ coordinates (*without* the bias term) has enough "spread" in all directions, and covers standard distributions such as uniform over a ball of radius $c$. Other similar assumptions are made in previous works (e.g. Yehudai and Shamir [2020], Frei et al. [2020]). We note that in Yehudai and Shamir [2020] it is shown that without any assumption on the distribution, it is impossible to ensure convergence, hence we must have some kind of assumption for this problem to be learnable with gradient methods. Under this assumption we can characterize the critical points of the objective.

**Theorem 4.2.** *Consider the objective in Eq. (2) with $\mathbf{v} \neq \mathbf{0}$, and assume that the distribution $\tilde{\mathcal{D}}$ on the first $d$ coordinates satisfies Assumption 4.1. Then $\mathbf{w} \neq \mathbf{0}$ is a critical point of $F$ (i.e., is a root of Eq. (3)) if and only if it satisfies one of the following:*

- *$\mathbf{w} = \mathbf{v}$, in which case $\mathbf{w}$ is a global minimum.*

- *$\mathbf{w} = (\tilde{\mathbf{w}}, b_{\mathbf{w}})$ where $\tilde{\mathbf{w}} = \mathbf{0}$ and $b_{\mathbf{w}} < 0$.*

- *$\tilde{\mathbf{w}} \neq \mathbf{0}$ and $-\frac{b_{\mathbf{w}}}{\|\tilde{\mathbf{w}}\|} \geq c$.*

*In the latter two cases, $F(\mathbf{w}) = F(\mathbf{0})$. Hence, if $\mathbf{0}$ is not a global minimum, then $\mathbf{w}$ is not a global minimum.*

We note that $F(\mathbf{0}) = \frac{1}{2}\mathbb{E}_{\mathbf{x}}[\sigma(\mathbf{v}^{\top}\mathbf{x})^2]$, so $\mathbf{0}$ is a global minimum only if the target neuron returns $0$ with probability 1.

**Remark 4.3** (The case $\mathbf{w} = \mathbf{0}$). *We intentionally avoided characterizing the point $\mathbf{w} = \mathbf{0}$, since the objective is not differentiable there (this is the only point of non-differentiability), and the gradient there is determined by the value of the ReLU activation at $0$. For $\sigma'(0) = 0$ the gradient at $\mathbf{w} = \mathbf{0}$ is zero, and this is a non-differentiable saddle point. For $\sigma'(0) = 1$ (or any other positive value), the gradient at $\mathbf{w} = \mathbf{0}$ is non-zero, and it will point at a direction which depends on the distribution. We note that in Soltanolkotabi [2017] the authors define $\sigma'(0) = 1$, and use a symmetric distribution, in which case the gradient at $\mathbf{w} = \mathbf{0}$ points exactly at the direction of the target $\mathbf{v}$. This is a crucial part of their convergence analysis.*

We emphasize that with a bias term, there is a non-zero measure manifold of critical points (corresponding to the third bullet in the theorem). On the other hand, without a bias term the only critical point besides the global minimum (under mild assumptions on the input distribution) is at the origin $\mathbf{w} = \mathbf{0}$ (cf. Yehudai and Shamir [2020]). The full proof is in Appendix C.

The assumption on the support of $\tilde{\mathcal{D}}$ is made for simplicity. It can be relaxed to having a distribution with exponentially bounded tail, e.g. standard Gaussian. In this case, some of the critical points will instead have a non-zero gradient which is exponentially small. We emphasize that when running optimization algorithms on finite-precision machines, which are used in practice, these "almost" critical points behave essentially like critical points since the gradient is extremely small.

Revisiting the negative examples from Section 3, the first example (Theorem 3.1) shows that if we do not initialize the bias of $\mathbf{w}$ to zero, then there is a positive probability to initialize at a critical point which is not the global minimum. The second example (Theorem 3.2) shows that even if we initialize the bias of $\mathbf{w}$ to be zero, there is still a positive probability to converge to a critical point which is not the global minimum. Hence, in order to guarantee convergence we need to have more assumptions on either the input distribution, the target $\mathbf{v}$ or the initialization. In the next section, we show that adding such assumptions are indeed sufficient to get positive convergence guarantees.

## 5 Convergence for initialization with loss slightly better than trivial

In this section, we show that under some assumptions on the input distribution, if gradient descent is initialized such that $F(\mathbf{w}_0) < F(\mathbf{0})$ then it is guaranteed to converge to the global minimum. In Subsection 5.2, we study under what conditions this is likely to occur with standard random initialization.

### 5.1 Convergence if $F(\mathbf{w}_0) < F(\mathbf{0})$

To state our results, we need the following assumption:
**Assumption 5.1.**

1. *The distribution $\mathcal{D}$ is supported on $\{\mathbf{x} \in \mathbb{R}^{d+1} : \|\mathbf{x}\| \leq c\}$ for some $c \geq 1$.*

2. *The distribution $\tilde{\mathcal{D}}$ over the first $d$ coordinates is bounded in all directions: there is $c' > 0$ such that for every $\tilde{\mathbf{u}}$ with $\|\tilde{\mathbf{u}}\| = 1$ and every $a \in \mathbb{R}$ and $b \geq 0$, we have $\Pr_{\tilde{\mathbf{x}} \sim \tilde{\mathcal{D}}}\left[\tilde{\mathbf{u}}^\top \tilde{\mathbf{x}} \in [a, a+b]\right] \leq b \cdot c'$.*

3. *We assume w.l.o.g. that $\|\mathbf{v}\| = 1$ and $c' \geq 1$.*

Assumption (3) helps simplifying some expressions in our convergence result, and is not necessary. Assumption (2) requires that the distribution is not too concentrated in a short interval. For example, if $\tilde{\mathcal{D}}$ is spherically symmetric then the marginal density of the first (or any other) coordinate is bounded by $c'$. Note that we do not assume that $\tilde{\mathcal{D}}$ is spherically symmetric.

**Theorem 5.2.** *Under Assumption 5.1 we have the following. Let $\delta > 0$ and let $\mathbf{w}_0 \in \mathbb{R}^{d+1}$ such that $F(\mathbf{w}_0) \leq F(\mathbf{0}) - \delta$. Let $\gamma = \frac{\delta^3}{3 \cdot 12^2 (\|\mathbf{w}_0\| + 2)^3 c^8 c'^2}$. Assume that gradient descent runs starting from $\mathbf{w}_0$ with step size $\eta \leq \frac{\gamma}{c^4}$. Then, for every $t$ we have*

$$\|\mathbf{w}_t - \mathbf{v}\|^2 \leq \|\mathbf{w}_0 - \mathbf{v}\|^2 (1 - \gamma\eta)^t .$$

The formal proof appears in Appendix D, but we provide the main ideas below. First, note that
$$\|\mathbf{w}_{t+1} - \mathbf{v}\|^2 = \|\mathbf{w}_t - \eta\nabla F(\mathbf{w}_t) - \mathbf{v}\|^2$$
$$= \|\mathbf{w}_t - \mathbf{v}\|^2 - 2\eta\langle\nabla F(\mathbf{w}_t), \mathbf{w}_t - \mathbf{v}\rangle + \eta^2\|\nabla F(\mathbf{w}_t)\|^2 .$$

Hence, in order to show that $\|\mathbf{w}_{t+1} - \mathbf{v}\|^2 \le \|\mathbf{w}_t - \mathbf{v}\|^2 (1 - \gamma\eta)$ we need to obtain an upper bound for $\|\nabla F(\mathbf{w}_t)\|$ and a lower bound for $\langle \nabla F(\mathbf{w}_t), \mathbf{w}_t - \mathbf{v}\rangle$. Achieving the lower bound for $\langle \nabla F(\mathbf{w}_t), \mathbf{w}_t - \mathbf{v}\rangle$ is the challenging part, and we show that in order to establish such a bound it suffices to obtain a lower bound for $\Pr_{\mathbf{x}}\left[\mathbf{w}_t^\top \mathbf{x} \ge 0, \mathbf{v}^\top \mathbf{x} \ge 0\right]$. We prove that if $F(\mathbf{w}_t) \le F(\mathbf{0}) - \delta$ then $\Pr_{\mathbf{x}}\left[\mathbf{w}_t^\top \mathbf{x} \ge 0, \mathbf{v}^\top \mathbf{x} \ge 0\right] \ge \frac{\delta}{c^2 \|\mathbf{w}_t\|}$. Hence, if $F(\mathbf{w}_t)$ remains at most $F(\mathbf{0}) - \delta$ for every $t$, then a lower bound for $\langle \nabla F(\mathbf{w}_t), \mathbf{w}_t - \mathbf{v}\rangle$ can be achieved, which completes the proof. However, it is not obvious that $F(\mathbf{w}_t)$ remains at most $F(\mathbf{0}) - \delta$ throughout the training process. When running gradient descent on a smooth loss function we can choose a sufficiently small step size such that the loss decreases in each step, but here the function $F(\mathbf{w})$ is highly non-smooth around $\mathbf{w} = \mathbf{0}$. That is, the Lipschitz constant of $\nabla F(\mathbf{w})$ is unbounded. We show that if $F(\mathbf{w}_t) \le F(\mathbf{0}) - \delta$ then $\mathbf{w}_t$ is sufficiently far from $\mathbf{0}$, and hence the smoothness of $F$ around $\mathbf{w}_t$ can be bounded, which allows us to choose a small step size that ensures that $F(\mathbf{w}_{t+1}) \le F(\mathbf{w}_t) \le F(\mathbf{0}) - \delta$. Hence, it follows that $F(\mathbf{w}_t)$ remains at most $F(\mathbf{0}) - \delta$ for every $t$.

As an aside, recall that in Section 4 we showed that other than $\mathbf{w} = \mathbf{v}$ all critical points of $F(\mathbf{w})$ are in a flat region where $F(\mathbf{w}) = F(\mathbf{0})$. Hence, the fact that $F(\mathbf{w}_t)$ remains at most $F(\mathbf{0}) - \delta$ for every $t$ implies that $\mathbf{w}_t$ does not reach the region of bad critical points, which explains the asymptotic convergence to the global minimum.

We also note that although we assume that the distribution has a bounded support, this assumption is mainly made for simplicity, and can be relaxed to have sub-Gaussian distributions with bounded moments. These distributions include, e.g. Gaussian distributions.

### 5.2 Convergence for Random Initialization

In Theorem 5.2 we showed that if $F(\mathbf{w}_0) < F(\mathbf{0})$ then gradient descent converges to the global minimum. We now show that under mild assumptions on the input distribution, a random initialization of $\mathbf{w}_0$ near zero satisfies this requirement. We will need the following assumption, also used in Yehudai and Shamir [2020], Frei et al. [2020]:

**Assumption 5.3.** *There are $\alpha, \beta > 0$ s.t the distribution $\tilde{\mathcal{D}}$ satisfies the following: For any vector $\tilde{\mathbf{w}} \ne \tilde{\mathbf{v}}$, let $\tilde{\mathcal{D}}_{\tilde{\mathbf{w}}, \tilde{\mathbf{v}}}$ denote the marginal distribution of $\tilde{\mathcal{D}}$ on the subspace spanned by $\tilde{\mathbf{w}}, \tilde{\mathbf{v}}$ (as a distribution over $\mathbb{R}^2$). Then any such distribution has a density function $p_{\tilde{\mathbf{w}}, \tilde{\mathbf{v}}}(\hat{\mathbf{x}})$ over $\mathbb{R}^2$ such that $\inf_{\hat{\mathbf{x}}: \|\hat{\mathbf{x}}\| \le \alpha} p_{\tilde{\mathbf{w}}, \tilde{\mathbf{v}}}(\hat{\mathbf{x}}) \ge \beta$.*

The main technical tool for proving convergence under random initialization is the following:

**Theorem 5.4.** *Assume that the input distribution $\mathcal{D}$ is supported on $\{\mathbf{x} \in \mathbb{R}^{d+1} : \|\mathbf{x}\| \le c\}$ for some $c \ge 1$, and Assumption 5.3 holds. Let $\mathbf{v} \in \mathbb{R}^{d+1}$ such that $\|\mathbf{v}\| = 1$ and $-\frac{b_{\mathbf{v}}}{\|\tilde{\mathbf{v}}\|} \le \alpha \cdot \frac{\sin\left(\frac{\pi}{8}\right)}{4}$. Let $M = \frac{\alpha^4 \beta \sin^3\left(\frac{\pi}{8}\right)}{256c}$. Let $\mathbf{w} \in \mathbb{R}^{d+1}$ such that $b_{\mathbf{w}} = 0$, $\theta(\tilde{\mathbf{w}}, \tilde{\mathbf{v}}) \le \frac{3\pi}{4}$ and $\|\mathbf{w}\| < \frac{2M}{c^2}$. Then, $F(\mathbf{w}) \le F(\mathbf{0}) + \|\mathbf{w}\|^2 \cdot \frac{c^2}{2} - \|\mathbf{w}\| \cdot M < F(\mathbf{0})$.*

We prove the theorem in Appendix D.1. The main idea is that since

$$F(\mathbf{w}) = \mathbb{E}_{\mathbf{x}} \left[ \frac{1}{2} \left( \sigma(\mathbf{w}^\top \mathbf{x}) - \sigma(\mathbf{v}^\top \mathbf{x}) \right)^2 \right]$$

$$= F(\mathbf{0}) + \frac{1}{2} \mathbb{E}_{\mathbf{x}} \left[ \left( \sigma(\mathbf{w}^\top \mathbf{x}) \right)^2 \right] - \mathbb{E}_{\mathbf{x}} \left[ \sigma(\mathbf{w}^\top \mathbf{x}) \sigma(\mathbf{v}^\top \mathbf{x}) \right]$$

$$\le F(\mathbf{0}) + \|\mathbf{w}\|^2 \cdot \frac{c^2}{2} - \|\mathbf{w}\| \cdot \mathbb{E}_{\mathbf{x}} \left[ \sigma(\bar{\mathbf{w}}^\top \mathbf{x}) \sigma(\mathbf{v}^\top \mathbf{x}) \right] \,,$$

then it suffices to obtain a lower bound for $\mathbb{E}_{\mathbf{x}} \left[ \sigma(\bar{\mathbf{w}}^\top \mathbf{x}) \sigma(\mathbf{v}^\top \mathbf{x}) \right]$. In the proof we show that such a bound can be achieved if the conditions of the theorem hold.

Suppose that $\mathbf{w}_0$ is such that $\tilde{\mathbf{w}}_0$ is drawn from a spherically symmetric distribution and $b_{\mathbf{w}_0} = 0$. By standard concentration of measure arguments, it holds w.p. at least $1 - e^{\Omega(d)}$ that $\theta(\tilde{\mathbf{w}}_0, \tilde{\mathbf{v}}) \le \frac{3\pi}{4}$ (where the notation $\Omega(d)$ hides only numerical constants, namely, it does not depend on other parameters of the problem). Therefore, if $\tilde{\mathbf{w}}_0$ is drawn from the uniform distribution on a sphere of radius $\rho < \frac{2M}{c^2}$, then the theorem implies that w.h.p. we have $F(\mathbf{w}_0) < F(\mathbf{0})$. For such initialization Theorem 5.2 implies that gradient descent converges to the global minimum. For example, for $\rho = \frac{M}{c^2}$

we have w.h.p. that $F(\mathbf{w}_0) \leq F(\mathbf{0}) + \frac{\rho^2 c^2}{2} - \rho M = F(\mathbf{0}) - \frac{M^2}{2c^2}$, and thus Theorem 5.2 applies with $\delta = \frac{M^2}{2c^2}$. Thus, we have the following corollary:

**Corollary 5.5.** *Under Assumption 5.1 and Assumption 5.3 we have the following. Let $M = \frac{\alpha^4 \beta \sin^3\left(\frac{\pi}{8}\right)}{256c}$, let $\rho = \frac{M}{c^2}$, let $\delta = \frac{M^2}{2c^2}$, and let $\gamma = \frac{\delta^3}{3 \cdot 12^2 (\rho+2)^3 c^8 c'^2}$. Suppose that $-\frac{b_{\mathbf{v}}}{\|\tilde{\mathbf{v}}\|} \leq \alpha \cdot \frac{\sin\left(\frac{\pi}{8}\right)}{4}$, and $\mathbf{w}_0$ is such that $b_{\mathbf{w}_0} = 0$ and $\tilde{\mathbf{w}}_0$ is drawn from the uniform distribution on a sphere of radius $\rho$. Consider gradient descent with step size $\eta \leq \frac{\gamma}{c^4}$. Then, with probability at least $1 - e^{\Omega(d)}$ over the choice of $\mathbf{w}_0$ we have for every $t$: $\|\mathbf{w}_t - \mathbf{v}\|^2 \leq \|\mathbf{w}_0 - \mathbf{v}\|^2 (1 - \gamma\eta)^t$.*

We note that a similar result holds also if $\tilde{\mathbf{w}}_0$ is drawn from a normal distribution $\mathcal{N}(\mathbf{0}, \frac{\rho^2}{d} I)$.

**Remark 5.6** (The assumption on $b_{\mathbf{v}}$). *The assumption $-\frac{b_{\mathbf{v}}}{\|\tilde{\mathbf{v}}\|} \leq \alpha \cdot \frac{\sin\left(\frac{\pi}{8}\right)}{4}$ implies that the bias term $b_{\mathbf{v}}$ may be either positive or negative, but in case it is negative then it cannot be too large. This assumption is indeed crucial for the proof, but for "well-behaved" distributions, if this assumption is not satisfied (for a large enough $\alpha$), then the loss at $F(\mathbf{0})$ is already good enough. For example, for a standard Gaussian distribution and for every $\epsilon > 0$, we can choose $\alpha$ large enough such that for any bias term (positive or negative) we either: (1) converge to the global minimum with a loss of zero, or; (2) converge to a local minimum with a loss of $F(\mathbf{0})$, which is smaller then $\epsilon$. Moreover, we can show that by choosing $\alpha$ appropriately, and using the example in Theorem 3.2, if $-\frac{b_{\mathbf{v}}}{\|\tilde{\mathbf{v}}\|} \geq 2\alpha$ then gradient flow will converge to a non-global minimum with loss of $F(\mathbf{0})$. This means that our bound on $\alpha$ is tight up to a constant factor. For a further discussion on the assumption on $b_{\mathbf{v}}$, and how to choose $\alpha$ see Appendix E.*

Previous papers have shown separation between random features (or kernel) methods and neural networks in terms of their approximation power (see Yehudai and Shamir [2019], Kamath et al. [2020], and the discussion in Malach et al. [2021]). These works show that under a standard Gaussian distribution, random features cannot even approximate a single ReLU neuron, unless the number of features is exponential in the input dimension. That analysis crucially relies on the single neuron having a non-zero bias term. In this work we complete the picture by showing that gradient descent can indeed find a near-optimal neuron with non-zero bias. Thus, we see there is indeed essentially a separation between what can be learned using random features and using gradient descent over neural networks.

**Remark 5.7** (Learning a neuron without bias). *Yehudai and Shamir [2020] studied the problem of learning a single ReLU neuron without bias using gradient descent on a single neuron without bias. For input distributions that are not spherically symmetric they showed that gradient descent with random initialization near zero converges to the global minimum w.p. at least $\frac{1}{2} - o_d(1)$. Their result is also under Assumption 5.3. An immediate corollary from the discussion above is that if we learn a single neuron without bias using gradient descent with random initialization on a single neuron with bias, then the algorithm converges to the global minimum w.p. at least $1 - e^{\Omega(d)}$. Moreover, our proof technique can be easily adapted to the setting of learning a single neuron without bias using gradient descent on a single neuron without bias, namely, the setting studied in Yehudai and Shamir [2020]. It can be shown that in this setting gradient descent converges w.h.p to the global minimum. Thus, our technique allows us to improve the result of Yehudai and Shamir [2020] from probability $\frac{1}{2} - o_d(1)$ to probability $1 - e^{\Omega(d)}$.*

# 6 Convergence for spread and symmetric distributions

In this section we show that under a certain set of assumptions, different from the assumptions in Section 5, it is possible to show linear convergence of gradient descent to the global minimum. The assumptions we make for this theorem are as follows:

**Assumption 6.1.**

1. *The target vector $\mathbf{v}$ satisfies that $b_{\mathbf{v}} \geq 0$ and $\|\tilde{\mathbf{v}}\| = 1$.*

2. *The distribution $\tilde{\mathcal{D}}$ over the first $d$ coordinates is spherically symmetric.*

3. *Assumption 5.3 holds, and denoting by $\tau := \frac{\mathbb{E}_{\tilde{\mathbf{x}} \sim \tilde{\mathcal{D}}}[|\tilde{x}_1 \tilde{x}_2|]}{\mathbb{E}_{\tilde{\mathbf{x}} \sim \tilde{\mathcal{D}}}[\tilde{x}_1^2]}$, then $\alpha \geq 2.5\sqrt{2} \cdot \max\left\{1, \frac{1}{\sqrt{\tau}}\right\}$ where $\alpha$ is from Assumption 5.3.*

4. *Denote by $c := \mathbb{E}_{\tilde{\mathbf{x}} \sim \tilde{\mathcal{D}}}\left[\|\tilde{\mathbf{x}}\|^4\right]$, then $c < \infty$.*

We note that item (3) considers $x_1, x_2$, but due to the assumption on the symmetry of the distribution (item (2)), the assumption in item (3) holds for every $x_i, x_j$. Under these assumptions, we prove the following theorem:

**Theorem 6.2.** *Assume we initialize $\mathbf{w}_0$ such that $\|\mathbf{w}_0 - \mathbf{v}\|^2 < 1$, $b_{\mathbf{w}_0} \geq 0$ and that Assumption 6.1 holds. Then, there is a universal constant $C$, such that using gradient descent on $F(\mathbf{w})$ with step size $\eta < C \cdot \frac{\beta}{c\alpha^2} \min\{1, \tau\}$ yields that for every $t$ we have $\|\mathbf{w}_t - \mathbf{v}\|^2 \leq (1 - \eta\lambda)^t \|\mathbf{w}_0 - \mathbf{v}\|^2$, for $\lambda = C \cdot \frac{\beta}{c\alpha^2}$.*

This result has several advantages and disadvantages compared to those of the previous section. The main disadvantage is that the assumptions are generally more stringent: We focus only on positive target biases ($b_{\mathbf{v}} \geq 0$) and spherically symmetric distributions $\tilde{\mathcal{D}}$. Also we require a certain technical assumption on the the distribution, as specified by $\tau$, which are satisfied for standard spherically symmetric distributions, but is a bit non-trivial[3]. Finally, the assumption on the initialization ($\|\mathbf{w}_0 - \mathbf{v}\|^2 < 1$ and $b_{\mathbf{w}_0} \geq 0$) is much more restrictive (although see Remark 6.3 below). In contrast, the initialization assumption in the previous section holds with probability close to 1 with random initialization. On the positive side, the convergence rate does not depend on the initialization, i.e., here by initializing with any $\mathbf{w}_0$ such that $\|\mathbf{w}_0 - \mathbf{v}\|^2 < 1$ and $b_{\mathbf{w}_0} \geq 0$, we get a convergence rate that only depends on the input distribution. On the other hand, in Theorem 5.2, the convergence rate depends on the parameter $\delta$ which depends on the initialization. Also, the distribution is not necessarily bounded – we only require its fourth moment to be bounded.

**Remark 6.3** (Random initialization). *For $b_{\mathbf{v}} = 0$ the initialization assumption ($\|\mathbf{w}_0 - \mathbf{v}\|^2 < 1$) is satisfied with probability close to $1/2$ with standard initializations, see Lemma 5.1 from Yehudai and Shamir [2020]). For $b_{\mathbf{v}} > 0$, a similar argument applies if $b_{\mathbf{w}}$ is initialized close enough to $b_{\mathbf{v}}$.*

The proof of the theorem is quite different from the proofs in Section 5, and is more geometrical in nature, extending previously used techniques from Yehudai and Shamir [2020], Frei et al. [2020]. It contains two major parts: The first part is an extension of the methods from Yehudai and Shamir [2020] to the case of adding a bias term. Specifically, we show a lower bound on $\langle \nabla F(\mathbf{w}), \mathbf{w} - \mathbf{v} \rangle$, which depends on both the angle between $\tilde{\mathbf{w}}$ and $\tilde{\mathbf{v}}$, and the bias terms $b_{\mathbf{w}}$ and $b_{\mathbf{v}}$ (see Theorem A.2). This result implies that for suitable values of $\mathbf{w}$, gradient descent will decrease the distance from $\mathbf{v}$. The second part of the proof is showing that throughout the optimization process, $\mathbf{w}$ will stay in an area where we can apply the result above. Specifically, the intricate part is showing that the term $-\frac{b_{\mathbf{w}}}{\|\tilde{\mathbf{w}}\|}$ does not get too large. Note that due to Theorem 4.2, we know that keeping this term small means that $\mathbf{w}$ stays away from the cone of bad critical points which are not the global minimum. The full proof can be found in Appendix F.

## 7 Discussion

In this work we studied the problem of learning a single neuron with a bias term using gradient descent. We showed several negative results, indicating that adding a bias term makes the problem more difficult than without a bias term. Next, we gave a characterization of the critical points of the problem under some assumptions on the input distribution, showing that there is a manifold of critical points which are not the global minimum. We proved two convergence results using different techniques and under different assumptions. Finally, we showed that under mild assumptions on the input distribution, reaching the global minimum can be achieved by standard random initialization.

We emphasize that previous works studying the problem of a single neuron either considered non-standard algorithms (e.g. Isotron), or required assumptions on the input distribution which do not

---

[3]For example, for standard Gaussian distribution, we have that $\tau = \frac{2}{\pi} \approx 0.63$, hence we can take $\alpha = 4.5$, and $\beta = O(1)$. Since the distribution $\tilde{\mathcal{D}}$ is symmetric, we present the assumption w.l.o.g with respect to the first 2 coordinates.

allow a bias term. Hence, this is the first work we are aware of which gives positive and negative results on the problem of learning a single neuron with a bias term using gradient methods.

In this work we focused on the gradient descent algorithm. We believe that our results can also be extended to the commonly used SGD algorithm, using similar techniques to Yehudai and Shamir [2020], Shamir [2015], and leave it for future work. Another interesting future direction is analyzing other previously studied settings, but with the addition of a bias term. These settings can include convolutional networks, two layers neural networks, and agnostic learning of a single neuron.

### Acknowledgements

This research is supported in part by European Research Council (ERC) grant 754705.

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
