## Appendices

## A   Auxiliary Results

In this appendix we extend several key results from [23] for the case of adding a bias term. Specifically, we extend Theorem 4.2 from [23] which shows that under mild assumptions on the distribution, the gradient of the loss points in a good direction which depends on the angle between the learned vector $\mathbf{w}$ and the target $\mathbf{v}$. We also bound the volume of a certain set in $\mathbb{R}^2$, which can be seen as an extension of Lemma B.1 from [23].

**Lemma A.1.** *Let* $P = \{\mathbf{y} \in \mathbb{R}^2 : \mathbf{w}^\top \mathbf{y} > b, \mathbf{v}^\top \mathbf{y} > b, \|\mathbf{y}\| \leq \alpha\}$ *for* $b \in \mathbb{R}$ *and* $\mathbf{w}, \mathbf{v} \in \mathbb{R}^2$ *with* $\|\mathbf{w}\|, \|\mathbf{v}\| = 1$ *and* $\theta(\mathbf{w}, \mathbf{v}) \leq \pi - \delta$ *for* $\delta \in [0, \pi]$. *If* $b < \alpha \sin\left(\frac{\delta}{2}\right)$ *then* $Vol(P) \geq \frac{\left(\alpha \sin\left(\frac{\delta}{2}\right) - b\right)^2}{4 \sin\left(\frac{\delta}{2}\right)}$.

*Proof.* The volume of $P$ is smallest when the angle is exactly $\pi - \delta$, thus we can lower bound the volume by assuming that $\theta(\mathbf{w}, \mathbf{v}) = \pi - \delta$. Next, we can rotate to coordinates to consider without loss of generality the volume of the set

$$P' = \left\{(y_1, y_2) \in \mathbb{R}^2 : \theta((y_1, y_2 - b'), \mathbf{e}_2) \leq \delta/2, \|(y_1, y_2)\| \leq \alpha\right\},$$

where $b' = \frac{b}{\sin(\delta/2)}$ and $\mathbf{e}_2 = (0, 1)$. Let $P'' = \{(x, y) \in \mathbb{R}^2 : x^2 + (y - b')^2 \leq (\alpha - b')^2\}$ be the disc of radius $\alpha - b'$ around the point $(0, b')$. It is enough to bound the volume of $P' \cap P''$. We define the rectangular sets:

$$P_1 = \left[\frac{(\alpha - b')}{2} \sin\left(\frac{\delta}{4}\right), (\alpha - b') \sin\left(\frac{\delta}{4}\right)\right] \times \left[b' + \frac{(\alpha - b')}{2} \cos\left(\frac{\delta}{4}\right), b' + (\alpha - b') \cos\left(\frac{\delta}{4}\right)\right]$$

$$P_2 = \left[-(\alpha - b') \sin\left(\frac{\delta}{4}\right), -\frac{(\alpha - b')}{2} \sin\left(\frac{\delta}{4}\right)\right] \times \left[b' + \frac{(\alpha - b')}{2} \cos\left(\frac{\delta}{4}\right), b' + (\alpha - b') \cos\left(\frac{\delta}{4}\right)\right]$$

See Figure 1 for an illustration. We have that $P_1, P_2 \subseteq P' \cap P''$. We will show it for $P_1$, the same argument also works for $P_2$. First, $P_1 \subseteq P''$ is immediate by the definition of the two sets. For $P'$, the straight line in the boundary of $P'$ is defined by $y_2 = b' + y_1 \cdot \frac{\cos\left(\frac{\delta}{2}\right)}{\sin\left(\frac{\delta}{2}\right)}$. It can be seen that each vertex of the rectangle $P_1$, is above this line. Moreover, the norm of each vertex of $P_1$ is at most $\alpha$. Hence all the vertices are inside $P'$, which means that $P_1 \subseteq P'$. In total we get:

$$\mathrm{Vol}(P) \geq \mathrm{Vol}(P' \cap P'') \geq \mathrm{Vol}(P_1 \cup P_2)$$
$$= \frac{(\alpha - b')^2}{2} \sin\left(\frac{\delta}{4}\right) \cos\left(\frac{\delta}{4}\right)$$
$$= \frac{\left(\alpha \sin\left(\frac{\delta}{2}\right) - b\right)^2}{4 \sin\left(\frac{\delta}{2}\right)}$$

$\square$

**Theorem A.2.** *Let* $\mathbf{w}, \mathbf{v} \in \mathbb{R}^{d+1}$, *denote by* $\tilde{\mathbf{w}}, \tilde{\mathbf{v}}$ *their first* $d$ *coordinates and by* $b_\mathbf{w}, b_\mathbf{v}$ *their last coordinate. Assume that* $\theta(\tilde{\mathbf{w}}, \tilde{\mathbf{v}}) \leq \pi - \delta$ *for some* $\delta \in [0, \pi)$, *and that the distribution* $\mathcal{D}$ *is such that its first* $d$ *coordinates satisfy Assumption 4.1 (1) from [23], and that its last coordinate is a constant* 1. *Denote* $b' = \max\{-b_\mathbf{w}/\|\tilde{\mathbf{w}}\|, -b_\mathbf{v}/\|\tilde{\mathbf{v}}\|, 0\} \cdot \frac{1}{\sin\left(\frac{\delta}{2}\right)}$, *and assume that* $b' < \alpha$, *then:*

$$\langle \nabla F(\mathbf{w}), \mathbf{w} - \mathbf{v} \rangle \geq \frac{(\alpha - b')^4 \sin\left(\frac{\delta}{4}\right)^3 \beta}{8^4} \cdot \min\left\{1, \frac{1}{\alpha^2}\right\} \|\mathbf{w} - \mathbf{v}\|^2$$

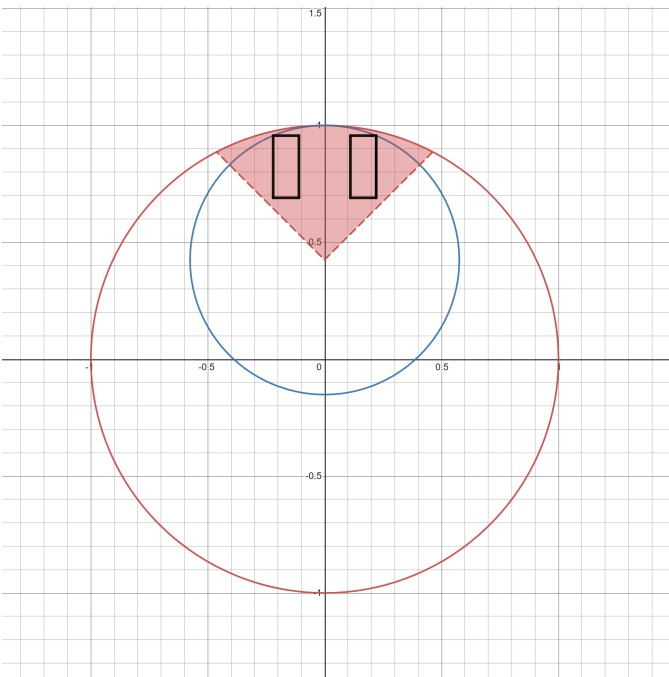

Figure 1: An illustration of the set $P'$ (in red), the circle $P''$ (in blue) and the two rectangles $P_1, P_2$ (in black), for the case of $\delta = \pi/2$, $\alpha = 1$ and $b = 0.3$. For $b = 0$, $P'$ would be a pie slice, and the blue circle $P''$ will coincide with the red circle.

*Proof.* Let $\tilde{\mathbf{x}}$ be the first $d$ coordinates of $\mathbf{x}$. We have that:

$$
\begin{aligned}
\langle \nabla F(\mathbf{w}), \mathbf{w} - \mathbf{v} \rangle &= \mathbb{E}_{\mathbf{x} \sim \mathcal{D}} \left[ \sigma'(\mathbf{w}^\top \mathbf{x})(\sigma(\mathbf{w}^\top \mathbf{x}) - \sigma(\mathbf{v}^\top \mathbf{x}))(\mathbf{w}^\top \mathbf{x} - \mathbf{v}^\top \mathbf{x}) \right] \\
&\geq \mathbb{E}_{\mathbf{x} \sim \mathcal{D}} \left[ \mathbb{1}(\mathbf{w}^\top \mathbf{x} > 0, \mathbf{v}^\top \mathbf{x} > 0)(\mathbf{w}^\top \mathbf{x} - \mathbf{v}^\top \mathbf{x})^2 \right] \\
&= \|\mathbf{w} - \mathbf{v}\|^2 \cdot \mathbb{E}_{\mathbf{x} \sim \mathcal{D}} \left[ \mathbb{1}(\tilde{\mathbf{w}}^\top \tilde{\mathbf{x}} > -b_\mathbf{w}, \tilde{\mathbf{v}}^\top \tilde{\mathbf{x}} > -b_\mathbf{v})((\overline{\mathbf{w} - \mathbf{v}})^\top \mathbf{x})^2 \right] \\
&\geq \|\mathbf{w} - \mathbf{v}\|^2 \cdot \inf_{\mathbf{u} \in \mathrm{span}\{\mathbf{w}, \mathbf{v}\}, \|\mathbf{u}\| = 1} \mathbb{E}_{\mathbf{x} \sim \mathcal{D}} \left[ \mathbb{1}(\tilde{\mathbf{w}}^\top \tilde{\mathbf{x}} > -b_\mathbf{w}, \tilde{\mathbf{v}}^\top \tilde{\mathbf{x}} > -b_\mathbf{v})(\mathbf{u}^\top \mathbf{x})^2 \right]
\end{aligned}
$$

Let $b = \max\{-b_\mathbf{w}/\|\tilde{\mathbf{w}}\|, -b_\mathbf{v}/\|\tilde{\mathbf{v}}\|, 0\}$, then we can bound the above equation by:

$$
\begin{aligned}
&\|\mathbf{w} - \mathbf{v}\|^2 \cdot \inf_{\mathbf{u} \in \mathrm{span}\{\mathbf{w}, \mathbf{v}\}, \|\mathbf{u}\| = 1} \mathbb{E}_{\mathbf{x} \sim \mathcal{D}} \left[ \mathbb{1}(\overline{\tilde{\mathbf{w}}}^\top \tilde{\mathbf{x}} > b, \overline{\tilde{\mathbf{v}}}^\top \tilde{\mathbf{x}} > b)(\mathbf{u}^\top \mathbf{x})^2 \right] \\
&\geq \|\mathbf{w} - \mathbf{v}\|^2 \cdot \inf_{\mathbf{u} \in \mathrm{span}\{\mathbf{w}, \mathbf{v}\}, \|\mathbf{u}\| = 1} \mathbb{E}_{\tilde{\mathbf{x}} \sim \tilde{\mathcal{D}}} \left[ \mathbb{1}(\overline{\tilde{\mathbf{w}}}^\top \tilde{\mathbf{x}} > b, \overline{\tilde{\mathbf{v}}}^\top \tilde{\mathbf{x}} > b, \|\tilde{\mathbf{x}}\| \leq \alpha)(\tilde{\mathbf{u}}^\top \tilde{\mathbf{x}} + b_\mathbf{u})^2 \right] \quad (4)
\end{aligned}
$$

Here $b_\mathbf{u}$ is the bias term of $\mathbf{u}$, $\tilde{\mathbf{u}}$ are the first $d$ coordinates of $\mathbf{u}$ and $\tilde{\mathcal{D}}$ is the marginal distribution of $\mathbf{x}$ on its first $d$ coordinates. Note that since the last coordinate represents the bias term, then the distribution on the last coordinate of $\mathbf{x}$ is a constant 1. The condition that $\|\mathbf{u}\| = 1$ (equivalently $\|\mathbf{u}\|^2 = 1$) translates to $\|\tilde{\mathbf{u}}\|^2 + b_\mathbf{u}^2 = 1$.

Our goal is to bound the term inside the infimum. Note that the expression inside the distribution depends just on inner products of $\tilde{\mathbf{x}}$ with $\tilde{\mathbf{w}}$ or $\tilde{\mathbf{v}}$, hence we can consider the marginal distribution $\mathcal{D}_{\tilde{\mathbf{w}}, \tilde{\mathbf{v}}}$ of $\tilde{\mathbf{x}}$ on the 2-dimensional subspace spanned by $\tilde{\mathbf{w}}$ and $\tilde{\mathbf{v}}$ (with density function $p_{\tilde{\mathbf{w}}, \tilde{\mathbf{v}}}$). Let $\hat{\mathbf{w}}$ and $\hat{\mathbf{v}}$ be the projections of $\tilde{\mathbf{w}}$ and $\tilde{\mathbf{v}}$ on that subspace. Let $P = \{\mathbf{y} \in \mathbb{R}^2 : \overline{\hat{\mathbf{w}}}^\top \mathbf{y} > b, \overline{\hat{\mathbf{v}}}^\top \mathbf{y} > b, \|\mathbf{y}\| \leq \alpha\}$, then we can bound Eq. (4) with:

$$\|\mathbf{w}-\mathbf{v}\|^2 \cdot \inf_{\mathbf{u}\in\mathbb{R}^2, b_{\mathbf{u}}\in\mathbb{R}:\|\mathbf{u}\|^2+b_{\mathbf{u}}^2=1} \mathbb{E}_{\mathbf{y}\sim\mathcal{D}_{\tilde{\mathbf{w}},\tilde{\mathbf{v}}}} \left[\mathbb{1}(\mathbf{y}\in P)\cdot(\mathbf{u}^\top\mathbf{y}+b_{\mathbf{u}})^2\right]$$

$$= \|\mathbf{w}-\mathbf{v}\|^2 \cdot \inf_{\mathbf{u}\in\mathbb{R}^2, b_{\mathbf{u}}\in\mathbb{R}:\|\mathbf{u}\|^2+b_{\mathbf{u}}^2=1} \int_{\mathbf{y}\in\mathbb{R}^2} \mathbb{1}(\mathbf{y}\in P)\cdot(\mathbf{u}^\top\mathbf{y}+b_{\mathbf{u}})^2 p_{\tilde{\mathbf{w}},\tilde{\mathbf{v}}}(\mathbf{y})d\mathbf{y}$$

$$\geq \beta\|\mathbf{w}-\mathbf{v}\|^2 \cdot \inf_{\mathbf{u}\in\mathbb{R}^2, b_{\mathbf{u}}\in\mathbb{R}:\|\mathbf{u}\|^2+b_{\mathbf{u}}^2=1} \int_{\mathbf{y}\in P} (\mathbf{u}^\top\mathbf{y}+b_{\mathbf{u}})^2 d\mathbf{y}$$

Combining with Proposition A.3 finishes the proof $\qquad\square$

**Proposition A.3.** *Let* $P = \{\mathbf{y}\in\mathbb{R}^2 : \overline{\hat{\mathbf{w}}}^\top\mathbf{y} > b, \overline{\hat{\mathbf{v}}}^\top\mathbf{y} > b, \|\mathbf{y}\| \leq \alpha\}$ *for* $b\in\mathbb{R}$ *and* $\hat{\mathbf{w}}, \hat{\mathbf{v}}\in\mathbb{R}^2$
*with* $\theta(\hat{\mathbf{w}}, \hat{\mathbf{v}}) \leq \pi - \delta$ *for* $\delta \in [0,\pi]$. *Then*

$$\inf_{\mathbf{u}\in\mathbb{R}^2, b_{\mathbf{u}}\in\mathbb{R}:\|\mathbf{u}\|^2+b_{\mathbf{u}}^2=1} \int_{\mathbf{y}\in P} (\mathbf{u}^\top\mathbf{y}+b_{\mathbf{u}})^2 d\mathbf{y} \geq \frac{(\alpha-b')^4 \sin\left(\frac{\delta}{4}\right)^3}{8^4} \cdot \min\left\{1, \frac{1}{\alpha^2}\right\}$$

*for* $b' = \frac{b}{\sin\left(\frac{\delta}{2}\right)}$.

*Proof.* As in the proof of Lemma A.1, we consider the rectangular sets:

$$P_1 = \left[\frac{(\alpha-b')}{2}\sin\left(\frac{\delta}{4}\right), (\alpha-b')\sin\left(\frac{\delta}{4}\right)\right] \times \left[b' + \frac{(\alpha-b')}{2}\cos\left(\frac{\delta}{4}\right), b' + (\alpha-b')\cos\left(\frac{\delta}{4}\right)\right]$$

$$P_2 = \left[-(\alpha-b')\sin\left(\frac{\delta}{4}\right), -\frac{(\alpha-b')}{2}\sin\left(\frac{\delta}{4}\right)\right] \times \left[b' + \frac{(\alpha-b')}{2}\cos\left(\frac{\delta}{4}\right), b' + (\alpha-b')\cos\left(\frac{\delta}{4}\right)\right]$$

with $b' = \frac{b}{\sin(\delta/2)}$. Since we have $P_1 \cup P_2 \subseteq P$, and the function inside the integral is positive, we can lower bound the target integral by integrating only over $P_1 \cup P_2$. Now we have:

$$\inf_{\mathbf{u}\in\mathbb{R}^2, b_{\mathbf{u}}\in\mathbb{R}:\|\mathbf{u}\|^2+b_{\mathbf{u}}^2=1} \int_{\mathbf{y}\in P} (\mathbf{u}^\top\mathbf{y}+b_{\mathbf{u}})^2 d\mathbf{y}$$

$$\geq \inf_{u_1,u_2,b_{\mathbf{u}}\in\mathbb{R}:u_1^2+u_2^2+b_{\mathbf{u}}^2=1} \int_{\mathbf{y}\in P_1\cup P_2} (u_1 y_1 + u_2 y_2 + b_{\mathbf{u}})^2 d\mathbf{y}$$

$$= \inf_{u_1,u_2,b_{\mathbf{u}}\in\mathbb{R}:u_1^2+u_2^2+b_{\mathbf{u}}^2=1} \int_{\mathbf{y}\in P_1\cup P_2} (u_1 y_1)^2 d\mathbf{y} + \int_{\mathbf{y}\in P_1\cup P_2} (u_2 y_2 + b_{\mathbf{u}})^2 d\mathbf{y} + \int_{\mathbf{y}\in P_1\cup P_2} 2u_1 y_1 (u_2 y_2 + b_{\mathbf{u}}) d\mathbf{y}$$

$$= \inf_{u_1,u_2,b_{\mathbf{u}}\in\mathbb{R}:u_1^2+u_2^2+b_{\mathbf{u}}^2=1} \int_{\mathbf{y}\in P_1\cup P_2} (u_1 y_1)^2 d\mathbf{y} + \int_{\mathbf{y}\in P_1\cup P_2} (u_2 y_2 + b_{\mathbf{u}})^2 d\mathbf{y}$$

where in the last equality we used that $P_1\cup P_2$ are symmetric around the $y_2$ axis, i.e. $(y_1, y_2) \in P_1\cup P_2$ iff $(-y_1, y_2) \in P_1 \cup P_2$. By the condition that $u_1^2 + u_2^2 + b_{\mathbf{u}}^2 = 1$ we know that either $u_1^2 \geq \frac{1}{2}$ or $u_2^2 + b_{\mathbf{u}}^2 \geq \frac{1}{2}$. Using that both integrals above are positive, we can lower bound:

$$\inf_{u_1,u_2,b_{\mathbf{u}}\in\mathbb{R}:u_1^2+u_2^2+b_{\mathbf{u}}^2=1} \int_{\mathbf{y}\in P_1\cup P_2} (u_1 y_1)^2 d\mathbf{y} + \int_{\mathbf{y}\in P_1\cup P_2} (u_2 y_2 + b_{\mathbf{u}})^2 d\mathbf{y}$$

$$\geq \min\left\{\frac{1}{2}\int_{\mathbf{y}\in P_1\cup P_2} y_1^2 d\mathbf{y}, \inf_{u_2,u_3\in\mathbb{R}:u_2^2+u_3^2=\frac{1}{2}} \int_{\mathbf{y}\in P_1\cup P_2} (u_2 y_2 + u_3)^2 d\mathbf{y}\right\} .$$

We will now lower bound both terms in the above equation. For the first term, note that for every $\mathbf{y} \in P_1 \cup P_2$ we have that $|y_1| \geq \frac{(\alpha - b')}{2} \sin\left(\frac{\delta}{4}\right)$. Hence we have:

$$\frac{1}{2} \int_{\mathbf{y} \in P_1 \cup P_2} y_1^2 d\mathbf{y} \geq$$

$$\geq \frac{1}{2} \int_{\mathbf{y} \in P_1 \cup P_2} \frac{(\alpha - b')^2}{4} \sin\left(\frac{\delta}{4}\right)^2 d\mathbf{y}$$

$$= \frac{(\alpha - b')^2}{8} \sin\left(\frac{\delta}{4}\right)^2 \cdot \frac{(\alpha - b')^2}{2} \sin\left(\frac{\delta}{4}\right) \cos\left(\frac{\delta}{4}\right)$$

$$\geq \frac{(\alpha - b')^4}{16\sqrt{2}} \sin\left(\frac{\delta}{4}\right)^3 \tag{5}$$

where in the last inequality we used that $\delta \in [0, \pi]$, hence $\delta/4 \in [0, \pi/4]$.

For the second term we have:

$$\inf_{u_2, u_3 \in \mathbb{R} : u_2^2 + u_3^2 = \frac{1}{2}} \int_{\mathbf{y} \in P_1 \cup P_2} (u_2 y_2 + u_3)^2 d\mathbf{y}$$

$$= \inf_{u \in \left[-\frac{1}{\sqrt{2}}, \frac{1}{\sqrt{2}}\right]} \int_{\mathbf{y} \in P_1 \cup P_2} \left(u y_2 + \sqrt{\frac{1}{2} - u^2}\right)^2 d\mathbf{y}$$

$$= (\alpha - b') \sin\left(\frac{\delta}{4}\right) \inf_{u \in \left[-\frac{1}{\sqrt{2}}, \frac{1}{\sqrt{2}}\right]} \int_{y_2 \in C} \left(u y_2 + \sqrt{\frac{1}{2} - u^2}\right)^2 dy_2 . \tag{6}$$

The last equality is given by changing the order of integration into integral over $y_2$ and then over $y_1$, denoting the interval $C = \left[b' + \frac{(\alpha - b')}{2} \cos\left(\frac{\delta}{4}\right), b' + (\alpha - b') \cos\left(\frac{\delta}{4}\right)\right]$, and noting that the term inside the integral does not depend on $y_1$.

Fix some $u \in \left[-\frac{1}{\sqrt{2}}, \frac{1}{\sqrt{2}}\right]$. If $u = 0$, then we can bound Eq. (6) by $\frac{(\alpha - b')^2}{4} \sin\left(\frac{\delta}{4}\right) \cos\left(\frac{\delta}{4}\right)$. Assume $u \neq 0$, we split into cases and bound the term inside the integral:

**Case I:** $\left|\frac{\sqrt{\frac{1}{2} - u^2}}{u}\right| \geq b' + \frac{3}{4} \cdot (\alpha - b') \cos\left(\frac{\delta}{4}\right)$. In this case, solving the inequality for $u$ we have $|u| \leq \sqrt{\frac{1}{2 + 2\left(b' + \frac{3}{4}(\alpha - b')\cos\left(\frac{\delta}{4}\right)\right)^2}}$. Hence, we can also bound:

$$\sqrt{\frac{1}{2} - u^2} \geq \sqrt{\frac{1}{2} - \frac{1}{2 + 2\left(b' + \frac{3}{4}(\alpha - b')\cos\left(\frac{\delta}{4}\right)\right)^2}} = \sqrt{\frac{\left(b' + \frac{3}{4}(\alpha - b')\cos\left(\frac{\delta}{4}\right)\right)^2}{2 + 2\left(b' + \frac{3}{4}(\alpha - b')\cos\left(\frac{\delta}{4}\right)\right)^2}}$$

In particular, for every $y_2 \in \left[b' + \frac{(\alpha - b')}{2} \cos\left(\frac{\delta}{4}\right), b' + \frac{5(\alpha - b')}{8} \cos\left(\frac{\delta}{4}\right)\right]$ we get that:

$$\left|u y_2 + \sqrt{\frac{1}{2} - u^2}\right|$$

$$\geq \left|\sqrt{\frac{\left(b' + \frac{3}{4}(\alpha - b')\cos\left(\frac{\delta}{4}\right)\right)^2}{2 + 2\left(b' + \frac{3}{4}(\alpha - b')\cos\left(\frac{\delta}{4}\right)\right)^2}} - \sqrt{\frac{\left(b' + \frac{5}{8}(\alpha - b')\cos\left(\frac{\delta}{4}\right)\right)^2}{2 + 2\left(b' + \frac{3}{4}(\alpha - b')\cos\left(\frac{\delta}{4}\right)\right)^2}}\right|$$

$$\geq \frac{(\alpha - b')\cos\left(\frac{\delta}{4}\right)}{8\sqrt{2 + 2\left(b' + (\alpha - b')\cos\left(\frac{\delta}{4}\right)\right)^2}}$$

**Case II:** $\left|\frac{\sqrt{\frac{1}{2}-u^2}}{u}\right| < b' + \frac{3}{4} \cdot (\alpha - b') \cos\left(\frac{\delta}{4}\right)$. Using the same reasoning as above, we get for every $y_2 \in \left[b' + \frac{7(\alpha-b')}{8}\cos\left(\frac{\delta}{4}\right), b' + (\alpha - b')\cos\left(\frac{\delta}{4}\right)\right]$ that:

$$\left|uy_2 + \sqrt{\frac{1}{2} - u^2}\right| \geq \frac{(\alpha - b')\cos\left(\frac{\delta}{4}\right)}{8\sqrt{2 + 2\left(b' + (\alpha - b')\cos\left(\frac{\delta}{4}\right)\right)^2}}$$

Combining the above cases with Eq. (6) we get that:

$$\inf_{u_2, u_3 \in \mathbb{R}: u_2^2 + u_3^2 = \frac{1}{2}} \int_{\mathbf{y} \in P_1 \cup P_2} (u_2 y_2 + u_3)^2 d\mathbf{y}$$

$$\geq (\alpha - b')\sin\left(\frac{\delta}{4}\right) \int_{y_2 \in C} \frac{(\alpha - b')^2 \cos\left(\frac{\delta}{4}\right)^2}{8^2\left(2 + 2\left(b' + (\alpha - b')\cos\left(\frac{\delta}{4}\right)\right)^2\right)} dy_2$$

$$\geq \frac{(\alpha - b')^4 \cos\left(\frac{\delta}{4}\right)^3 \sin\left(\frac{\delta}{4}\right)}{2 \cdot 8^2 \left(2 + 2\left(b' + (\alpha - b')\cos\left(\frac{\delta}{4}\right)\right)\right)^2}$$

$$\geq \frac{(\alpha - b')^4 \sin\left(\frac{\delta}{4}\right)^3}{2 \cdot 8^2 \sqrt{2}\left(2 + 2\left(b' + (\alpha - b')\cos\left(\frac{\delta}{4}\right)\right)\right)^2}$$

$$\geq \frac{(\alpha - b')^4 \sin\left(\frac{\delta}{4}\right)^3}{8^4} \cdot \min\left\{1, \frac{1}{\alpha^2}\right\} \tag{7}$$

where in the second inequality we used that for $\delta \in [0, \pi]$ we have $\sin\left(\frac{\delta}{4}\right) \leq \cos\left(\frac{\delta}{4}\right)$, and in the last inequality we used that $b' \leq \alpha$, and $(\alpha - b')\cos\left(\frac{\delta}{4}\right) \leq \alpha$. Combining Eq. (5) with Eq. (7) finishes the proof.

$\square$

# B  Proofs from Section 3

## B.1  Proof of Theorem 3.1

Let $\epsilon > 0$, for the input distribution, we consider the uniform distribution on the ball of radius $\epsilon$. Let $b_{\mathbf{w}}$ be the last coordinate of $\mathbf{w}$, and denote by $\tilde{\mathbf{w}}, \tilde{\mathbf{x}}$ the first $d$ coordinates of $\mathbf{w}$ and $\mathbf{x}$. Using the assumption on the initialization of $\mathbf{w}_0$ and on the boundness of the distribution $\tilde{\mathcal{D}}$ we have:

$$|\langle \tilde{\mathbf{w}}_0, \tilde{\mathbf{x}} \rangle| \leq \|\tilde{\mathbf{w}}_0\|\|\tilde{\mathbf{x}}\| \leq \epsilon\sqrt{d}.$$

Since $b_{\mathbf{w}_0}$ is also initialized with $U([-1, 1])$, w.p $> 1/2 - \epsilon\sqrt{d}$ we have that $b_{\mathbf{w}_0} < -\epsilon\sqrt{d}$. If this event happens, since the activation is ReLU we get that $\sigma'(\langle \mathbf{w}_0, \mathbf{x} \rangle) = \mathbb{1}(\langle \tilde{\mathbf{w}}_0, \tilde{\mathbf{x}} \rangle + b_{\mathbf{w}_0} > 0) = 0$ for every $\tilde{\mathbf{x}}$ in the support of the distribution. Using Eq. (3) we get that $\nabla F(\mathbf{w}_0) = 0$, hence gradient flow will get stuck at its initial value.

## B.2  Proof of Theorem 3.2

**Lemma B.1.** *Let $\mathbf{w} \in \mathbb{R}^{d+1}$ such that $b_{\mathbf{w}} = 0$, $\tilde{w}_1 < -\frac{4}{\sqrt{d}}$, and $\|\tilde{\mathbf{w}}_{2:d}\| \leq 2\sqrt{d}$. Then,*

$$\Pr_{\mathbf{x} \sim \mathcal{D}}\left[\mathbf{w}^\top \mathbf{x} \geq 0, \mathbf{v}^\top \mathbf{x} \geq 0\right] = 0.$$

*Proof.* If $\mathbf{v}^\top \mathbf{x} \geq 0$ then $x_1 \geq r - \frac{r}{2d^2}$ and hence $x_1^2 \geq r^2 - \frac{r^2}{d^2}$. Since we also have $\|\tilde{\mathbf{x}}\| \leq r$ then

$$\|\tilde{\mathbf{x}}_{2:d}\|^2 = \|\tilde{\mathbf{x}}\|^2 - x_1^2 \leq r^2 - \left(r^2 - \frac{r^2}{d^2}\right) = \frac{r^2}{d^2}.$$

Hence,

$$\Pr_{\mathbf{x}\sim\mathcal{D}}\left[\mathbf{w}^\top\mathbf{x}\geq 0, \mathbf{v}^\top\mathbf{x}\geq 0\right] \leq \Pr_{\mathbf{x}\sim\mathcal{D}}\left[\mathbf{w}^\top\mathbf{x}\geq 0, x_1 \geq r - \frac{r}{2d^2}, \|\tilde{\mathbf{x}}_{2:d}\| \leq \frac{r}{d}\right]\,.$$

Since $b_{\mathbf{w}} = 0$, $\|\tilde{\mathbf{w}}_{2:d}\| \leq 2\sqrt{d}$ and $\tilde{w}_1 < -\frac{4}{\sqrt{d}}$, then for every $\tilde{\mathbf{x}} \in \mathcal{B}$ such that $x_1 \geq r - \frac{r}{2d^2} \geq \frac{r}{2}$ and $\|\tilde{\mathbf{x}}_{2:d}\| \leq \frac{r}{d}$ we have

$$\mathbf{w}^\top\mathbf{x} = \tilde{\mathbf{w}}^\top\tilde{\mathbf{x}} = \tilde{w}_1\tilde{x}_1 + \langle\tilde{\mathbf{w}}_{2:d}, \tilde{\mathbf{x}}_{2:d}\rangle < -\frac{4}{\sqrt{d}}\cdot\frac{r}{2} + 2\sqrt{d}\cdot\frac{r}{d} = 0\,.$$

Therefore, $\Pr_{\mathbf{x}\sim\mathcal{D}}\left[\mathbf{w}^\top\mathbf{x}\geq 0, \mathbf{v}^\top\mathbf{x}\geq 0\right] = 0$. $\qquad\square$

**Lemma B.2.** *With probability $\frac{1}{2} - o_d(1)$ over the choice of $\mathbf{w}_0$, we have*

$$\Pr_{\mathbf{x}\sim\mathcal{D}}\left[\mathbf{w}_0^\top\mathbf{x}\geq 0, \mathbf{v}^\top\mathbf{x}\geq 0\right] = 0\,.$$

*Proof.* Let $\mathbf{w} \in \mathbb{R}^{d+1}$ such that $b_{\mathbf{w}} = 0$ and $\tilde{\mathbf{w}} \sim \mathcal{N}(\mathbf{0}, I_d)$. Since $\tilde{w}_1$ has a standard normal distribution, then we have $\tilde{w}_1 < -\frac{4}{\sqrt{d}}$ with probability $\frac{1}{2} - o_d(1)$. Moreover, note that $\|\tilde{\mathbf{w}}_{2:d}\|^2$ has a chi-square distribution and the probability of $\|\tilde{\mathbf{w}}_{2:d}\|^2 \leq 4d$ is $1 - o_d(1)$. Hence, by Lemma B.1, with probability $\frac{1}{2} - o_d(1)$ over the choice of $\mathbf{w}$, we have

$$\Pr_{\mathbf{x}\sim\mathcal{D}}\left[\mathbf{w}^\top\mathbf{x}\geq 0, \mathbf{v}^\top\mathbf{x}\geq 0\right] = 0\,.$$

Therefore,

$$\Pr_{\mathbf{x}\sim\mathcal{D}}\left[\rho\frac{\mathbf{w}^\top}{\|\mathbf{w}\|}\mathbf{x}\geq 0, \mathbf{v}^\top\mathbf{x}\geq 0\right] = \Pr_{\mathbf{x}\sim\mathcal{D}}\left[\mathbf{w}^\top\mathbf{x}\geq 0, \mathbf{v}^\top\mathbf{x}\geq 0\right] = 0\,.$$

Since $\rho\frac{\mathbf{w}^\top}{\|\mathbf{w}\|}$ has the distribution of $\mathbf{w}_0$, the lemma follows. $\qquad\square$

**Lemma B.3.** *Assume that $\mathbf{w}_0$ satisfies $\Pr_{\mathbf{x}\sim\mathcal{D}}\left[\mathbf{w}_0^\top\mathbf{x}\geq 0, \mathbf{v}^\top\mathbf{x}\geq 0\right] = 0$. Let $\gamma > 0$ and let $\mathbf{w} \in \mathbb{R}^{d+1}$ such that $\tilde{\mathbf{w}} = \gamma\tilde{\mathbf{w}}_0$, and $b_{\mathbf{w}} \leq 0$. Then, $\Pr_{\mathbf{x}\sim\mathcal{D}}\left[\mathbf{w}^\top\mathbf{x}\geq 0, \mathbf{v}^\top\mathbf{x}\geq 0\right] = 0$. Moreover, we have*

- *If $-\frac{b_{\mathbf{w}}}{\|\tilde{\mathbf{w}}\|} < r$, then $\frac{d\tilde{\mathbf{w}}}{dt} = -s\tilde{\mathbf{w}}$ for some $s > 0$, and $\frac{db_{\mathbf{w}}}{dt} < 0$.*

- *If $-\frac{b_{\mathbf{w}}}{\|\tilde{\mathbf{w}}\|} \geq r$, then $\frac{d\tilde{\mathbf{w}}}{dt} = \mathbf{0}$ and $\frac{db_{\mathbf{w}}}{dt} = 0$.*

*Proof.* For every $\mathbf{x}$ we have: If $\mathbf{w}^\top\mathbf{x} = \gamma\tilde{\mathbf{w}}_0^\top\tilde{\mathbf{x}} + b_{\mathbf{w}} \geq 0$ then $\gamma\tilde{\mathbf{w}}_0^\top\tilde{\mathbf{x}} \geq 0$, and therefore $\mathbf{w}_0^\top\mathbf{x} = \tilde{\mathbf{w}}_0^\top\tilde{\mathbf{x}} \geq 0$. Thus

$$\Pr_{\mathbf{x}\sim\mathcal{D}}\left[\mathbf{w}^\top\mathbf{x}\geq 0, \mathbf{v}^\top\mathbf{x}\geq 0\right] \leq \Pr_{\mathbf{x}\sim\mathcal{D}}\left[\mathbf{w}_0^\top\mathbf{x}\geq 0, \mathbf{v}^\top\mathbf{x}\geq 0\right] = 0\,. \tag{8}$$

We have

$$\begin{aligned}
-\frac{d\tilde{\mathbf{w}}}{dt} = \nabla_{\tilde{\mathbf{w}}}F(\mathbf{w}) &= \mathbb{E}_{\mathbf{x}}\left(\sigma(\mathbf{w}^\top\mathbf{x}) - \sigma(\mathbf{v}^\top\mathbf{x})\right)\sigma'(\mathbf{w}^\top\mathbf{x})\tilde{\mathbf{x}} \\
&= \mathbb{E}_{\mathbf{x}}\left(\sigma(\mathbf{w}^\top\mathbf{x}) - \sigma(\mathbf{v}^\top\mathbf{x})\right)\mathbb{1}(\mathbf{w}^\top\mathbf{x}\geq 0)\tilde{\mathbf{x}} \\
&= \mathbb{E}_{\mathbf{x}}\left(\sigma(\mathbf{w}^\top\mathbf{x}) - \sigma(\mathbf{v}^\top\mathbf{x})\right)\mathbb{1}(\mathbf{w}^\top\mathbf{x}\geq 0, \mathbf{v}^\top\mathbf{x} < 0)\tilde{\mathbf{x}} \\
&\quad + \mathbb{E}_{\mathbf{x}}\left(\sigma(\mathbf{w}^\top\mathbf{x}) - \sigma(\mathbf{v}^\top\mathbf{x})\right)\mathbb{1}(\mathbf{w}^\top\mathbf{x}\geq 0, \mathbf{v}^\top\mathbf{x}\geq 0)\tilde{\mathbf{x}} \\
&\overset{(Eq.\ (8))}{=} \mathbb{E}_{\mathbf{x}}\left(\sigma(\mathbf{w}^\top\mathbf{x}) - \sigma(\mathbf{v}^\top\mathbf{x})\right)\mathbb{1}(\mathbf{w}^\top\mathbf{x}\geq 0, \mathbf{v}^\top\mathbf{x} < 0)\tilde{\mathbf{x}} \\
&= \mathbb{E}_{\mathbf{x}}\left(\sigma(\mathbf{w}^\top\mathbf{x})\right)\tilde{\mathbf{x}} \\
&= \mathbb{E}_{\tilde{\mathbf{x}}}\mathbb{1}(\tilde{\mathbf{w}}^\top\tilde{\mathbf{x}} > -b_{\mathbf{w}})(\tilde{\mathbf{w}}^\top\tilde{\mathbf{x}} + b_{\mathbf{w}})\tilde{\mathbf{x}}\,.
\end{aligned}$$

If $-\frac{b_{\mathbf{w}}}{\|\tilde{\mathbf{w}}\|} \geq r$ then for every $\tilde{\mathbf{x}} \in \mathcal{B}$ we have $\tilde{\mathbf{w}}^\top \tilde{\mathbf{x}} \leq \|\tilde{\mathbf{w}}\| r \leq -b_{\mathbf{w}}$ and hence $\frac{d\tilde{\mathbf{w}}}{dt} = \mathbf{0}$. Note that if $-\frac{b_{\mathbf{w}}}{\|\tilde{\mathbf{w}}\|} < r$, i.e., $\|\tilde{\mathbf{w}}\| r > -b_{\mathbf{w}}$, then $\Pr_{\tilde{\mathbf{x}}}\left[\tilde{\mathbf{w}}^\top \tilde{\mathbf{x}} > -b_{\mathbf{w}}\right] > 0$. Since $\tilde{\mathcal{D}}$ is spherically symmetric, then we obtain $\frac{d\tilde{\mathbf{w}}}{dt} = -s\tilde{\mathbf{w}}$ for some $s > 0$.

Next, we have

$$
\begin{aligned}
-\frac{db_{\mathbf{w}}}{dt} = \nabla_{b_{\mathbf{w}}} F(\mathbf{w}) &= \mathbb{E}_{\mathbf{x}}\left(\sigma(\mathbf{w}^\top \mathbf{x}) - \sigma(\mathbf{v}^\top \mathbf{x})\right) \sigma'(\mathbf{w}^\top \mathbf{x}) \cdot 1 \\
&= \mathbb{E}_{\mathbf{x}}\left(\sigma(\mathbf{w}^\top \mathbf{x}) - \sigma(\mathbf{v}^\top \mathbf{x})\right) \mathbb{1}(\mathbf{w}^\top \mathbf{x} \geq 0) \\
&\overset{(Eq.\ (8))}{=} \mathbb{E}_{\mathbf{x}}\left(\sigma(\mathbf{w}^\top \mathbf{x}) - \sigma(\mathbf{v}^\top \mathbf{x})\right) \mathbb{1}(\mathbf{w}^\top \mathbf{x} \geq 0, \mathbf{v}^\top \mathbf{x} < 0) \\
&= \mathbb{E}_{\mathbf{x}}\left(\sigma(\mathbf{w}^\top \mathbf{x})\right) \\
&= \mathbb{E}_{\tilde{\mathbf{x}}} \mathbb{1}(\tilde{\mathbf{w}}^\top \tilde{\mathbf{x}} > -b_{\mathbf{w}})(\tilde{\mathbf{w}}^\top \tilde{\mathbf{x}} + b_{\mathbf{w}}) .
\end{aligned}
$$

If $-\frac{b_{\mathbf{w}}}{\|\tilde{\mathbf{w}}\|} \geq r$ then for every $\tilde{\mathbf{x}} \in \mathcal{B}$ we have $\tilde{\mathbf{w}}^\top \tilde{\mathbf{x}} \leq \|\tilde{\mathbf{w}}\| r \leq -b_{\mathbf{w}}$ and hence $\frac{db_{\mathbf{w}}}{dt} = 0$. Otherwise, we have $\frac{db_{\mathbf{w}}}{dt} < 0$. $\qquad\square$

*Proof of Theorem 3.2.* By Lemma B.2 $\mathbf{w}_0$ satisfies $\Pr_{\mathbf{x} \sim \mathcal{D}}\left[\mathbf{w}_0^\top \mathbf{x} \geq 0, \mathbf{v}^\top \mathbf{x} \geq 0\right] = 0$ w.p. at least $\frac{1}{2} - o_d(1)$. Then, by Lemma B.3 we have for every $t > 0$ that $\tilde{\mathbf{w}}_t = \gamma_t \tilde{\mathbf{w}}_0$ for some $\gamma_t > 0$, $b_{\mathbf{w}_t} < 0$, and $-\frac{b_{\mathbf{w}_t}}{\|\tilde{\mathbf{w}}_t\|} \leq r$. Moreover, we have $\Pr_{\mathbf{x} \sim \mathcal{D}}\left[\mathbf{w}_t^\top \mathbf{x} \geq 0, \mathbf{v}^\top \mathbf{x} \geq 0\right] = 0$. Hence, for every $t$ we have

$$
\begin{aligned}
F(\mathbf{w}_t) &= \frac{1}{2} \cdot \mathbb{E}_{\mathbf{x}}\left(\sigma(\mathbf{w}_t^\top \mathbf{x}) - \sigma(\mathbf{v}^\top \mathbf{x})\right)^2 \\
&= \frac{1}{2} \cdot \mathbb{E}_{\mathbf{x}}\left(\sigma(\mathbf{w}_t^\top \mathbf{x})\right)^2 + \frac{1}{2} \cdot \mathbb{E}_{\mathbf{x}}\left(\sigma(\mathbf{v}^\top \mathbf{x})\right)^2 - \mathbb{E}_{\mathbf{x}}\left(\sigma(\mathbf{w}_t^\top \mathbf{x})\sigma(\mathbf{v}^\top \mathbf{x})\right) \\
&= \frac{1}{2} \cdot \mathbb{E}_{\mathbf{x}}\left(\sigma(\mathbf{w}_t^\top \mathbf{x})\right)^2 + \frac{1}{2} \cdot \mathbb{E}_{\mathbf{x}}\left(\sigma(\mathbf{v}^\top \mathbf{x})\right)^2 \\
&\geq \frac{1}{2} \cdot \mathbb{E}_{\mathbf{x}}\left(\sigma(\mathbf{v}^\top \mathbf{x})\right)^2 = F(\mathbf{0}) .
\end{aligned}
$$

Thus, gradient flow does not converge to the global minimum $F(\mathbf{v}) = 0 < F(\mathbf{0})$. $\qquad\square$

## C   Proofs from Section 4

*Proof of Theorem 4.2.* The gradient of the objective is:

$$
\nabla F(\mathbf{w}) = \mathbb{E}_{\mathbf{x} \sim \mathcal{D}}\left[\left(\sigma(\mathbf{w}^\top \mathbf{x}) - \sigma(\mathbf{v}^\top \mathbf{x})\right) \cdot \sigma'(\mathbf{w}^\top \mathbf{x})\mathbf{x}\right] .
$$

We can rewrite it using that $\sigma$ is the ReLU activation, and separating the bias terms:

$$
\nabla F(\mathbf{w}) = \mathbb{E}_{\tilde{\mathbf{x}} \sim \tilde{\mathcal{D}}}\left[\left(\sigma(\tilde{\mathbf{w}}^\top \tilde{\mathbf{x}} + b_{\mathbf{w}}) - \sigma(\tilde{\mathbf{v}}^\top \tilde{\mathbf{x}} + b_{\mathbf{v}})\right) \cdot \mathbb{1}(\tilde{\mathbf{w}}^\top \tilde{\mathbf{x}} + b_{\mathbf{w}} > 0)\mathbf{x}\right] .
$$

First, notice that if $\tilde{\mathbf{w}} = 0$ and $b_{\mathbf{w}} < 0$ then $\mathbb{1}(\tilde{\mathbf{w}}^\top \tilde{\mathbf{x}} + b_{\mathbf{w}} > 0) = 0$ for all $\tilde{\mathbf{x}}$, hence $\nabla F(\mathbf{w}) = 0$. Second, using Cauchy-Schwartz we have that $|\langle \tilde{\mathbf{w}}, \tilde{\mathbf{x}}\rangle| \leq c \cdot \|\tilde{\mathbf{w}}\|$. Hence, for $\mathbf{w}$ with $\tilde{\mathbf{w}} \neq 0$ and $-\frac{b_{\mathbf{w}}}{\|\tilde{\mathbf{w}}\|} \geq c$ we have that $\mathbb{1}(\tilde{\mathbf{w}}^\top \tilde{\mathbf{x}} + b_{\mathbf{w}} > 0) = 0$ for all $\tilde{\mathbf{x}}$ in the support of the distribution, hence $\nabla F(\mathbf{w}) = 0$. Lastly, it is clear that for $\mathbf{w} = \mathbf{v}$ we have that $\nabla F(\mathbf{w}) = 0$. This shows that the points described in the statement of the proposition are indeed critical points. Next we will show that these are the only critical points.

Let $\mathbf{w} \in \mathbb{R}^{d+1}$ which is not a critical point defined above - i.e. either $\tilde{\mathbf{w}} = \mathbf{0}$ and $b_{\mathbf{w}} > 0$, or $\tilde{\mathbf{w}} \neq \mathbf{0}$ and $-\frac{b_{\mathbf{w}}}{\|\tilde{\mathbf{w}}\|} < c$. Then we have:

$$
\begin{aligned}
\langle \nabla F(\mathbf{w}), \mathbf{w} - \mathbf{v} \rangle &= \mathbb{E}_{\mathbf{x} \sim \mathcal{D}} \left[ \sigma'(\mathbf{w}^\top \mathbf{x})(\sigma(\mathbf{w}^\top \mathbf{x}) - \sigma(\mathbf{v}^\top \mathbf{x}))(\mathbf{w}^\top \mathbf{x} - \mathbf{v}^\top \mathbf{x}) \right] \\
&= \mathbb{E}_{\mathbf{x} \sim \mathcal{D}} \left[ \mathbb{1}(\mathbf{w}^\top \mathbf{x} > 0, \mathbf{v}^\top \mathbf{x} > 0)(\sigma(\mathbf{w}^\top \mathbf{x}) - \sigma(\mathbf{v}^\top \mathbf{x}))(\mathbf{w}^\top \mathbf{x} - \mathbf{v}^\top \mathbf{x}) \right] + \\
&\quad + \mathbb{E}_{\mathbf{x} \sim \mathcal{D}} \left[ \mathbb{1}(\mathbf{w}^\top \mathbf{x} > 0, \mathbf{v}^\top \mathbf{x} \leq 0) \sigma(\mathbf{w}^\top \mathbf{x})(\mathbf{w}^\top \mathbf{x} - \mathbf{v}^\top \mathbf{x}) \right] \\
&\geq \mathbb{E}_{\mathbf{x} \sim \mathcal{D}} \left[ \mathbb{1}(\mathbf{w}^\top \mathbf{x} > 0, \mathbf{v}^\top \mathbf{x} > 0)(\mathbf{w}^\top \mathbf{x} - \mathbf{v}^\top \mathbf{x})^2 \right] + \\
&\quad + \mathbb{E}_{\mathbf{x} \sim \mathcal{D}} \left[ \mathbb{1}(\mathbf{w}^\top \mathbf{x} > 0, \mathbf{v}^\top \mathbf{x} \leq 0)(\mathbf{w}^\top \mathbf{x})^2 \right] . \\
&= \mathbb{E}_{\mathbf{x} \sim \mathcal{D}} \left[ \mathbb{1}(\tilde{\mathbf{w}}^\top \tilde{\mathbf{x}} > -b_{\mathbf{w}}, \tilde{\mathbf{v}}^\top \tilde{\mathbf{x}} > -b_{\mathbf{v}})(\mathbf{w}^\top \mathbf{x} - \mathbf{v}^\top \mathbf{x})^2 \right] + \\
&\quad + \mathbb{E}_{\mathbf{x} \sim \mathcal{D}} \left[ \mathbb{1}(\tilde{\mathbf{w}}^\top \tilde{\mathbf{x}} > -b_{\mathbf{w}}, \tilde{\mathbf{v}}^\top \tilde{\mathbf{x}} \leq -b_{\mathbf{v}})(\mathbf{w}^\top \mathbf{x})^2 \right] . \quad (9)
\end{aligned}
$$

Denote:

$$
A_1 := \{\tilde{\mathbf{x}} \in \mathbb{R}^d : \tilde{\mathbf{w}}^\top \tilde{\mathbf{x}} > -b_{\mathbf{w}}, \tilde{\mathbf{v}}^\top \tilde{\mathbf{x}} > -b_{\mathbf{v}}, \|\tilde{\mathbf{x}}\| < c\}
$$
$$
A_2 := \{\tilde{\mathbf{x}} \in \mathbb{R}^d : \tilde{\mathbf{w}}^\top \tilde{\mathbf{x}} > -b_{\mathbf{w}}, \tilde{\mathbf{v}}^\top \tilde{\mathbf{x}} \leq -b_{\mathbf{v}}, \|\tilde{\mathbf{x}}\| < c\}
$$

Since $\mathbf{w}$ is not a critical point as defined above, we know that the set $\{\tilde{\mathbf{x}} \in \mathbb{R}^d : \tilde{\mathbf{w}}^\top \tilde{\mathbf{x}} > -b_{\mathbf{w}}, \|\tilde{\mathbf{x}}\| < c\}$ has a positive measure, hence either $A_1$ or $A_2$ have a positive measure. Assume w.l.o.g that $A_1$ have a positive measure, the other case is similar. Since both terms inside the expectations of Eq. (9) are positive, we can lower bound it with:

$$
\begin{aligned}
&\mathbb{E}_{\mathbf{x} \sim \mathcal{D}} \left[ \mathbb{1}(\tilde{\mathbf{w}}^\top \tilde{\mathbf{x}} > -b_{\mathbf{w}}, \tilde{\mathbf{v}}^\top \tilde{\mathbf{x}} > -b_{\mathbf{v}})(\mathbf{w}^\top \mathbf{x} - \mathbf{v}^\top \mathbf{x})^2 \right] \\
&= \|\mathbf{w} - \mathbf{v}\|^2 \mathbb{E}_{\mathbf{x} \sim \mathcal{D}} \left[ \mathbb{1}(\tilde{\mathbf{x}} \in A_1)((\overline{\mathbf{w} - \mathbf{v}})^\top \mathbf{x})^2 \right] \quad (10)
\end{aligned}
$$

Denote $\mathbf{u} := \overline{\mathbf{w} - \mathbf{v}}$, and note that $\mathbf{w} \neq \mathbf{v}$, hence $\|\mathbf{u}\| = 1$. Denote by $p(\tilde{\mathbf{x}})$ the pdf of $\tilde{\mathcal{D}}$, then we can rewrite Eq. (10) as:

$$
\begin{aligned}
&\|\mathbf{w} - \mathbf{v}\|^2 \cdot \int_{\tilde{\mathbf{x}} \in \mathbb{R}^d} \mathbb{1}(\tilde{\mathbf{x}} \in A_1) \cdot (\tilde{\mathbf{u}}^\top \tilde{\mathbf{x}} + b_{\mathbf{u}})^2 p(\tilde{\mathbf{x}}) d\tilde{\mathbf{x}} \\
&= \|\mathbf{w} - \mathbf{v}\|^2 \cdot \int_{\tilde{\mathbf{x}} \in A_1} (\tilde{\mathbf{u}}^\top \tilde{\mathbf{x}} + b_{\mathbf{u}})^2 p(\tilde{\mathbf{x}}) d\tilde{\mathbf{x}} \quad (11)
\end{aligned}
$$

Since the set $A_1$ has a positive measure, and the set $\{\tilde{\mathbf{x}} : \tilde{\mathbf{u}}^\top \tilde{\mathbf{x}} + b_{\mathbf{u}} = 0\}$ is of zero measure, there is a point $\tilde{\mathbf{x}}_0$ such that $\tilde{\mathbf{u}}^\top \tilde{\mathbf{x}} + b_{\mathbf{u}} \neq 0$. By continuity, there is a small enough neighborhood $A$ of $\tilde{\mathbf{x}}_0$, such that $\tilde{\mathbf{u}}^\top \tilde{\mathbf{x}} + b_{\mathbf{u}} \neq 0$ for every $\tilde{\mathbf{x}} \in A$. Using Assumption 4.1 we can lower bound Eq. (11) by:

$$
\|\mathbf{w} - \mathbf{v}\|^2 \cdot \beta \int_{\tilde{\mathbf{x}} \in A} (\tilde{\mathbf{u}}^\top \tilde{\mathbf{x}} + b_{\mathbf{u}})^2 d\tilde{\mathbf{x}}
$$

where this integral is positive. This shows that $\langle \nabla F(\mathbf{w}), \mathbf{w} - \mathbf{v} \rangle > 0$, which shows that $\nabla F(\mathbf{w}) \neq \mathbf{0}$, hence $\mathbf{w}$ is not a critical point.

$\square$

## D  Proofs from Section 5

The following lemmas are required in order to prove Theorem 5.2. First, we show that if $F(\mathbf{w}) \leq F(\mathbf{0}) - \delta$ then we can lower bound $\|\mathbf{w}\|$ and $\Pr_{\mathbf{x}} \left[ \mathbf{w}^\top \mathbf{x} \geq 0, \mathbf{v}^\top \mathbf{x} \geq 0 \right]$.

**Lemma D.1.** *Let $\delta > 0$ and let $\mathbf{w} \in \mathbb{R}^{d+1}$ such that $F(\mathbf{w}) \leq F(\mathbf{0}) - \delta$. Then*

$$
\|\mathbf{w}\| \geq \frac{\delta}{c^2} ,
$$

*and*

$$
\Pr_{\mathbf{x}} \left[ \mathbf{w}^\top \mathbf{x} \geq 0, \mathbf{v}^\top \mathbf{x} \geq 0 \right] \geq \frac{\delta}{c^2 \|\mathbf{w}\|} .
$$

*Proof.* We have

$$F(\mathbf{0}) - \delta \geq F(\mathbf{w}) = \frac{1}{2}\mathbb{E}_{\mathbf{x}}(\sigma(\mathbf{w}^\top\mathbf{x}) - \sigma(\mathbf{v}^\top\mathbf{x}))^2$$

$$= \frac{1}{2}\mathbb{E}_{\mathbf{x}}(\sigma(\mathbf{w}^\top\mathbf{x}))^2 + \frac{1}{2}\mathbb{E}_{\mathbf{x}}(\sigma(\mathbf{v}^\top\mathbf{x}))^2 - \mathbb{E}_{\mathbf{x}}(\sigma(\mathbf{w}^\top\mathbf{x})\sigma(\mathbf{v}^\top\mathbf{x}))$$

$$\geq F(\mathbf{0}) - \mathbb{E}_{\mathbf{x}}(\sigma(\mathbf{w}^\top\mathbf{x})\sigma(\mathbf{v}^\top\mathbf{x})) \ .$$

Hence

$$\delta \leq \mathbb{E}_{\mathbf{x}}\sigma(\mathbf{w}^\top\mathbf{x})\sigma(\mathbf{v}^\top\mathbf{x}) = \mathbb{E}_{\mathbf{x}}\mathbb{1}(\mathbf{w}^\top\mathbf{x} \geq 0, \mathbf{v}^\top\mathbf{x} \geq 0)\cdot\mathbf{w}^\top\mathbf{x}\cdot\mathbf{v}^\top\mathbf{x}$$

$$\leq \|\mathbf{w}\|c^2 \cdot \Pr_{\mathbf{x}}\left[\mathbf{w}^\top\mathbf{x} \geq 0, \mathbf{v}^\top\mathbf{x} \geq 0\right] \ .$$

Thus,

$$\|\mathbf{w}\| \geq \frac{\delta}{c^2 \cdot \Pr_{\mathbf{x}}\left[\mathbf{w}^\top\mathbf{x} \geq 0, \mathbf{v}^\top\mathbf{x} \geq 0\right]} \geq \frac{\delta}{c^2} \ ,$$

and

$$\Pr_{\mathbf{x}}\left[\mathbf{w}^\top\mathbf{x} \geq 0, \mathbf{v}^\top\mathbf{x} \geq 0\right] \geq \frac{\delta}{c^2\|\mathbf{w}\|} \ .$$

$$\square$$

Using the above lemma, we now show that if $F(\mathbf{w}) \leq F(\mathbf{0}) - \delta$ then $\|\mathbf{w} - \mathbf{v}\|$ decreases.

**Lemma D.2.** *Let $\delta > 0$ and let $B > 1$. Let $\mathbf{w} \in \mathbb{R}^{d+1}$ such that $F(\mathbf{w}) \leq F(\mathbf{0}) - \delta$ and $\|\mathbf{w} - \mathbf{v}\| \leq B - 1$. Let $\gamma = \frac{\delta^3}{3\cdot 12^2 B^3 c^8 c'^2}$ and let $0 < \eta \leq \frac{\gamma}{c^4}$. Let $\mathbf{w}' = \mathbf{w} - \eta\nabla F(\mathbf{w})$. Then,*

$$\|\mathbf{w}' - \mathbf{v}\|^2 \leq \|\mathbf{w} - \mathbf{v}\|^2 \cdot (1 - \gamma\eta) \leq (B-1)^2 \ .$$

*Proof.* We have

$$\|\mathbf{w}' - \mathbf{v}\|^2 = \|\mathbf{w} - \eta\nabla F(\mathbf{w}) - \mathbf{v}\|^2$$

$$= \|\mathbf{w} - \mathbf{v}\|^2 - 2\eta\langle\nabla F(\mathbf{w}), \mathbf{w} - \mathbf{v}\rangle + \eta^2\|\nabla F(\mathbf{w})\|^2 \ . \tag{12}$$

We first bound $\|\nabla F(\mathbf{w})\|^2$. By Jensen's inequality and since $\sigma$ is 1-Lipschitz, we have:

$$\|\nabla F(\mathbf{w})\|^2 \leq \mathbb{E}_{\mathbf{x}}\left[\left(\sigma(\mathbf{w}^\top\mathbf{x}) - \sigma(\mathbf{v}^\top\mathbf{x})\right)^2 \sigma'(\mathbf{w}^\top\mathbf{x})\|\mathbf{x}\|^2\right]$$

$$\leq c^2\mathbb{E}_{\mathbf{x}}\left[\left(\sigma(\mathbf{w}^\top\mathbf{x}) - \sigma(\mathbf{v}^\top\mathbf{x})\right)^2\right]$$

$$\leq c^2\mathbb{E}_{\mathbf{x}}\left[\left(\mathbf{w}^\top\mathbf{x} - \mathbf{v}^\top\mathbf{x}\right)^2\right]$$

$$= c^2\mathbb{E}_{\mathbf{x}}\left[\left((\mathbf{w} - \mathbf{v})^\top\mathbf{x}\right)^2\right]$$

$$\leq c^4\|\mathbf{w} - \mathbf{v}\|^2 \ . \tag{13}$$

Next, we bound $\langle\nabla F(\mathbf{w}), \mathbf{w} - \mathbf{v}\rangle$. Let $\mathbf{u} = \overline{\mathbf{w} - \mathbf{v}}$. We have

$$\langle\nabla F(\mathbf{w}), \mathbf{w} - \mathbf{v}\rangle = \mathbb{E}_{\mathbf{x}}\left(\sigma(\mathbf{w}^\top\mathbf{x}) - \sigma(\mathbf{v}^\top\mathbf{x})\right)\sigma'(\mathbf{w}^\top\mathbf{x})(\mathbf{w}^\top\mathbf{x} - \mathbf{v}^\top\mathbf{x})$$

$$= \mathbb{E}_{\mathbf{x}}\left(\mathbf{w}^\top\mathbf{x} - \mathbf{v}^\top\mathbf{x}\right)^2\mathbb{1}(\mathbf{w}^\top\mathbf{x} \geq 0, \mathbf{v}^\top\mathbf{x} \geq 0)+$$

$$\mathbb{E}_{\mathbf{x}}\mathbf{w}^\top\mathbf{x}\cdot(\mathbf{w}^\top\mathbf{x} - \mathbf{v}^\top\mathbf{x})\mathbb{1}(\mathbf{w}^\top\mathbf{x} \geq 0, \mathbf{v}^\top\mathbf{x} < 0)$$

$$\geq \|\mathbf{w} - \mathbf{v}\|^2 \cdot \mathbb{E}_{\mathbf{x}}\mathbb{1}(\mathbf{w}^\top\mathbf{x} \geq 0, \mathbf{v}^\top\mathbf{x} \geq 0)(\mathbf{u}^\top\mathbf{x})^2 \ .$$

Let $\xi = \frac{\delta}{12Bc^3c'}$. The above is at least

$$\|\mathbf{w} - \mathbf{v}\|^2 \cdot \xi^2 \cdot \Pr_{\mathbf{x}}\left[\mathbf{w}^\top\mathbf{x} \geq 0, \mathbf{v}^\top\mathbf{x} \geq 0, (\mathbf{u}^\top\mathbf{x})^2 \geq \xi^2\right]$$

$$= \|\mathbf{w} - \mathbf{v}\|^2 \cdot \xi^2 \cdot \left(\Pr_{\mathbf{x}}\left[\mathbf{w}^\top\mathbf{x} \geq 0, \mathbf{v}^\top\mathbf{x} \geq 0\right] - \Pr_{\mathbf{x}}\left[\mathbf{w}^\top\mathbf{x} \geq 0, \mathbf{v}^\top\mathbf{x} \geq 0, (\mathbf{u}^\top\mathbf{x})^2 < \xi^2\right]\right) \ .$$

By Lemma D.1, and since $\|\mathbf{w}\| \le \|\mathbf{w} - \mathbf{v}\| + \|\mathbf{v}\| \le B - 1 + 1 = B$, the above is at least

$$\|\mathbf{w} - \mathbf{v}\|^2 \cdot \xi^2 \cdot \left( \frac{\delta}{c^2 \|\mathbf{w}\|} - \Pr_{\mathbf{x}} \left[ \mathbf{w}^\top \mathbf{x} \ge 0, \mathbf{v}^\top \mathbf{x} \ge 0, |\mathbf{u}^\top \mathbf{x}| < \xi \right] \right)$$

$$\ge \|\mathbf{w} - \mathbf{v}\|^2 \cdot \xi^2 \cdot \left( \frac{\delta}{c^2 B} - \Pr_{\mathbf{x}} \left[ |\mathbf{u}^\top \mathbf{x}| \le \xi \right] \right)$$

$$= \|\mathbf{w} - \mathbf{v}\|^2 \cdot \xi^2 \cdot \left( \frac{\delta}{c^2 B} - \Pr_{\mathbf{x}} \left[ |\tilde{\mathbf{u}}^\top \tilde{\mathbf{x}} + b_{\mathbf{u}}| \le \xi \right] \right) . \tag{14}$$

We now bound $\Pr_{\mathbf{x}} \left[ |\tilde{\mathbf{u}}^\top \tilde{\mathbf{x}} + b_{\mathbf{u}}| \le \xi \right]$. We denote $a = \|\tilde{\mathbf{u}}\|$. If $a \le \frac{1}{4c}$, then since $\|\mathbf{u}\| = 1$ we have $b_{\mathbf{u}} \ge \sqrt{1 - \frac{1}{16c^2}} \ge \sqrt{1 - \frac{1}{16}} = \frac{\sqrt{15}}{4}$. Hence, for every $\mathbf{x}$ with $\|\mathbf{x}\| \le c$ we have

$$|\tilde{\mathbf{u}}^\top \tilde{\mathbf{x}} + b_{\mathbf{u}}| \ge |b_{\mathbf{u}}| - |\tilde{\mathbf{u}}^\top \tilde{\mathbf{x}}| \ge \frac{\sqrt{15}}{4} - ac \ge \frac{\sqrt{15}}{4} - \frac{1}{4} > \frac{1}{2} .$$

Note that

$$\xi = \frac{\delta}{12 B c^3 c'} \le \frac{F(\mathbf{0})}{12 B c^3 c'} = \frac{1}{12 B c^3 c'} \cdot \frac{1}{2} \mathbb{E}_{\mathbf{x}} (\sigma(\mathbf{v}^\top \mathbf{x}))^2 \le \frac{1}{12 B c^3 c'} \cdot \frac{1}{2} c^2 = \frac{1}{24 B c c'} \le \frac{1}{24} ,$$

where the last inequality is since $B, c, c' \ge 1$. Therefore, $|\tilde{\mathbf{u}}^\top \tilde{\mathbf{x}} + b_{\mathbf{u}}| > \xi$. Thus,

$$\Pr_{\mathbf{x}} \left[ |\tilde{\mathbf{u}}^\top \tilde{\mathbf{x}} + b_{\mathbf{u}}| \le \xi \right] = 0 .$$

Assume now that $a \ge \frac{1}{4c}$. We have

$$\Pr_{\mathbf{x}} \left[ |\tilde{\mathbf{u}}^\top \tilde{\mathbf{x}} + b_{\mathbf{u}}| \le \xi \right] = \Pr_{\mathbf{x}} \left[ \tilde{\mathbf{u}}^\top \tilde{\mathbf{x}} \in [-\xi - b_{\mathbf{u}}, \xi - b_{\mathbf{u}}] \right]$$

$$= \Pr_{\mathbf{x}} \left[ \bar{\tilde{\mathbf{u}}}^\top \tilde{\mathbf{x}} \in [-\frac{\xi}{a} - \frac{b_{\mathbf{u}}}{a}, \frac{\xi}{a} - \frac{b_{\mathbf{u}}}{a}] \right]$$

$$\le c' \cdot 2 \cdot \frac{\xi}{a}$$

$$\le 8 c c' \xi .$$

Combining the above with Eq. (14), we obtain

$$\langle \nabla F(\mathbf{w}), \mathbf{w} - \mathbf{v} \rangle \ge \|\mathbf{w} - \mathbf{v}\|^2 \cdot \xi^2 \cdot \left( \frac{\delta}{c^2 B} - 8 c c' \xi \right)$$

$$= \|\mathbf{w} - \mathbf{v}\|^2 \frac{\delta^2}{12^2 B^2 c^6 c'^2} \cdot \left( \frac{\delta}{c^2 B} - 8 c c' \cdot \frac{\delta}{12 B c^3 c'} \right)$$

$$= \|\mathbf{w} - \mathbf{v}\|^2 \frac{\delta^2}{12^2 B^2 c^6 c'^2} \cdot \left( \frac{\delta}{3 c^2 B} \right)$$

$$= \|\mathbf{w} - \mathbf{v}\|^2 \frac{\delta^3}{3 \cdot 12^2 B^3 c^8 c'^2} . \tag{15}$$

Combining Eq. (12), (13) and (15), and using $\gamma = \frac{\delta^3}{3 \cdot 12^2 B^3 c^8 c'^2}$, we have

$$\|\mathbf{w}' - \mathbf{v}\|^2 \le \|\mathbf{w} - \mathbf{v}\|^2 - 2\eta \|\mathbf{w} - \mathbf{v}\|^2 \cdot \gamma + \eta^2 c^4 \|\mathbf{w} - \mathbf{v}\|^2$$

$$= \|\mathbf{w} - \mathbf{v}\|^2 \cdot \left( 1 - 2\eta\gamma + \eta^2 c^4 \right) .$$

Since $\eta \le \frac{\gamma}{c^4}$, we obtain

$$\|\mathbf{w}' - \mathbf{v}\|^2 \le \|\mathbf{w} - \mathbf{v}\|^2 \cdot \left( 1 - 2\eta\gamma + \eta c^4 \cdot \frac{\gamma}{c^4} \right)$$

$$= \|\mathbf{w} - \mathbf{v}\|^2 \cdot (1 - \gamma\eta) \le \|\mathbf{w} - \mathbf{v}\|^2 \le (B - 1)^2 .$$

$\square$

Next, we show that $F(\mathbf{w})$ remains smaller than $F(\mathbf{0}) - \delta$ during the training. In the following two lemmas we obtain a bound for the smoothness of $F$ in the relevant region, and in the two lemmas that follow we use this bound to show that $F(\mathbf{w})$ indeed remains small.

**Lemma D.3.** *Let $\mathbf{w} \in \mathbb{R}^{d+1}$ such that $F(\mathbf{w}) \leq F(\mathbf{0})$. Then, $\|\nabla F(\mathbf{w})\| \leq c\sqrt{2F(\mathbf{0})}$.*

*Proof.* By Jensen's inequality, we have

$$\|\nabla F(\mathbf{w})\|^2 \leq \mathop{\mathbb{E}}_{\mathbf{x}} \left(\sigma(\mathbf{w}^\top \mathbf{x}) - \sigma(\mathbf{v}^\top \mathbf{x})\right)^2 \sigma'(\mathbf{w}^\top \mathbf{x})\|\mathbf{x}\|^2$$

$$\leq c^2 \mathop{\mathbb{E}}_{\mathbf{x}} \left(\sigma(\mathbf{w}^\top \mathbf{x}) - \sigma(\mathbf{v}^\top \mathbf{x})\right)^2$$

$$\leq c^2 2F(\mathbf{w}) \leq 2c^2 F(\mathbf{0}) .$$

$\square$

**Lemma D.4.** *Let $M, B > 0$ and let $\mathbf{w}, \mathbf{w}' \in \mathbb{R}^{d+1}$ be such that for every $s \in [0,1]$ we have $M \leq \|\mathbf{w} + s(\mathbf{w}' - \mathbf{w})\| \leq B$. Then,*

$$\|\nabla F(\mathbf{w}) - \nabla F(\mathbf{w}')\| \leq \|\mathbf{w} - \mathbf{w}'\| \cdot c^2 \left(1 + \frac{8(B+1)c'c^2}{M}\right) .$$

*Proof.* We assume w.l.o.g. that $\|\mathbf{w} - \mathbf{w}'\| \leq \frac{M}{2c}$. Indeed, let $0 = s_0 < \ldots < s_k = 1$ for some integer $k$, let $\mathbf{w}_i = \mathbf{w} + s_i(\mathbf{w}' - \mathbf{w})$, and assume that $\|\mathbf{w}_i - \mathbf{w}_{i+1}\| \leq \frac{M}{2c}$ for every $i$. If the claim holds for every pair $\mathbf{w}_i, \mathbf{w}_{i+1}$, then we have

$$\|\nabla F(\mathbf{w}) - \nabla F(\mathbf{w}')\| = \|\sum_{i=0}^{k-1} \nabla F(\mathbf{w}_i) - \nabla F(\mathbf{w}_{i+1})\|$$

$$\leq \sum_{i=0}^{k-1} \|\nabla F(\mathbf{w}_i) - \nabla F(\mathbf{w}_{i+1})\|$$

$$\leq \sum_{i=0}^{k-1} \|\mathbf{w}_i - \mathbf{w}_{i+1}\| \cdot c^2 \left(1 + \frac{8(B+1)c'c^2}{M}\right)$$

$$= c^2 \left(1 + \frac{8(B+1)c'c^2}{M}\right) \|\mathbf{w} - \mathbf{w}'\| .$$

We have

$$\|\nabla F(\mathbf{w}) - \nabla F(\mathbf{w}')\|$$

$$= \|\mathop{\mathbb{E}}_{\mathbf{x}} (\sigma(\mathbf{w}^\top \mathbf{x}) - \sigma(\mathbf{v}^\top \mathbf{x}))\sigma'(\mathbf{w}^\top \mathbf{x})\mathbf{x} - (\sigma(\mathbf{w}'^\top \mathbf{x}) - \sigma(\mathbf{v}^\top \mathbf{x}))\sigma'(\mathbf{w}'^\top \mathbf{x})\mathbf{x}\|$$

$$\leq \|\mathop{\mathbb{E}}_{\mathbf{x}} \mathbb{1}(\mathbf{w}^\top \mathbf{x} \geq 0, \mathbf{w}'^\top \mathbf{x} \geq 0) \left(\mathbf{w}^\top \mathbf{x} - \sigma(\mathbf{v}^\top \mathbf{x}) - \mathbf{w}'^\top \mathbf{x} + \sigma(\mathbf{v}^\top \mathbf{x})\right) \mathbf{x}\| +$$

$$\|\mathop{\mathbb{E}}_{\mathbf{x}} \mathbb{1}(\mathbf{w}^\top \mathbf{x} \geq 0, \mathbf{w}'^\top \mathbf{x} < 0) \left(\mathbf{w}^\top \mathbf{x} - \sigma(\mathbf{v}^\top \mathbf{x})\right) \mathbf{x}\| +$$

$$\|\mathop{\mathbb{E}}_{\mathbf{x}} \mathbb{1}(\mathbf{w}^\top \mathbf{x} < 0, \mathbf{w}'^\top \mathbf{x} \geq 0) \left(\mathbf{w}'^\top \mathbf{x} - \sigma(\mathbf{v}^\top \mathbf{x})\right) \mathbf{x}\| .$$

By Jensen's inequality and Cauchy-Shwartz, the above is at most

$$\mathop{\mathbb{E}}_{\mathbf{x}} \mathbb{1}(\mathbf{w}^\top \mathbf{x} \geq 0, \mathbf{w}'^\top \mathbf{x} \geq 0)\|\mathbf{w} - \mathbf{w}'\| \cdot \|\mathbf{x}\| \cdot \|\mathbf{x}\| +$$

$$\mathop{\mathbb{E}}_{\mathbf{x}} \mathbb{1}(\mathbf{w}^\top \mathbf{x} \geq 0, \mathbf{w}'^\top \mathbf{x} < 0) (\|\mathbf{w}\| \cdot \|\mathbf{x}\| + \|\mathbf{v}\| \cdot \|\mathbf{x}\|) \cdot \|\mathbf{x}\| +$$

$$\mathop{\mathbb{E}}_{\mathbf{x}} \mathbb{1}(\mathbf{w}^\top \mathbf{x} < 0, \mathbf{w}'^\top \mathbf{x} \geq 0) (\|\mathbf{w}'\| \cdot \|\mathbf{x}\| + \|\mathbf{v}\| \cdot \|\mathbf{x}\|) \cdot \|\mathbf{x}\| .$$

By our assumption we have $\|\mathbf{x}\| \leq c$ and $\|\mathbf{w}\|, \|\mathbf{w}'\| \leq B$. Hence, the above is at most

$$\|\mathbf{w} - \mathbf{w}'\|c^2 + \mathop{\Pr}_{\mathbf{x}} \left[\mathbf{w}^\top \mathbf{x} \geq 0, \mathbf{w}'^\top \mathbf{x} < 0\right] \cdot c^2 \cdot (B+1)$$

$$+ \mathop{\Pr}_{\mathbf{x}} \left[\mathbf{w}^\top \mathbf{x} < 0, \mathbf{w}'^\top \mathbf{x} \geq 0\right] \cdot c^2 \cdot (B+1) . \tag{16}$$

Now, we bound $\Pr_{\mathbf{x}}\left[\mathbf{w}^\top\mathbf{x} \geq 0, \mathbf{w}'^\top\mathbf{x} < 0\right]$. If $\mathbf{w}^\top\mathbf{x} \geq 0$ and $\mathbf{w}'^\top\mathbf{x} < 0$ then

$$\mathbf{w}^\top\mathbf{x} = \mathbf{w}'^\top\mathbf{x} + (\mathbf{w} - \mathbf{w}')^\top\mathbf{x} < 0 + \|\mathbf{w} - \mathbf{w}'\| \cdot \|\mathbf{x}\| \leq c \cdot \|\mathbf{w} - \mathbf{w}'\| .$$

Hence, we only need to bound

$$\Pr_{\mathbf{x}}\left[\mathbf{w}^\top\mathbf{x} \in [0, c \cdot \|\mathbf{w} - \mathbf{w}'\|]\right] = \Pr_{\mathbf{x}}\left[\tilde{\mathbf{w}}^\top\tilde{\mathbf{x}} + b_{\mathbf{w}} \in [0, c \cdot \|\mathbf{w} - \mathbf{w}'\|]\right] .$$

We denote $a = \|\tilde{\mathbf{w}}\|$. If $a \leq \frac{M}{4c}$, then since $\|\mathbf{w}\| \geq M$ we have $|b_{\mathbf{w}}| \geq \sqrt{M^2 - \left(\frac{M}{4c}\right)^2} = M\sqrt{1 - 1/(16c^2)}$. Hence for every $\mathbf{x}$ we have

$$|\tilde{\mathbf{w}}^\top\tilde{\mathbf{x}} + b_{\mathbf{w}}| \geq |b_{\mathbf{w}}| - |\tilde{\mathbf{w}}^\top\tilde{\mathbf{x}}| \geq |b_{\mathbf{w}}| - ac \geq M\sqrt{1 - 1/(16c^2)} - \frac{M}{4} \geq M\sqrt{1 - 1/16} - \frac{M}{4}$$

$$= M \cdot \frac{\sqrt{15} - 1}{4} > M/2 \geq c\|\mathbf{w} - \mathbf{w}'\| .$$

Thus, $\Pr_{\mathbf{x}}\left[\tilde{\mathbf{w}}^\top\tilde{\mathbf{x}} + b_{\mathbf{w}} \in [0, c \cdot \|\mathbf{w} - \mathbf{w}'\|]\right] = 0$.

Assume now that $a \geq \frac{M}{4c}$. Hence, $\frac{c}{a} \leq \frac{4c^2}{M}$. Therefore, we have

$$\Pr_{\mathbf{x}}\left[\tilde{\mathbf{w}}^\top\tilde{\mathbf{x}} + b_{\mathbf{w}} \in [0, c \cdot \|\mathbf{w} - \mathbf{w}'\|]\right] = \Pr_{\mathbf{x}}\left[\bar{\tilde{\mathbf{w}}}^\top\tilde{\mathbf{x}} \in [-\frac{b_{\mathbf{w}}}{a}, -\frac{b_{\mathbf{w}}}{a} + \frac{c}{a} \cdot \|\mathbf{w} - \mathbf{w}'\|]\right]$$

$$\leq \Pr_{\mathbf{x}}\left[\bar{\tilde{\mathbf{w}}}^\top\tilde{\mathbf{x}} \in [-\frac{b_{\mathbf{w}}}{a}, -\frac{b_{\mathbf{w}}}{a} + \frac{4c^2}{M} \cdot \|\mathbf{w} - \mathbf{w}'\|]\right]$$

$$\leq c' \cdot \frac{4c^2}{M} \cdot \|\mathbf{w} - \mathbf{w}'\| .$$

Hence, $\Pr_{\mathbf{x}}\left[\mathbf{w}^\top\mathbf{x} \geq 0, \mathbf{w}'^\top\mathbf{x} < 0\right] \leq c' \cdot \frac{4c^2}{M} \cdot \|\mathbf{w} - \mathbf{w}'\|$. By similar arguments, this inequality holds also for $\Pr_{\mathbf{x}}\left[\mathbf{w}^\top\mathbf{x} < 0, \mathbf{w}'^\top\mathbf{x} \geq 0\right]$. Plugging it into Eq. (16), we have

$$\|\nabla F(\mathbf{w}) - \nabla F(\mathbf{w}')\| \leq \|\mathbf{w} - \mathbf{w}'\|\left(c^2 + 2 \cdot c^2 \cdot (B+1) \cdot c' \cdot \frac{4c^2}{M}\right)$$

$$= \|\mathbf{w} - \mathbf{w}'\| \cdot c^2\left(1 + \frac{8(B+1)c'c^2}{M}\right) .$$

$\square$

**Lemma D.5.** *Let $f : \mathbb{R}^d \to \mathbb{R}$ and let $L > 0$. Let $\mathbf{x}, \mathbf{y} \in \mathbb{R}^d$ be such that for every $s \in [0, 1]$ we have $\|\nabla f(\mathbf{x} + s(\mathbf{y} - \mathbf{x})) - \nabla f(\mathbf{x})\| \leq Ls\|\mathbf{y} - \mathbf{x}\|$. Then,*

$$f(\mathbf{y}) - f(\mathbf{x}) \leq \nabla f(\mathbf{x})^\top(\mathbf{y} - \mathbf{x}) + \frac{L}{2}\|\mathbf{y} - \mathbf{x}\|^2 .$$

*Proof.* The proof follows a standard technique (cf. [2]). We represent $f(\mathbf{y}) - f(\mathbf{x})$ as an integral, apply Cauchy-Schwarz and then use the $L$-smoothness.

$$f(\mathbf{y}) - f(\mathbf{x}) - \nabla f(\mathbf{x})^\top(\mathbf{y} - \mathbf{x}) = \int_0^1 \nabla f(\mathbf{x} + s(\mathbf{y} - \mathbf{x}))^\top(\mathbf{y} - \mathbf{x})ds - \nabla f(\mathbf{x})^\top(\mathbf{y} - \mathbf{x})$$

$$\leq \int_0^1 \|\nabla f(\mathbf{x} + s(\mathbf{y} - \mathbf{x})) - \nabla f(\mathbf{x})\| \cdot \|\mathbf{y} - \mathbf{x}\|ds$$

$$\leq \int_0^1 Ls\|\mathbf{y} - \mathbf{x}\|^2 ds$$

$$= \frac{L}{2}\|\mathbf{y} - \mathbf{x}\|^2 .$$

Hence, we have

$$f(\mathbf{y}) - f(\mathbf{x}) \leq \nabla f(\mathbf{x})^\top(\mathbf{y} - \mathbf{x}) + \frac{L}{2}\|\mathbf{y} - \mathbf{x}\|^2 .$$

$\square$

**Lemma D.6.** *Let $B, \delta > 0$ and let $L = c^2 \left(1 + \frac{16(B+1)c'c^4}{\delta}\right)$. Let $\mathbf{w} \in \mathbb{R}^{d+1}$ such that $F(\mathbf{w}) \leq F(\mathbf{0}) - \delta$ and let $\mathbf{w}' = \mathbf{w} - \eta \cdot \nabla F(\mathbf{w})$, where $\eta \leq \min\left\{\frac{\delta}{2c^3\sqrt{2F(\mathbf{0})}}, \frac{1}{L}\right\} = \min\left\{\frac{\delta}{2c^3\sqrt{2F(\mathbf{0})}}, \frac{\delta}{\delta c^2 + 16(B+1)c'c^6}\right\}$. Assume that $\|\mathbf{w}\|, \|\mathbf{w}'\| \leq B$. Then, we have $F(\mathbf{w}') - F(\mathbf{w}) \leq -\eta\left(1 - \frac{L}{2}\eta\right)\|\nabla F(\mathbf{w})\|^2$, and $F(\mathbf{w}') \leq F(\mathbf{w}) \leq F(\mathbf{0}) - \delta$.*

*Proof.* Let $M = \frac{\delta}{2c^2}$. By Lemmas D.1 and D.3, we have $\|\mathbf{w}\| \geq \frac{\delta}{c^2}$ and $\|\nabla F(\mathbf{w})\| \leq c\sqrt{2F(\mathbf{0})}$. Hence for every $\lambda \in [0, 1]$ we have

$$\|\mathbf{w} - \lambda\eta\nabla F(\mathbf{w})\| \geq \frac{\delta}{c^2} - \eta \cdot c\sqrt{2F(\mathbf{0})} \geq \frac{\delta}{c^2} - \frac{\delta}{2c^3\sqrt{2F(\mathbf{0})}} \cdot c\sqrt{2F(\mathbf{0})} = \frac{\delta}{2c^2} = M .$$

Since $\|\mathbf{w}\|, \|\mathbf{w}'\| \leq B$, we also have $\|\mathbf{w} - \lambda\eta\nabla F(\mathbf{w})\| \leq B$. By Lemma D.4, we have for every $\lambda \in [0, 1]$ that

$$\|\nabla F(\mathbf{w}) - \nabla F(\mathbf{w} - \lambda\eta\nabla F(\mathbf{w}))\| \leq \lambda\eta\|\nabla F(\mathbf{w})\| \cdot c^2\left(1 + \frac{8(B+1)c'c^2}{M}\right) .$$

We have $L = c^2\left(1 + \frac{16(B+1)c'c^4}{\delta}\right) = c^2\left(1 + \frac{8(B+1)c'c^2}{M}\right)$. By Lemma D.5 we have

$$F(\mathbf{w} - \eta\nabla F(\mathbf{w})) - F(\mathbf{w}) \leq -\eta\|\nabla F(\mathbf{w})\|^2 + \frac{L}{2}\eta^2\|\nabla F(\mathbf{w})\|^2 .$$

Since $\eta \leq \frac{1}{L}$, we also have $F(\mathbf{w} - \eta\nabla F(\mathbf{w})) \leq F(\mathbf{w}) \leq F(\mathbf{0}) - \delta$. $\square$

We are now ready to prove the theorem:

*Proof of Theorem 5.2.* Let $B = \|\mathbf{w}_0\| + 2$. Assume that $\eta \leq \min\left\{\frac{\delta}{2c^3\sqrt{2F(\mathbf{0})}}, \frac{\delta}{\delta c^2 + 16(B+1)c'c^6}, \frac{\gamma}{c^4}\right\}$. We have $\|\mathbf{w}_0 - \mathbf{v}\| \leq \|\mathbf{w}_0\| + \|\mathbf{v}\| = \|\mathbf{w}_0\| + 1 \leq B - 1$. By Lemmas D.2 and D.6, for every $t$ we have $\|\mathbf{w}_t - \mathbf{v}\| \leq B - 1$ (thus, $\|\mathbf{w}_t\| \leq B$) and $F(\mathbf{w}_t) \leq F(\mathbf{0}) - \delta$. Moreover, by Lemma D.2, we have for every $t$ that $\|\mathbf{w}_{t+1} - \mathbf{v}\|^2 \leq \|\mathbf{w}_t - \mathbf{v}\|^2 \cdot (1 - \gamma\eta)$. Therefore, $\|\mathbf{w}_t - \mathbf{v}\|^2 \leq \|\mathbf{w}_0 - \mathbf{v}\|^2 (1 - \gamma\eta)^t$.

It remains to show that

$$\min\left\{\frac{\delta}{2c^3\sqrt{2F(\mathbf{0})}}, \frac{\delta}{\delta c^2 + 16(B+1)c'c^6}, \frac{\gamma}{c^4}\right\} = \frac{\gamma}{c^4} .$$

Note that we have $\delta \leq F(\mathbf{0}) = \frac{1}{2}\mathbb{E}_\mathbf{x}(\sigma(\mathbf{v}^\top\mathbf{x}))^2 \leq \frac{1}{2} \cdot c^2$. Thus

$$\frac{\gamma}{c^4} = \frac{\delta^3}{3 \cdot 12^2 B^3 c^{12} c'^2} \leq \frac{\delta}{3 \cdot 12^2 B^3 c^{12} c'^2} \cdot \frac{c^4}{4} = \frac{\delta}{12^3 B^3 c^8 c'^2} .$$

We have

$$\frac{\delta}{2c^3\sqrt{2F(\mathbf{0})}} \geq \frac{\delta}{2c^4} \geq \frac{\gamma}{c^4} ,$$

where the last inequality is since $B, c, c' \geq 1$. Finally,

$$\frac{\delta}{\delta c^2 + 16(B+1)c'c^6} \geq \frac{\delta}{\frac{c^4}{2} + 16(B+1)c'c^6} \geq \frac{\delta}{17(B+1)c'c^6} \geq \frac{\delta}{34Bc'c^6} \geq \frac{\gamma}{c^4} .$$

$\square$

### D.1 Proofs from Subsection 5.2

**Proof of Theorem 5.4**

We have

$$
\begin{aligned}
F(\mathbf{w}) &= \frac{1}{2} \mathop{\mathbb{E}}_{\mathbf{x}} \left( \sigma(\mathbf{w}^\top \mathbf{x}) - \sigma(\mathbf{v}^\top \mathbf{x}) \right)^2 \\
&= F(\mathbf{0}) + \frac{1}{2} \mathop{\mathbb{E}}_{\mathbf{x}} \left( \sigma(\mathbf{w}^\top \mathbf{x}) \right)^2 - \mathop{\mathbb{E}}_{\mathbf{x}} \sigma(\mathbf{w}^\top \mathbf{x}) \sigma(\mathbf{v}^\top \mathbf{x}) \\
&\leq F(\mathbf{0}) + \frac{\|\mathbf{w}\|^2 c^2}{2} - \|\mathbf{w}\| \mathop{\mathbb{E}}_{\mathbf{x}} \sigma(\bar{\mathbf{w}}^\top \mathbf{x}) \sigma(\mathbf{v}^\top \mathbf{x}) \, .
\end{aligned}
\tag{17}
$$

Let $\xi = \frac{\alpha}{4\sqrt{c}} \sin\left(\frac{\pi}{8}\right)$. We have

$$
\begin{aligned}
\mathop{\mathbb{E}}_{\mathbf{x}} \sigma(\bar{\mathbf{w}}^\top \mathbf{x}) \sigma(\mathbf{v}^\top \mathbf{x}) &\geq \xi^2 \cdot \Pr_{\mathbf{x}} \left[ \sigma(\bar{\mathbf{w}}^\top \mathbf{x}) \sigma(\mathbf{v}^\top \mathbf{x}) \geq \xi^2 \right] \\
&\geq \xi^2 \cdot \Pr_{\mathbf{x}} \left[ \bar{\mathbf{w}}^\top \mathbf{x} \geq 2\sqrt{c}\xi, \mathbf{v}^\top \mathbf{x} \geq \frac{\xi}{2\sqrt{c}} \right] \, .
\end{aligned}
\tag{18}
$$

In the following two lemmas we bound $\Pr_{\mathbf{x}} \left[ \bar{\mathbf{w}}^\top \mathbf{x} \geq 2\sqrt{c}\xi, \mathbf{v}^\top \mathbf{x} \geq \frac{\xi}{2\sqrt{c}} \right]$.

**Lemma D.7.** *If $b_{\mathbf{v}} \geq 0$ then*

$$
\Pr_{\mathbf{x}} \left[ \bar{\mathbf{w}}^\top \mathbf{x} \geq 2\sqrt{c}\xi, \mathbf{v}^\top \mathbf{x} \geq \frac{\xi}{2\sqrt{c}} \right] \geq \frac{\beta \left( \alpha \sin\left(\frac{\pi}{8}\right) - 2\sqrt{c}\xi \right)^2}{4 \sin\left(\frac{\pi}{8}\right)} \, .
$$

*Proof.* If $\|\tilde{\mathbf{v}}\| \geq \frac{1}{4c}$, then we have

$$
\begin{aligned}
\Pr_{\mathbf{x}} \left[ \bar{\mathbf{w}}^\top \mathbf{x} \geq 2\sqrt{c}\xi, \mathbf{v}^\top \mathbf{x} \geq \frac{\xi}{2\sqrt{c}} \right] &\geq \Pr_{\mathbf{x}} \left[ \bar{\mathbf{w}}^\top \mathbf{x} \geq 2\sqrt{c}\xi, \tilde{\mathbf{v}}^\top \tilde{\mathbf{x}} \geq \frac{\xi}{2\sqrt{c}} \right] \\
&= \Pr_{\mathbf{x}} \left[ \tilde{\bar{\mathbf{w}}}^\top \tilde{\mathbf{x}} \geq 2\sqrt{c}\xi, \bar{\tilde{\mathbf{v}}}^\top \tilde{\mathbf{x}} \geq \frac{\xi}{2\sqrt{c}\|\tilde{\mathbf{v}}\|} \right] \\
&\geq \Pr_{\mathbf{x}} \left[ \tilde{\bar{\mathbf{w}}}^\top \tilde{\mathbf{x}} \geq 2\sqrt{c}\xi, \bar{\tilde{\mathbf{v}}}^\top \tilde{\mathbf{x}} \geq 2\sqrt{c}\xi \right] \\
&\geq \frac{\beta \left( \alpha \sin\left(\frac{\pi}{8}\right) - 2\sqrt{c}\xi \right)^2}{4 \sin\left(\frac{\pi}{8}\right)} \, ,
\end{aligned}
$$

where the last inequality is due to Lemma A.1, since $\theta(\tilde{\mathbf{w}}, \tilde{\mathbf{v}}) \leq \frac{3\pi}{4}$.

If $\|\tilde{\mathbf{v}}\| \leq \frac{1}{4c}$, then

$$
b_{\mathbf{v}} \geq \sqrt{1 - \frac{1}{16c^2}} \geq \sqrt{1 - \frac{1}{16}} = \frac{\sqrt{15}}{4} > \frac{3}{4} \, ,
$$

and hence

$$
\mathbf{v}^\top \mathbf{x} = \tilde{\mathbf{v}}^\top \tilde{\mathbf{x}} + b_{\mathbf{v}} > -\frac{1}{4c} \cdot c + \frac{3}{4} = \frac{1}{2} \geq \xi \geq \frac{\xi}{2\sqrt{c}} \, .
$$

Therefore,

$$
\Pr_{\mathbf{x}} \left[ \bar{\mathbf{w}}^\top \mathbf{x} \geq 2\sqrt{c}\xi, \mathbf{v}^\top \mathbf{x} \geq \frac{\xi}{2\sqrt{c}} \right] = \Pr_{\mathbf{x}} \left[ \bar{\mathbf{w}}^\top \mathbf{x} \geq 2\sqrt{c}\xi \right] = \Pr_{\mathbf{x}} \left[ \tilde{\bar{\mathbf{w}}}^\top \tilde{\mathbf{x}} \geq 2\sqrt{c}\xi \right] \, .
$$

For $\tilde{\mathbf{u}} \in \mathbb{R}^d$ such that $\|\tilde{\mathbf{u}}\| = 1$ and $\theta(\tilde{\mathbf{w}}, \tilde{\mathbf{u}}) = \frac{3\pi}{4}$, Lemma A.1 implies that the above is at least

$$
\Pr_{\mathbf{x}} \left[ \tilde{\bar{\mathbf{w}}}^\top \tilde{\mathbf{x}} \geq 2\sqrt{c}\xi, \tilde{\mathbf{u}}^\top \tilde{\mathbf{x}} \geq 2\sqrt{c}\xi \right] \geq \frac{\beta \left( \alpha \sin\left(\frac{\pi}{8}\right) - 2\sqrt{c}\xi \right)^2}{4 \sin\left(\frac{\pi}{8}\right)} \, .
$$

$\square$

**Lemma D.8.** *If $b_{\mathbf{v}} < 0$ and $-\frac{b_{\mathbf{v}}}{\|\tilde{\mathbf{v}}\|} \leq \alpha \cdot \frac{\sin\left(\frac{\pi}{8}\right)}{4}$, then*

$$\Pr_{\mathbf{x}}\left[\bar{\mathbf{w}}^{\top}\mathbf{x} \geq 2\sqrt{c}\xi, \mathbf{v}^{\top}\mathbf{x} \geq \frac{\xi}{2\sqrt{c}}\right] \geq \frac{\beta\left(\alpha\sin\left(\frac{\pi}{8}\right) - 2\sqrt{c}\xi\right)^2}{4\sin\left(\frac{\pi}{8}\right)} .$$

*Proof.*

$$\Pr_{\mathbf{x}}\left[\bar{\mathbf{w}}^{\top}\mathbf{x} \geq 2\sqrt{c}\xi, \mathbf{v}^{\top}\mathbf{x} \geq \frac{\xi}{2\sqrt{c}}\right] = \Pr_{\mathbf{x}}\left[\bar{\mathbf{w}}^{\top}\mathbf{x} \geq 2\sqrt{c}\xi, \tilde{\mathbf{v}}^{\top}\tilde{\mathbf{x}} \geq \frac{\xi}{2\sqrt{c}} - b_{\mathbf{v}}\right]$$

$$= \Pr_{\mathbf{x}}\left[\tilde{\bar{\mathbf{w}}}^{\top}\tilde{\mathbf{x}} \geq 2\sqrt{c}\xi, \bar{\tilde{\mathbf{v}}}^{\top}\tilde{\mathbf{x}} \geq \frac{\xi}{2\sqrt{c}\|\tilde{\mathbf{v}}\|} - \frac{b_{\mathbf{v}}}{\|\tilde{\mathbf{v}}\|}\right] . \qquad (19)$$

Moreover, we have

$$\left(\alpha \cdot \frac{\sin\left(\frac{\pi}{8}\right)}{4}\right)^2 \geq \left(\frac{b_{\mathbf{v}}}{\|\tilde{\mathbf{v}}\|}\right)^2 = \frac{1 - \|\tilde{\mathbf{v}}\|^2}{\|\tilde{\mathbf{v}}\|^2} = \frac{1}{\|\tilde{\mathbf{v}}\|^2} - 1 ,$$

and hence

$$\|\tilde{\mathbf{v}}\|^2 \geq \frac{16}{\alpha^2\sin^2\left(\frac{\pi}{8}\right) + 16} \geq \frac{16}{\left(\alpha\sin\left(\frac{\pi}{8}\right) + 4\right)^2} \geq \frac{16}{(c \cdot 1 + 4c)^2} ,$$

where in the last inequality we used $c \geq \alpha$ and $c \geq 1$. Thus,

$$\|\tilde{\mathbf{v}}\| \geq \frac{4}{5c} \geq \frac{1}{2c} .$$

Combining the above with Eq. (19), and using $-\frac{b_{\mathbf{v}}}{\|\tilde{\mathbf{v}}\|} \leq \alpha \cdot \frac{\sin\left(\frac{\pi}{8}\right)}{4}$, we have

$$\Pr_{\mathbf{x}}\left[\bar{\mathbf{w}}^{\top}\mathbf{x} \geq 2\sqrt{c}\xi, \mathbf{v}^{\top}\mathbf{x} \geq \frac{\xi}{2\sqrt{c}}\right] \geq \Pr_{\mathbf{x}}\left[\tilde{\bar{\mathbf{w}}}^{\top}\tilde{\mathbf{x}} \geq 2\sqrt{c}\xi, \bar{\tilde{\mathbf{v}}}^{\top}\tilde{\mathbf{x}} \geq \sqrt{c}\xi + \alpha \cdot \frac{\sin\left(\frac{\pi}{8}\right)}{4}\right]$$

$$= \Pr_{\mathbf{x}}\left[\tilde{\bar{\mathbf{w}}}^{\top}\tilde{\mathbf{x}} \geq 2\sqrt{c}\xi, \bar{\tilde{\mathbf{v}}}^{\top}\tilde{\mathbf{x}} \geq 2\sqrt{c}\xi\right]$$

$$\geq \frac{\beta\left(\alpha\sin\left(\frac{\pi}{8}\right) - 2\sqrt{c}\xi\right)^2}{4\sin\left(\frac{\pi}{8}\right)} ,$$

where the last inequality is due to Lemma A.1, since $\theta(\tilde{\mathbf{w}}, \tilde{\mathbf{v}}) \leq \frac{3\pi}{4}$. $\qquad \square$

Combining Eq. (18) with Lemmas D.7 and D.8, we have

$$\mathbb{E}_{\mathbf{x}}\,\sigma(\bar{\mathbf{w}}^{\top}\mathbf{x})\sigma(\mathbf{v}^{\top}\mathbf{x}) \geq \xi^2 \cdot \frac{\beta\left(\alpha\sin\left(\frac{\pi}{8}\right) - 2\sqrt{c}\xi\right)^2}{4\sin\left(\frac{\pi}{8}\right)} = \frac{\alpha^2\sin^2\left(\frac{\pi}{8}\right)}{16c} \cdot \frac{\beta\left(\frac{\alpha}{2}\sin\left(\frac{\pi}{8}\right)\right)^2}{4\sin\left(\frac{\pi}{8}\right)}$$

$$= \frac{\alpha^4\beta\sin^3\left(\frac{\pi}{8}\right)}{256c} = M .$$

Plugging the above into Eq. (17) we have

$$F(\mathbf{w}) \leq F(\mathbf{0}) + \frac{\|\mathbf{w}\|^2 c^2}{2} - \|\mathbf{w}\| \cdot M .$$

The above expression is smaller than $F(\mathbf{0})$ if $\|\mathbf{w}\| < \frac{2M}{c^2}$.

## E   Discussion on the Assumption on $b_{\mathbf{v}}$

In Corollary 5.5 we had an assumption that $-\frac{b_{\mathbf{v}}}{\|\tilde{\mathbf{v}}\|} \leq \alpha \cdot \frac{\sin\left(\frac{\pi}{8}\right)}{4}$. This implies that either the bias term $b_{\mathbf{v}}$ is positive, or it is negative but not too large. Here we discuss why this assumption is crucial for the proof of the theorem, and what can we still say when this assumption does not hold.

In Theorem 3.2 we showed an example with $b_{\mathbf{v}} < 0$ where gradient descent with random initialization does not converge w.h.p. to a global minimum even asymptotically[4]. In the example from Theorem 3.2 we have $-\frac{b_{\mathbf{v}}}{\|\tilde{\mathbf{v}}\|} = r\left(1 - \frac{1}{2d^2}\right)$, and the input distribution is uniform over a ball of radius $r$. In this case, we must choose $\alpha$ from Assumption 5.3 to be smaller than $r$ (otherwise $\beta = 0$) and hence $-\frac{b_{\mathbf{v}}}{\|\tilde{\mathbf{v}}\|} > \alpha\left(1 - \frac{1}{2d^2}\right)$. Therefore it does not satisfy the assumption $-\frac{b_{\mathbf{v}}}{\|\tilde{\mathbf{v}}\|} \le \alpha \cdot \frac{\sin\left(\frac{\pi}{8}\right)}{4}$ (already for $d > 1$). If we choose, e.g., $\alpha = \frac{r}{2}$, then the example from Theorem 3.2 satisfies $-\frac{b_{\mathbf{v}}}{\|\tilde{\mathbf{v}}\|} = \alpha\left(2 - \frac{1}{d^2}\right) \le 2\alpha$. It implies that our assumption on $-\frac{b_{\mathbf{v}}}{\|\tilde{\mathbf{v}}\|}$ is tight up to a constant factor, and is also crucial for the proof, since already for $-\frac{b_{\mathbf{v}}}{\|\tilde{\mathbf{v}}\|} = 2\alpha$ we have an example of convergence to a non-global minimum.

On the other hand, if $-\frac{b_{\mathbf{v}}}{\|\tilde{\mathbf{v}}\|} > \alpha \cdot \frac{\sin\left(\frac{\pi}{8}\right)}{4}$ (i.e. the assumption does not hold) we can calculate the loss at zero:

$$
\begin{aligned}
F(\mathbf{0}) &= \frac{1}{2} \cdot \mathbb{E}_{\mathbf{x}}\left[\left(\sigma(\mathbf{v}^\top \mathbf{x})\right)^2\right] = \frac{1}{2} \cdot \mathbb{E}_{\mathbf{x}}\left[\mathbb{1}(\tilde{\mathbf{v}}^\top \tilde{\mathbf{x}} + b_{\mathbf{v}} \ge 0)\left(\tilde{\mathbf{v}}^\top \tilde{\mathbf{x}} + b_{\mathbf{v}}\right)^2\right] \\
&= \frac{1}{2} \cdot \mathbb{E}_{\mathbf{x}}\left[\mathbb{1}\left(\bar{\tilde{\mathbf{v}}}^\top \tilde{\mathbf{x}} \ge -\frac{b_{\mathbf{v}}}{\|\tilde{\mathbf{v}}\|}\right)\|\tilde{\mathbf{v}}\|^2\left(\bar{\tilde{\mathbf{v}}}^\top \tilde{\mathbf{x}} + \frac{b_{\mathbf{v}}}{\|\tilde{\mathbf{v}}\|}\right)^2\right] \\
&\le \frac{\|\tilde{\mathbf{v}}\|^2}{2} \cdot \mathbb{E}_{\mathbf{x}}\left[\mathbb{1}\left(\bar{\tilde{\mathbf{v}}}^\top \tilde{\mathbf{x}} \ge \alpha \cdot \frac{\sin\left(\frac{\pi}{8}\right)}{4}\right)\left(\bar{\tilde{\mathbf{v}}}^\top \tilde{\mathbf{x}}\right)^2\right] .
\end{aligned}
$$

Let $\epsilon > 0$ be a small constant. Suppose that the distribution $\tilde{\mathcal{D}}$ is spherically symmetric, and that $\alpha$ is large, such that the above expectation is smaller than $\epsilon$. For such $\alpha$, we either have $-\frac{b_{\mathbf{v}}}{\|\tilde{\mathbf{v}}\|} \le \alpha \cdot \frac{\sin\left(\frac{\pi}{8}\right)}{4}$, in which case gradient descent converges w.h.p. to the global minimum, or $-\frac{b_{\mathbf{v}}}{\|\tilde{\mathbf{v}}\|} > \alpha \cdot \frac{\sin\left(\frac{\pi}{8}\right)}{4}$, in which case the loss at $\mathbf{w} = \mathbf{0}$ is already almost as good as the global minimum. For standard Gaussian distribution, we can choose $\alpha$ to be a large enough constant that depends only on $\epsilon$ (independent of the input dimension), hence $\beta$ will also be independent of $d$. This means that for standard Gaussian distribution, for every constant $\epsilon > 0$ we can ensure either convergence to a global minimum, or the loss at $\mathbf{0}$ is already $\epsilon$-optimal.

Note that in Remark 3.3 we have shown another distribution which is non-symmetric and depends on the target $\mathbf{v}$, such that the loss $F(\mathbf{0})$ is highly sub-optimal, but gradient flow converges to such a point with probability close to $\frac{1}{2}$.

# F   Proofs from Section 6

Before proving Theorem 6.2, we first proof two auxiliary propositions which bounds certain areas for which the vector $\mathbf{w}$ cannot reach during the optimization process. The first proposition shows that if the norm of $\tilde{\mathbf{w}}$ is small, and its bias is close to zero, then the bias must get larger. The second proposition shows that if the norm of $\tilde{\mathbf{w}}$ is small, and the bias is negative, then the norm of $\tilde{\mathbf{w}}$ must get larger.

**Proposition F.1.** *Assume that $\|\tilde{\mathbf{w}} - \tilde{\mathbf{v}}\|^2 \le 1$, and that Assumption 6.1 holds. If $\|\tilde{\mathbf{w}}\| \le 0.4$ and $b_{\mathbf{w}} \in \left[0, \frac{\alpha^3 \beta}{640}\right]$ then $(\nabla F(\mathbf{w}))_{d+1} \le -\frac{\alpha^3 \beta}{640}$.*

*Proof.* The $d + 1$ coordinate of the distribution $\mathcal{D}$ is a constant 1. We denote by $\tilde{\mathcal{D}}$ the first $d$ coordinates of the distribution $\mathcal{D}$. Hence, we can write:

$$
\begin{aligned}
(\nabla F(\mathbf{w}))_{d+1} &= \mathbb{E}_{\mathbf{x} \sim \mathcal{D}}\left[(\sigma(\mathbf{w}^\top \mathbf{x}) - \sigma(\mathbf{v}^\top \mathbf{x}))\mathbb{1}(\mathbf{w}^\top \mathbf{x} > 0)\right] \\
&= \mathbb{E}_{\tilde{\mathbf{x}} \sim \tilde{\mathcal{D}}}\left[(\sigma(\tilde{\mathbf{w}}^\top \tilde{\mathbf{x}} + b_{\mathbf{w}}) - \sigma(\tilde{\mathbf{v}}^\top \tilde{\mathbf{x}} + b_{\mathbf{v}}))\mathbb{1}(\tilde{\mathbf{w}}^\top \tilde{\mathbf{x}} > -b_{\mathbf{w}})\right] \\
&= \mathbb{E}_{\tilde{\mathbf{x}} \sim \tilde{\mathcal{D}}}\left[(\tilde{\mathbf{w}}^\top \tilde{\mathbf{x}} + b_{\mathbf{w}}) \cdot \mathbb{1}(\tilde{\mathbf{w}}^\top \tilde{\mathbf{x}} > -b_{\mathbf{w}})\right] - \\
&\quad - \mathbb{E}_{\tilde{\mathbf{x}} \sim \tilde{\mathcal{D}}}\left[(\tilde{\mathbf{v}}^\top \tilde{\mathbf{x}} + b_{\mathbf{v}}) \cdot \mathbb{1}(\tilde{\mathbf{w}}^\top \tilde{\mathbf{x}} > -b_{\mathbf{w}}, \tilde{\mathbf{v}}^\top \tilde{\mathbf{x}} > -b_{\mathbf{v}})\right] \quad (20)
\end{aligned}
$$

---

[4]In Theorem 3.2 we have $\|\mathbf{v}\| \ne 1$, but it still holds if we normalize $\mathbf{v}$, namely, replace $\mathbf{v}$ with $\frac{\mathbf{v}}{\|\mathbf{v}\|}$.

We will bound each term in Eq. (20) separately. Using the assumption that $\tilde{\mathcal{D}}$ is spherically symmetric, we can assume w.l.o.g that $\tilde{\mathbf{w}} = \|\tilde{\mathbf{w}}\|\mathbf{e}_1$, the first unit vector. Hence we have that :

$$
\begin{aligned}
&\mathbb{E}_{\tilde{\mathbf{x}}\sim\tilde{\mathcal{D}}}\left[(\tilde{\mathbf{w}}^\top\tilde{\mathbf{x}} + b_{\mathbf{w}}) \cdot \mathbb{1}(\tilde{\mathbf{w}}^\top\tilde{\mathbf{x}} > -b_{\mathbf{w}})\right] \\
=&\mathbb{E}_{\tilde{\mathbf{x}}\sim\tilde{\mathcal{D}}}\left[(\|\tilde{\mathbf{w}}\|x_1 + b_{\mathbf{w}}) \cdot \mathbb{1}\left(x_1 > -\frac{b_{\mathbf{w}}}{\|\tilde{\mathbf{w}}\|}\right)\right] \\
=&\|\tilde{\mathbf{w}}\|\mathbb{E}_{\tilde{\mathbf{x}}\sim\tilde{\mathcal{D}}}\left[x_1\mathbb{1}\left(x_1 > -\frac{b_{\mathbf{w}}}{\|\tilde{\mathbf{w}}\|}\right)\right] + b_{\mathbf{w}}\mathbb{E}_{\tilde{\mathbf{x}}\sim\tilde{\mathcal{D}}}\left[\mathbb{1}\left(x_1 > -\frac{b_{\mathbf{w}}}{\|\tilde{\mathbf{w}}\|}\right)\right] \\
\overset{(a)}{\leq}&0.4\mathbb{E}_{\tilde{\mathbf{x}}\sim\tilde{\mathcal{D}}}\left[x_1\mathbb{1}\left(x_1 > -\frac{b_{\mathbf{w}}}{\|\tilde{\mathbf{w}}\|}\right)\right] + b_{\mathbf{w}} \\
\overset{(b)}{\leq}&0.4\mathbb{E}_{\tilde{\mathbf{x}}\sim\tilde{\mathcal{D}}}\left[x_1\mathbb{1}(x_1 > 0)\right] + b_{\mathbf{w}} \; .
\end{aligned}
\tag{21}
$$

Here, (a) is since $\|\tilde{\mathbf{w}}\| \leq 0.4$, and $\mathbb{E}_{\tilde{\mathbf{x}}\sim\tilde{\mathcal{D}}}\left[\mathbb{1}\left(x_1 > -\frac{b_{\mathbf{w}}}{\|\tilde{\mathbf{w}}\|}\right)\right] \leq 1$, (b) is since $b_{\mathbf{w}} \geq 0$, hence

$$
\mathbb{E}_{\tilde{\mathbf{x}}\sim\tilde{\mathcal{D}}}\left[x_1\mathbb{1}\left(0 > x_1 > -\frac{b_{\mathbf{w}}}{\|\tilde{\mathbf{w}}\|}\right)\right] \leq 0 \; .
$$

For the second term of Eq. (20), we assumed that $\|\tilde{\mathbf{w}} - \tilde{\mathbf{v}}\|^2 \leq 1$, which shows that $\theta(\tilde{\mathbf{w}}, \tilde{\mathbf{v}}) \leq \frac{\pi}{2}$, and the term is largest when this angle is largest. Hence, to lower bound this term we can assume that $\theta(\tilde{\mathbf{w}}, \tilde{\mathbf{v}}) = \frac{\pi}{2}$, and since the distribution is spherically symmetric we can also assume w.l.o.g that $\tilde{\mathbf{v}} = \mathbf{e}_2$, the second unit vector. Now we can bound:

$$
\begin{aligned}
&\mathbb{E}_{\tilde{\mathbf{x}}\sim\tilde{\mathcal{D}}}\left[(\tilde{\mathbf{v}}^\top\tilde{\mathbf{x}} + b_{\mathbf{v}}) \cdot \mathbb{1}(\tilde{\mathbf{w}}^\top\tilde{\mathbf{x}} > -b_{\mathbf{w}}, \tilde{\mathbf{v}}^\top\tilde{\mathbf{x}} > -b_{\mathbf{v}})\right] \\
\geq&\mathbb{E}_{\tilde{\mathbf{x}}\sim\tilde{\mathcal{D}}}\left[(x_2 + b_{\mathbf{v}}) \cdot \mathbb{1}\left(x_1 > -\frac{b_{\mathbf{w}}}{\|\tilde{\mathbf{w}}\|}, x_2 > -b_{\mathbf{v}}\right)\right] \\
\geq&\frac{1}{2}\mathbb{E}_{\tilde{\mathbf{x}}\sim\tilde{\mathcal{D}}}\left[(x_2 + b_{\mathbf{v}}) \cdot \mathbb{1}(x_2 > -b_{\mathbf{v}})\right] \\
\geq&\frac{1}{2}\mathbb{E}_{\tilde{\mathbf{x}}\sim\tilde{\mathcal{D}}}\left[x_2 \cdot \mathbb{1}(x_2 > 0)\right] + \frac{1}{2}\mathbb{E}_{\tilde{\mathbf{x}}\sim\tilde{\mathcal{D}}}\left[(x_2 + b_{\mathbf{v}}) \cdot \mathbb{1}(0 > x_2 > -b_{\mathbf{v}})\right] \\
\geq&\frac{1}{2}\mathbb{E}_{\tilde{\mathbf{x}}\sim\tilde{\mathcal{D}}}\left[x_2 \cdot \mathbb{1}(x_2 > 0)\right] = \frac{1}{2}\mathbb{E}_{\tilde{\mathbf{x}}\sim\tilde{\mathcal{D}}}\left[x_1 \cdot \mathbb{1}(x_1 > 0)\right] \; ,
\end{aligned}
\tag{22}
$$

where we used the assumption $b_{\mathbf{v}} \geq 0$ and the symmetry of the distribution. Combining Eq. (21), Eq. (22) with Eq. (20) we get:

$$
(\nabla F(\mathbf{w}))_{d+1} \leq b_{\mathbf{w}} - 0.1\mathbb{E}_{\tilde{\mathbf{x}}\sim\tilde{\mathcal{D}}}\left[x_1 \cdot \mathbb{1}(x_1 > 0)\right] \; .
$$

Let $\hat{\mathcal{D}}$ be the marginal distribution of $\tilde{\mathcal{D}}$ on the plane spanned by $\mathbf{e}_1$ and $\mathbf{e}_2$, and denote by $\hat{\mathbf{x}}$ the projection of $\tilde{\mathbf{x}}$ on this plane. By Assumption 6.1(3) we have that the pdf of this distribution is at least $\beta$ in a ball or radius $\alpha$ around the origin. This way we can bound:

$$
\begin{aligned}
\mathbb{E}_{\tilde{\mathbf{x}}\sim\tilde{\mathcal{D}}}\left[x_1 \cdot \mathbb{1}(x_1 > 0)\right] &= \mathbb{E}_{\hat{\mathbf{x}}\sim\hat{\mathcal{D}}}\left[x_1 \cdot \mathbb{1}(x_1 > 0)\right] \\
&\geq \frac{\alpha\beta}{2}\mathrm{P}(\alpha/2 < \|\hat{\mathbf{x}}\| < \alpha, \; x_1 > \alpha/2) \\
&\geq \frac{\alpha\beta}{2}\mathrm{P}(x_1 \in [\alpha/2, 3\alpha/4], \; x_2 \in [-\alpha/4, \alpha/4]) = \frac{\alpha^3\beta}{32} \; .
\end{aligned}
$$

Combining the above, and using the assumption on $b_{\mathbf{w}}$ we get that:

$$
(\nabla F(\mathbf{w}))_{d+1} \leq b_{\mathbf{w}} - \frac{\alpha^3\beta}{320} \leq -\frac{\alpha^3\beta}{640}
$$

$\square$

**Proposition F.2.** *Assume that $\|\tilde{\mathbf{w}} - \tilde{\mathbf{v}}\| < 1$, and Assumption 6.1 holds. Denote by $\tau = \frac{\mathbb{E}_{\tilde{\mathbf{x}} \sim \tilde{\mathcal{D}}}[|x_1 x_2|]}{\mathbb{E}_{\tilde{\mathbf{x}} \sim \tilde{\mathcal{D}}}[x_1^2]}$ where $\tilde{\mathcal{D}}$ is the projection of the distribution $\mathcal{D}$ on its first $d$ coordinates. If $\|\tilde{\mathbf{w}}\| \leq \frac{\tau}{2}$ and $b_{\mathbf{w}} \leq 0$ then $\langle \nabla F(\mathbf{w})_{1:d}, \tilde{\mathbf{w}} \rangle \leq 0$.*

*Proof.* Denote by $\tilde{\mathcal{D}}$ the projection of the distribution $\mathcal{D}$ on its first $d$ coordinates, we have that:

$$
\begin{aligned}
\langle \nabla F(\mathbf{w})_{1:d}, \tilde{\mathbf{w}} \rangle &= \mathbb{E}_{\mathbf{x} \sim \mathcal{D}} \left[ \left( \sigma(\mathbf{w}^\top \mathbf{x}) - \sigma(\mathbf{v}^\top \mathbf{x}) \right) \mathbb{1}(\mathbf{w}^\top \mathbf{x} > 0) \tilde{\mathbf{w}}^\top \tilde{\mathbf{x}} \right] \\
&= \mathbb{E}_{\tilde{\mathbf{x}} \sim \tilde{\mathcal{D}}} \left[ \left( \sigma(\tilde{\mathbf{w}}^\top \tilde{\mathbf{x}} + b_{\mathbf{w}}) - \sigma(\tilde{\mathbf{v}}^\top \tilde{\mathbf{x}} + b_{\mathbf{v}}) \right) \mathbb{1}(\tilde{\mathbf{w}}^\top \tilde{\mathbf{x}} > -b_{\mathbf{w}}) \tilde{\mathbf{w}}^\top \tilde{\mathbf{x}} \right] \\
&\leq \mathbb{E}_{\tilde{\mathbf{x}} \sim \tilde{\mathcal{D}}} \left[ \left( \tilde{\mathbf{w}}^\top \tilde{\mathbf{x}} + b_{\mathbf{w}} - \sigma(\tilde{\mathbf{v}}^\top \tilde{\mathbf{x}}) \right) \cdot \mathbb{1}(\tilde{\mathbf{w}}^\top \tilde{\mathbf{x}} > -b_{\mathbf{w}}) \tilde{\mathbf{w}}^\top \tilde{\mathbf{x}} \right] . \quad (23)
\end{aligned}
$$

The inequality above is since $b_{\mathbf{v}} \geq 0$. Recall that our goal is to prove that the above term is negative, hence we will divide it by $\|\tilde{\mathbf{w}}\|$. Also, since the distribution $\tilde{\mathcal{D}}$ is symmetric we can assume w.l.o.g that $\tilde{\mathbf{w}} = \|\tilde{\mathbf{w}}\| \mathbf{e}_1$. Hence, it is enough to prove that the following term is non-positive:

$$
\begin{aligned}
& \|\tilde{\mathbf{w}}\| \mathbb{E}_{\tilde{\mathbf{x}} \sim \tilde{\mathcal{D}}} \left[ \left( \|\tilde{\mathbf{w}}\| x_1 + b_{\mathbf{w}} - \sigma(\tilde{\mathbf{v}}^\top \tilde{\mathbf{x}}) \right) \cdot \mathbb{1}\left( x_1 > -\frac{b_{\mathbf{w}}}{\|\tilde{\mathbf{w}}\|} \right) x_1 \right] \\
&= \|\tilde{\mathbf{w}}\| \mathbb{E}_{\tilde{\mathbf{x}} \sim \tilde{\mathcal{D}}} \left[ \left( \|\tilde{\mathbf{w}}\| x_1^2 + b_{\mathbf{w}} x_1 \right) \cdot \mathbb{1}\left( x_1 > -\frac{b_{\mathbf{w}}}{\|\tilde{\mathbf{w}}\|} \right) \right] - \|\tilde{\mathbf{w}}\| \mathbb{E}_{\tilde{\mathbf{x}} \sim \tilde{\mathcal{D}}} \left[ x_1 \tilde{\mathbf{v}}^\top \tilde{\mathbf{x}} \cdot \mathbb{1}\left( x_1 > -\frac{b_{\mathbf{w}}}{\|\tilde{\mathbf{w}}\|}, \tilde{\mathbf{v}}^\top \tilde{\mathbf{x}} > -b_{\mathbf{v}} \right) \right] \\
&\leq \|\tilde{\mathbf{w}}\| \mathbb{E}_{\tilde{\mathbf{x}} \sim \tilde{\mathcal{D}}} \left[ \left( \|\tilde{\mathbf{w}}\| x_1^2 + b_{\mathbf{w}} x_1 \right) \cdot \mathbb{1}\left( x_1 > -\frac{b_{\mathbf{w}}}{\|\tilde{\mathbf{w}}\|} \right) \right] - \|\tilde{\mathbf{w}}\| \mathbb{E}_{\tilde{\mathbf{x}} \sim \tilde{\mathcal{D}}} \left[ x_1 \tilde{\mathbf{v}}^\top \tilde{\mathbf{x}} \cdot \mathbb{1}\left( x_1 > -\frac{b_{\mathbf{w}}}{\|\tilde{\mathbf{w}}\|}, \tilde{\mathbf{v}}^\top \tilde{\mathbf{x}} > 0 \right) \right] .
\end{aligned}
$$
$$(24)$$

We will first bound the second term above. Since the term only depend on inner products between $\tilde{\mathbf{w}}, \tilde{\mathbf{v}}$ with $\tilde{\mathbf{x}}$, we can consider the marginal distribution $\hat{\mathcal{D}}$, of $\tilde{\mathcal{D}}$ on the plane spanned by $\tilde{\mathbf{w}}$ and $\tilde{\mathbf{v}}$. Since $\tilde{\mathcal{D}}$ is symmetric we can assume w.l.o.g that $\hat{\mathcal{D}}$ is spanned by the first two coordinates $x_1$ and $x_2$. Let $\hat{\tilde{\mathbf{v}}}$ be the projection of $\tilde{\mathbf{v}}$ on this plane, then we can write $\hat{\tilde{\mathbf{v}}} = (v_1, v_2)$ where $v_1^2 + v_2^2 = 1$. Note that since the distribution $\hat{\mathcal{D}}$ is symmetric, we have that $\mathbb{E}[x_1^2] = \mathbb{E}[x_2^2]$. By Cauchy-Schwarz we have:

$$
|\mathrm{cov}_{\hat{\mathcal{D}}}(x_1, x_2)| \leq \sqrt{\mathrm{var}_{\hat{\mathcal{D}}}(x_1) \cdot \mathrm{var}_{\hat{\mathcal{D}}}(x_2)} = \mathrm{var}_{\hat{\mathcal{D}}}(x_1)
$$

Again, by symmetry of $\hat{\mathcal{D}}$ we have that $\mathbb{E}[x_1] = \mathbb{E}[x_2]$. Opening up the above terms we get that $\mathbb{E}[x_1 \cdot x_2] \leq \mathbb{E}[x_1^2]$. Also, we assumed that $\|\tilde{\mathbf{w}} - \tilde{\mathbf{v}}\| < 1$, then $\theta(\tilde{\mathbf{v}}, \tilde{\mathbf{w}}) \leq \frac{\pi}{2}$ which means that $v_1 \geq 0$. Hence, the second term of Eq. (24) is smallest when $\tilde{\mathbf{v}} = \mathbf{e}_2$. In total, we can bound Eq. (24) by:

$$
\begin{aligned}
& \|\tilde{\mathbf{w}}\| \mathbb{E}_{\tilde{\mathbf{x}} \sim \tilde{\mathcal{D}}} \left[ \left( \|\tilde{\mathbf{w}}\| x_1^2 + b_{\mathbf{w}} x_1 \right) \cdot \mathbb{1}\left( x_1 > -\frac{b_{\mathbf{w}}}{\|\tilde{\mathbf{w}}\|} \right) \right] - \|\tilde{\mathbf{w}}\| \mathbb{E}_{\tilde{\mathbf{x}} \sim \tilde{\mathcal{D}}} \left[ x_1 x_2 \cdot \mathbb{1}\left( x_1 > -\frac{b_{\mathbf{w}}}{\|\tilde{\mathbf{w}}\|}, x_2 > 0 \right) \right] \\
&\leq \|\tilde{\mathbf{w}}\| \mathbb{E}_{\tilde{\mathbf{x}} \sim \tilde{\mathcal{D}}} \left[ \left( \|\tilde{\mathbf{w}}\| x_1^2 + b_{\mathbf{w}} x_1 - \frac{1}{2} x_1 |x_2| \right) \cdot \mathbb{1}\left( x_1 > -\frac{b_{\mathbf{w}}}{\|\tilde{\mathbf{w}}\|} \right) \right] \\
&= \|\tilde{\mathbf{w}}\| \mathbb{E}_{\tilde{\mathbf{x}} \sim \tilde{\mathcal{D}}} \left[ \left( \|\tilde{\mathbf{w}}\| x_1 + b_{\mathbf{w}} - \frac{1}{2} |x_2| \right) \cdot x_1 \mathbb{1}\left( x_1 > -\frac{b_{\mathbf{w}}}{\|\tilde{\mathbf{w}}\|} \right) \right] \quad (25)
\end{aligned}
$$

By our assumption, $b_{\mathbf{w}} \leq 0$. Both terms inside the expectation in Eq. (25) are largest when $b_{\mathbf{w}} = 0$. Hence, we can bound Eq. (25) by:

$$
\begin{aligned}
& \|\tilde{\mathbf{w}}\| \mathbb{E}_{\tilde{\mathbf{x}} \sim \tilde{\mathcal{D}}} \left[ \left( \|\tilde{\mathbf{w}}\| x_1 - \frac{1}{2} |x_2| \right) \cdot x_1 \mathbb{1}(x_1 > 0) \right] \\
&= \frac{\|\tilde{\mathbf{w}}\|^2}{2} \mathbb{E}_{\tilde{\mathbf{x}} \sim \tilde{\mathcal{D}}} \left[ x_1^2 \right] - \frac{\|\tilde{\mathbf{w}}\|}{4} \mathbb{E}_{\tilde{\mathbf{x}} \sim \tilde{\mathcal{D}}} \left[ |x_1 x_2| \right] \\
&\leq \frac{\|\tilde{\mathbf{w}}\|^2}{2} \mathbb{E}_{\tilde{\mathbf{x}} \sim \tilde{\mathcal{D}}} \left[ x_1^2 \right] - \frac{\|\tilde{\mathbf{w}}\| \tau}{4} \mathbb{E}_{\tilde{\mathbf{x}} \sim \tilde{\mathcal{D}}} \left[ x_1^2 \right] = c_1 \left( \frac{\|\tilde{\mathbf{w}}\|^2}{2} - \frac{\|\tilde{\mathbf{w}}\| \tau}{4} \right) . \quad (26)
\end{aligned}
$$

In particular, for $\|\tilde{\mathbf{w}}\| \leq \frac{\tau}{2}$, Eq. (26) non-positive.

$\square$

We are now ready to prove the main theorem:

*Proof of Theorem 6.2.* Denote $b_t = \max\{0, -\frac{b_{\mathbf{w}_t}}{\|\tilde{\mathbf{w}}_t\|}\}$. We will show by induction on the iterations of gradient descent that throughout the optimization process $b_t < 2.4 \cdot \max\left\{1, \frac{1}{\sqrt{\tau}}\right\}$ and $\theta(\tilde{\mathbf{w}}_t, \tilde{\mathbf{v}}) \leq \frac{\pi}{2}$ for every $t \geq 0$.

By the assumption on the initialization we have that $\|\tilde{\mathbf{w}}_0 - \tilde{\mathbf{v}}\|^2 \leq \|\mathbf{w}_0 - \mathbf{v}\|^2 < 1$, and also $\|\tilde{\mathbf{v}}\| = 1$, hence $\theta(\tilde{\mathbf{w}}_0, \tilde{\mathbf{v}}) \leq \frac{\pi}{2}$. We also have that $b_{\mathbf{w}_0} \geq 0$, hence $b_0 = 0$ this proves the case of $t = 0$. Assume this is true for $t$. We will bound the norm of the gradient of the objective using Jensen's inequality:

$$\|\nabla F(\mathbf{w})\|^2 \leq \mathbb{E}_{\mathbf{x} \sim \mathcal{D}}\left[(\sigma(\mathbf{w}^\top \mathbf{x}) - \sigma(\mathbf{v}^\top \mathbf{x}))^2 \mathbb{1}(\mathbf{w}^\top \mathbf{x} > 0)\mathbf{x}^\top \mathbf{x}\right]$$
$$\leq \mathbb{E}_{\mathbf{x} \sim \mathcal{D}}\left[(\mathbf{w}^\top \mathbf{x} - \mathbf{v}^\top \mathbf{x})^2 \mathbf{x}^\top \mathbf{x}\right]$$
$$\leq \|\mathbf{w} - \mathbf{v}\|^2 \mathbb{E}_{\mathbf{x} \sim \mathcal{D}}\left[\|\mathbf{x}\|^4\right] = \|\mathbf{w} - \mathbf{v}\|^2 c. \tag{27}$$

For the $(t+1)$-th iteration of gradient descent we have that:

$$\|\mathbf{w}_{t+1} - \mathbf{v}\|^2 = \|\mathbf{w}_t - \eta\nabla F(\mathbf{w}_t) - \mathbf{v}\|^2$$
$$= \|\mathbf{w}_t - \mathbf{v}\|^2 - 2\eta\langle\nabla F(\mathbf{w}_t), \mathbf{w}_t - \mathbf{v}\rangle + \eta^2\|\nabla F(\mathbf{w}_t)\|^2$$
$$\leq \|\mathbf{w}_t - \mathbf{v}\|^2 - 2\eta\langle\nabla F(\mathbf{w}_t), \mathbf{w}_t - \mathbf{v}\rangle + \eta^2 c\|\mathbf{w}_t - \mathbf{v}\|^2. \tag{28}$$

By Theorem A.2, and the induction assumption on $\theta(\tilde{\mathbf{w}}_t, \tilde{\mathbf{v}})$ we get that there is a universal constant $c_0$, such that $\langle\nabla F(\mathbf{w}_t), \mathbf{w}_t - \mathbf{v}\rangle \geq \frac{c_0\beta(\alpha - \sqrt{2}b_t)}{\alpha^2}\|\mathbf{w}_t - \mathbf{v}\|^2$. Using the induction assumption that $b_t < 2.4 \cdot \max\left\{1, \frac{1}{\sqrt{\tau}}\right\}$ and Assumption 6.1(3) we can bound $(\alpha - \sqrt{2}b_t) \geq 0.1$. In total we get that $\langle\nabla F(\mathbf{w}_t), \mathbf{w}_t - \mathbf{v}\rangle \geq \frac{c_0\beta}{10\alpha^2}\|\mathbf{w}_t - \mathbf{v}\|^2$. By taking $\eta \leq \frac{c_0\beta}{10c\alpha^2}$ and combining with Eq. (28) we have that:

$$\|\mathbf{w}_{t+1} - \mathbf{v}\|^2 < \|\mathbf{w}_t - \mathbf{v}\|^2.$$

In particular, $\|\tilde{\mathbf{w}}_{t+1} - \tilde{\mathbf{v}}\|^2 \leq \|\mathbf{w}_{t+1} - \mathbf{v}\|^2 < \|\mathbf{w}_t - \mathbf{v}\|^2 < 1$, which shows that $\theta(\tilde{\mathbf{w}}_{t+1}, \tilde{\mathbf{v}}) \leq \frac{\pi}{2}$, and concludes the first part of the induction.

The bound for $b_t$ is more intricate, for an illustration see Figure 2. Let $t'$ be the first iteration for which $\|\tilde{\mathbf{w}}_{t'}\| \geq 0.4$. First assume that $t \leq t'$, we will show that in this case $b_t = 0$. Assume otherwise, and let $t_0$ be the first iteration for which $b_{t_0} > 0$, this means that $b_{\mathbf{w}_{t_0}} < 0$ and $b_{\mathbf{w}_{t_0-1}} \geq 0$. We have that:

$$b_{\mathbf{w}_{t_0}} = b_{\mathbf{w}_{t_0-1}} - \eta\nabla F(\mathbf{w}_{t_0})_{d+1}.$$

If $b_{\mathbf{w}_{t_0-1}} \leq \frac{\alpha^3\beta}{640}$, then by Proposition F.1 the last coordinate of the gradient is negative, hence $b_{\mathbf{w}_{t_0}} > b_{\mathbf{w}_{t_0-1}} \geq 0$. Otherwise, assume that $b_{\mathbf{w}_{t_0-1}} > \frac{\alpha^3\beta}{640}$. By Eq. (27): $|\nabla F(\mathbf{w}_{t_0})_{d+1}| \leq \|\nabla F(\mathbf{w})\| \leq \sqrt{c}$. Hence, by taking $\eta < \frac{\beta}{640\sqrt{c}} \leq \frac{\alpha^3\beta}{640\sqrt{c}}$, we get that $b_{\mathbf{w}_{t_0}} \geq 0$, which is a contradiction (note that by Assumption 6.1(3), we have $\alpha \geq 1$). We proved that if $t \leq t'$ then $b_{\mathbf{w}_t} \geq 0$, which means that $b_t = 0$.

Assume now that $t > t'$. We will need the following calculation: Assume that $\|\tilde{\mathbf{w}}_t\| = \delta$, Then $\|\tilde{\mathbf{w}}_t - \tilde{\mathbf{v}}\|^2 \geq (1 - \delta)^2$, and the minimum is achieved at $\tilde{\mathbf{w}} = \delta\tilde{\mathbf{v}}$. Since we have:

$$\|\tilde{\mathbf{w}}_t - \tilde{\mathbf{v}}\|^2 + (b_{\mathbf{w}_t} - b_{\mathbf{v}})^2 = \|\mathbf{w}_t - \mathbf{v}\|^2 \leq 1,$$

we get that $(b_{\mathbf{w}_t} - b_{\mathbf{v}})^2 \leq 1 - (1 - \delta)^2 \leq 2\delta$. If we further assume that $b_{\mathbf{w}_t} \leq 0$, then $b_{\mathbf{w}_t}^2 \leq (b_{\mathbf{w}_t} - b_{\mathbf{v}})^2 \leq 2\delta$. Combining all the above, we get that if $\|\tilde{\mathbf{w}}_t\| = \delta$ then:

$$b_t = \max\left\{0, -\frac{b_{\mathbf{w}_t}}{\|\tilde{\mathbf{w}}_t\|}\right\} \leq \sqrt{\frac{2}{\delta}}. \tag{29}$$

To show the bound on $b_t$ we split into cases, depending on the norm of $\tilde{\mathbf{w}}_t$:

**Case I:** $\frac{2\tau}{5} < \|\tilde{\mathbf{w}}_t\| \leq \frac{\tau}{2}$ and $b_{\mathbf{w}_t} \leq 0$. In this case we have:

$$\|\tilde{\mathbf{w}}_{t+1}\|^2 = \|\tilde{\mathbf{w}}_t - \eta\nabla F(\mathbf{w}_t)_{1:d}\|^2$$
$$= \|\tilde{\mathbf{w}}_t\|^2 - 2\eta\langle\tilde{\mathbf{w}}_t, \nabla F(\mathbf{w}_t)_{1:d}\rangle + \eta^2\|\nabla F(\mathbf{w}_t)_{1:d}\|^2$$
$$\geq \|\tilde{\mathbf{w}}_t\|^2 - 2\eta\langle\tilde{\mathbf{w}}_t, \nabla F(\mathbf{w}_t)_{1:d}\rangle.$$

We can use Proposition F.2 to get that $\langle \tilde{\mathbf{w}}_t, \nabla F(\mathbf{w}_t)_{1:d} \rangle \leq 0$, hence $\|\tilde{\mathbf{w}}_{t+1}\|^2 \geq \|\tilde{\mathbf{w}}_t\|^2$. By Eq. (29) we get that $b_{t+1} \leq \sqrt{\frac{5}{\tau}} \leq \frac{2.4}{\sqrt{\tau}}$.

**Case II:** $\|\tilde{\mathbf{w}}_t\| \geq \min\{0.4, \frac{\tau}{2}\}$. In this case, by choosing a step size $\eta < \frac{1}{40c} \min\{1, \tau\}$ we can bound

$$
\begin{aligned}
\|\tilde{\mathbf{w}}_{t+1}\| &\geq \|\tilde{\mathbf{w}}_t\|^2 - 2\eta\langle\tilde{\mathbf{w}}_t, \nabla F(\mathbf{w}_t)_{1:d}\rangle \\
&\geq \|\tilde{\mathbf{w}}_t\|^2 - 2\eta\|\tilde{\mathbf{w}}_t\|\|\nabla F(\mathbf{w}_t)_{1:d}\| \\
&\geq \|\tilde{\mathbf{w}}_t\|^2 - 2\eta\|\tilde{\mathbf{w}}_t\|\|\nabla F(\mathbf{w}_t)\| \\
&\geq \|\tilde{\mathbf{w}}_t\|^2 - 2\eta \cdot 2c \geq \min\left\{0.39, \frac{2\tau}{5}\right\} \ .
\end{aligned}
$$

Again, by Eq. (29) we get that $b_{t+1} \leq \max\left\{\sqrt{5.2}, \frac{2.4}{\sqrt{\tau}}\right\} \leq 2.4 \cdot \max\left\{1, \frac{1}{\sqrt{\tau}}\right\}$. This concludes the induction.

**Case III:** $\|\tilde{\mathbf{w}}_t\| \leq \min\left\{0.4, \frac{2\tau}{5}\right\}$. We split into sub-cases depending on the previous iteration: (a) If $b_{\mathbf{w}_{t-1}} \leq 0$, then by Case I the norm of $\tilde{\mathbf{w}}$ cannot get below $\frac{2\tau}{5}$, hence this sub-case is not possible; (b) If $b_{\mathbf{w}_{t-1}} \geq 0$ and $\|\tilde{\mathbf{w}}_{t-1}\| \leq \min\left\{0.4, \frac{2\tau}{5}\right\}$, then by the same reasoning in the case of $t < t'$, $b_{\mathbf{w}_t}$ cannot get smaller than zero. Hence, we must have that $b_{\mathbf{w}_{t+1}} \geq 0$; (c) If $b_{\mathbf{w}_{t-1}} \geq 0$ and $\|\tilde{\mathbf{w}}_{t-1}\| \geq \min\left\{0.4, \frac{2\tau}{5}\right\}$ then the bound depend on whether $\|\tilde{\mathbf{w}}_{t-1}\|$ is larger than $0.4$ or not. If $\|\tilde{\mathbf{w}_{t-1}}\| \leq 0.4$, then using the same reasoning as the case of $t' < t$ twice (both for the $t-1$ and $t$ iterations) we get that $b_{t+1} \geq 0$. If $\|\tilde{\mathbf{w}}_{t-1}\| > 0.4$ and $b_{\mathbf{w}_t} \geq 0$, then again this is the same case as in the case of $t' < t$ (since $\|\tilde{\mathbf{w}}_t\| \leq 0.4$. The last case is when $\|\tilde{\mathbf{w}}_{t-1}\| > 0.4$ and $b_{\mathbf{w}_t} < 0$, here using the same calculation as in Case II, we have that $\|\tilde{\mathbf{w}}_t\| \geq 0.39$. Since $\|\tilde{\mathbf{w}}_t\| \leq \min\left\{0.4, \frac{2\tau}{5}\right\}$, using Proposition F.2, the norm of $\tilde{\mathbf{w}}_t$ can only grow, hence by the same reasoning as in Case I we can also bound $b_{t+1} < 2.4\max\left\{1, \frac{1}{\sqrt{\tau}}\right\}$.

Until now we have proven that throughout the entire optimization process we have that $\theta(\tilde{\mathbf{w}}_t, \tilde{\mathbf{v}}) \leq \frac{\pi}{2}$ and $b_t \leq 2.4 \cdot \max\left\{1, \frac{1}{\sqrt{\tau}}\right\}$. Let $\delta = \pi - \theta(\tilde{\mathbf{w}}_t, \tilde{\mathbf{v}})$, we now use Theorem A.2 and Eq. (27) to get that:

$$
\begin{aligned}
\|\mathbf{w}_{t+1} - \mathbf{v}\|^2 &= \|\mathbf{w}_t - \eta\nabla F(\mathbf{w}_t) - \mathbf{v}\|^2 \\
&= \|\mathbf{w}_t - \mathbf{v}\|^2 - 2\eta\langle\nabla F(\mathbf{w}_t), \mathbf{w}_t - \mathbf{v}\rangle + \eta^2\|\nabla F(\mathbf{w}_t)\|^2 \\
&\leq \|\mathbf{w}_t - \mathbf{v}\|^2 - 2\eta\frac{\left(\alpha - \frac{b_t}{\sin\left(\frac{\delta}{2}\right)}\right)^4\beta}{8^4\alpha^2}\sin\left(\frac{\delta}{4}\right)^3\|\mathbf{w}_t - \mathbf{v}\|^2 + \eta^2c\|\mathbf{w}_t - \mathbf{v}\|^2 \\
&\leq \|\mathbf{w}_t - \mathbf{v}\|^2 - \eta\frac{\left(\alpha - \sqrt{2}b_t\right)^4\beta}{8^4\alpha^2}\sin\left(\frac{\delta}{4}\right)^3\|\mathbf{w}_t - \mathbf{v}\|^2 + \eta^2c\|\mathbf{w}_t - \mathbf{v}\|^2 \\
&\leq \|\mathbf{w}_t - \mathbf{v}\|^2 - \frac{\eta\tilde{C}\beta}{\alpha^2}\|\mathbf{w}_t - \mathbf{v}\|^2 + \eta^2c\|\mathbf{w}_t - \mathbf{v}\|^2 \quad\quad (30)
\end{aligned}
$$

where $\tilde{C}$ is some universal constant, and we used the bounds from the induction above that $\delta \in \left[\frac{\pi}{2}, \pi\right]$, $b_t \leq 2.4 \cdot \max\left\{1, \frac{1}{\sqrt{\tau}}\right\}$, and by the assumption that $\alpha \geq 2.5\sqrt{2}\max\left\{1, \frac{1}{\sqrt{\tau}}\right\}$. By choosing $\eta \leq \frac{\tilde{C}\beta}{2c\alpha^2}$, and setting $\lambda = \frac{\tilde{C}\beta}{2c\alpha^2}$ we get that:

$$
\begin{aligned}
&\|\mathbf{w}_t - \mathbf{v}\|^2 - \eta\tilde{C}\beta\min\left\{1, \frac{1}{\alpha^2}\right\}\|\mathbf{w}_t - \mathbf{v}\|^2 + \eta^2c\|\mathbf{w}_t - \mathbf{v}\|^2 \\
&\leq (1 - \lambda\eta)\|\mathbf{w}_t - \mathbf{v}\|^2 \leq \cdots \leq (1 - \eta\lambda)^t\|\mathbf{w}_0 - \mathbf{v}\|^2 ,
\end{aligned}
$$

which finished the proof.

$\square$

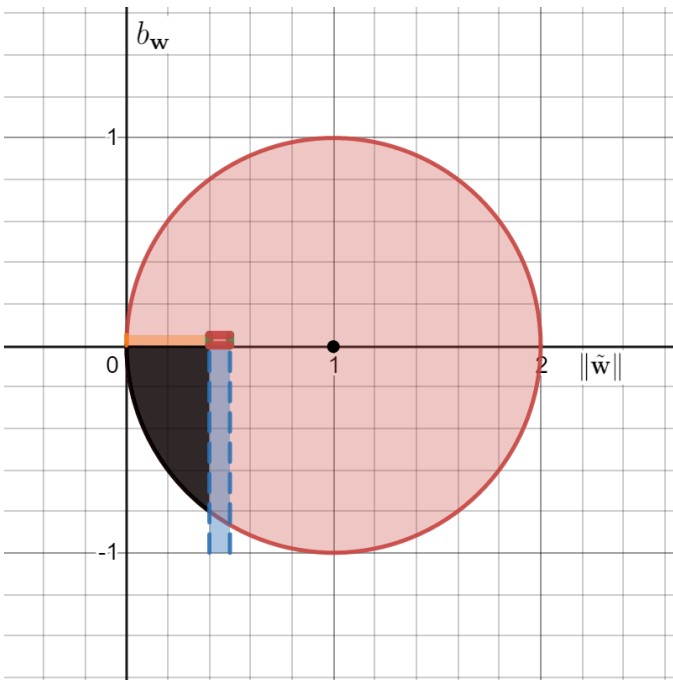

Figure 2: A 2-d illustration of the optimization landscape. The $x$ axis represents $\|\tilde{\mathbf{w}}\|$, and the $y$-axis represents $b_{\mathbf{w}}$. In the figure, for simplicity, we assume that $b_{\mathbf{v}} = 0$, and $\tau = 0.1$ which means that $\frac{2\tau}{5} = 0.4$. The red circle represents the area with $\|\mathbf{w} - \mathbf{v}\| \leq 1$, throughout the optimization process $\mathbf{w}_t$ stays in this circle. The black region represents the area where $b_t = -\frac{b_{\mathbf{w}}}{\|\tilde{\mathbf{w}}\|}$ can be potentially large, our goal is to show that $\mathbf{w}_t$ stays out of this region. Case I shows that $\mathbf{w}_t$ cannot cross the blue region. Case II shows that if $\mathbf{w}_t$ is to the right of the black region, then $b_t$ is upper bounded. Case III shows that $\mathbf{w}_t$ cannot cross the orange region (sub-cases (a) and (b)), and cannot cross from the green region directly to the black region (sub-case (c)).