# OpenReview forum: "Learning a Single Neuron with Bias Using Gradient Descent"
_NeurIPS.cc/2021/Conference — NeurIPS 2021 Poster_

### Official Review · Reviewer_mnNc · 2021-07-14

**Rating:** 5
**Confidence:** 5

**Summary:**

The work considers learning a single ReLU with the bias term using gradient based methods (and more specifically with gradient flow and gradient descent - which falls in the infinite data regime). Whereas previous works mostly considered this problem without the bias term, it (the bias term) is essential in a lot of practical applications. The paper considers gradient descent/flow with respect to standard squared error, which is practically relevant.

First, it is shown (Section 3) that learning a single ReLU with bias via. gradient based methods is much harder than without by constructing specific negative examples. Then, under natural assumptions on the data distribution, the stationary points of the loss function is characterized and it is shown that it is a cone in Section 4 (rather than a finite set as is the case when there is no bias). The  subsequent sections explore several assumptions under which convergence can be guaranteed with high probability and with random initialization.



**Limitations And Societal Impact:**

I am satisfied with the limitations mentioned in Remark 3.3 and discussion following Theorem 6.2. I would add that the lower bounds are with respect to artificial distributions which are not interesting in practice and that the problems with convergence disappear when we consider activations like LeakyReLU instead of ReLU. It would also be useful to expand on Remark 3.3 and/or consider prediction error bounds.

Discussion on societal impact not applicable since this is a purely theoretical work.

**Main Review:**

I believe that the paper is well written and easily accessible.

The proof techniques for the lower bounds involves construction rather artificial scenarios with negative bias where no non-trivial information can be extracted. This is easy to see since ReLU(t) = 0 whenever t < 0 and hence it is easy to construct scenarios when <w_0,x> < 0 almost surely x and w.p >= const. wrt initialization w_0 which ensures that gradient based methods don't move at all from the initialization (Theorem 3.1). Theorem 3.2 constructs a slightly less artificial examples which ensures that no movement is possible because of mismatch between labeling between w and v (i.e, they do not label any point positively at the same time).

A concern which is addressed in Remark 3.3 here is that one can obtain a good enough prediction error even when w = 0 and we do not have to recover the exact vector v.  All these concerns seem to disappear when a uniformly expansive function like Leaky ReLU is used instead of ReLU, where it is not possible to hide the `'information' about v outside the support of the activation function. I feel that the authors have to note this fact in the manuscript.

I believe that the non-trivial version of the problem occurs when <v,x> + b_v > 0 with some constant probability, which is partly addressed by Theorem 3.2 .

The characterization of the critical points is crisp and useful, which is then used to provide practical conditions for convergence of gradient descent under natural assumptions on the input x. One such assumption is that of positive bias which makes the problem fairly easy. I believe that the most notable contribution of the paper is Theorem 5.2 which along with Theorem 5.4 shows that when the initialization is close to 0, then we can expect global convergence with high probability.

Other comments:
1. Theorems 5.2 and 5.4 use various constants like b,c, c' etc. It would be useful to explicitly write out these constants for simple distributions like the uniform distribution on the sphere.
2. Typo: 1-exp(\Omega(d)) instead of 1-exp(-\Omega(d)).
3. The authors mention that GLMtron is not gradient based - which I believe is incorrect. It is indeed gradient based but with the convex proxy loss instead of the square loss.
4. The notation in Assumption 5.3 is a bit confusing since w and v are also used for the iterate and actual parameter respectively. I understand that this choice is natural since this is exactly the subspace where this assumption is used in the proofs, but I urge the authors to reconsider.
5. In Theorem 3.1, epsilon needs to be chosen to be very small (1/sqrt{d}) which seems artificial.
6. In Theorem 3.2, the probability of failure is 1/2 - o_d(1). How does o_d(1)  depend on the radius parameters r and rho? In theorem 3.1, the radius had to be chosen as 1/sqrt{d} to obtain non-trivial lower bounds.

**Time Spent Reviewing:**

4 hours

---

> ### Author Response · Authors · 2021-08-10
> **Response**
>
> We thank the reviewer for the comments, which we address below.
>
> 1) The assumptions in the lower bounds: We emphasize that our positive results give strong guarantees for convergence, w.h.p over the initialization and under mild assumptions. In light of that, it is not surprising that our negative results are a bit limited. If we could show very strong negative results for standard settings then we could not obtain such strong positive results. The main motivation for studying the negative results are: (1) To show that the setting of learning a neuron with bias is significantly different from the bias-less case. (2) To understand what assumptions are required in order to obtain positive results.
>
>
> 2) LeakyReLU vs. ReLU: In Yehudai and Shamir 2020 it is shown that for strictly monotonic activations (e.g. LeakyReLU) the problem becomes significantly easier (see Thm. 3.2 in Yehudai and Shamir). We believe that as in the bias-less case, it is possible to show positive results for LeakyReLU that apply even in the setting where we proved negative results for ReLU. Hence, it is no surprise that the negative results may not apply to this activation. In this work, we decided to focus on the ReLU activation which gives a more interesting case-study where there is a manifold of non-global minima to which gradient descent may converge to.
>
>
> 3) “One such assumption is that of positive bias which makes the problem fairly easy”: We emphasize that in all the results from Section 5 we did not assume positive bias. As explained in Remark 5.6 our results hold for negative biases as well.
>
>
> 4) o_d(1) in Theorem 3.2: The o_d(1) term does not depend on r and \rho. Note that such a dependence is included in b_v. As stated in Lemma B.2 in the appendix, the probability of initializing w such that it does not label positively any point that v labels positively depends only on d.
>
>
> 5) We will fix and clarify following the other comments in the camera ready version.

---

> > ### Comment · Reviewer_mnNc · 2021-09-10
> > **Response**
> >
> > Thank you for the clarification. I choose to retain my score.

---

### Official Review · Reviewer_K5Jf · 2021-07-16

**Rating:** 7
**Confidence:** 3

**Summary:**

The paper considers the task of learning ReLU with a bias term and shows that gradient descent converges to global minimum under reasonable data and initialization assumptions. Moreover, they also prove a two "hardness" theorems showing that having the bias makes the landscape more complicated, and this justifies the problem deserves further research.

**Limitations And Societal Impact:**

The paper addresses some limitations of their work throughout their paper.

I believe the paper will not lead to any potential negative societal impact, due to its theoretical nature.

**Main Review:**

Update
----
I have read the author(s)' response to my review and maintain my rating.

Originality
---
Learning a single ReLU is a classic task. The paper considers the task of learning ReLU with a bias term, using GD method, which has not been studied before in the literature.

Some proof techniques in the paper are not new, as they largely follow from similar paradigm from Yehudai and Shamir [2020].

The paper discusses related work in sufficient depth. A missing reference might be an early work on learning single ReLU (with non GD method): [Surbhi Goel, Varun Kanade, Adam Klivans, and Justin Thaler. "Reliably learning the ReLU in polynomial time." In Conference on Learning Theory, 2017].

Quality
----
I've read the main paper and some proofs in appendix. I do not see any major issue.

For Theorem 3.1: based on the statement and its proof, the parameter $\varepsilon$ needs to be sufficiently small, say, $\varepsilon \ll 1/\sqrt{d}$. The authors may want to clarify any condition on $\varepsilon$ here.

For Theorem 6.1, item 3 of Assumption 6.1 looks a bit confusing. Is it the case that only $x_1, x_2$ need to satisfy the bound?

Clarity
----
Overall, I find the paper well-structured and well-written. The theorem statements are motivated and supported by intuitive explanations.

Some minor points:
1. Line 194: "note that in Yehudai and Shamir [2020] it is shown that without any assumption on the distribution, it is impossible to ensure convergence".  Here the work [Surbhi Goel, Adam Klivans, Pasin Manurangsi, Daniel Reichman. "Tight Hardness Results for Training Depth-2 ReLU Networks", In ITCS 2021] also gives worst-case hardness result (under no data assumption). It makes sense to refer to this paper as well.

2. Line 269: maybe make a further comment that this is done by bounding the norm of all the training data whp, which is guaranteed by sub-gaussian distributions.

Significance
----
The paper builds upon and extends an established line of work, the most closely related one being Yehudai and Shamir [2020]. I believe this work takes a good step along this line.

**Time Spent Reviewing:**

4

---

> ### Author Response · Authors · 2021-08-10
> **Response**
>
> We thank the reviewer for the constructive comments.
>
> Regarding Thm 3.1, we will clarify the condition on \epsilon.
>
> Regarding Assumption 6.1, note that item 3 considers x_1,x_2, but due to the assumption on the symmetry of the distribution (item 2), the assumption in item 3 holds for every x_i, x_j. We will clarify this issue.
>
> We thank you for the relevant references and will add them to the paper.

---

### Official Review · Reviewer_9NfA · 2021-07-16

**Rating:** 7
**Confidence:** 3

**Summary:**

In this paper, the authors consider the task of learning a single ReLu neuron with bias using gradient descent, which has a significantly different behavior than the unbiased case. For example, adding a bias leads to examples where gradient descent fails to find the global minimum with probability $1/2$, whereas GD succeeds wp 1 with no bias. This is explain by GD getting stuck at critical points which are characterized in Section 4. The authors then show that if the GD is initialized at a point with loss better than the trivial loss, then it converges to the global minimum. This good start happens with high probability for standard symmetric initialization (and 0 bias initialization), and can be shown to converge at linear rate. Finally, in section 6, they provide an alternative set of assumptions that leads to linear convergence, with rate that do not depend on initialization (but with lower probability on initialization).


--- after rebuttal ---

I am satisfied with the authors response. I am keeping the same rating.

**Limitations And Societal Impact:**

Yes

**Main Review:**

The paper is well written and very pleasant to read. Each idea is introduced carefully and well explained. Plenty of intuition is provided on the proofs and the assumptions. Despite its simplicity, the problem addressed is fundamental and can provide some building block intuition on learning a single neuron. For these reasons, I recommend the paper to be accepted.

Some comments:
- The proofs and intuitions depend highly on the choice of the ReLu activation. While this is indeed the most standard non-linearity, I expect a lot of the results to break down when choosing an other activation. I wonder what happens in this case.

- Some of the previous work consider gradient descent on the empirical loss (or SGD), instead of the population loss. This would also lead to a quite different analysis. I wonder if some concentration of landscape type analysis, similar to Mei et al. 2016 (this reference should be updated), could work, and if not what would break down in the analysis?

- The negative results seem also to break down if several neurons are used.

To the best of my knowledge, this is indeed the first work that considers the case with non-zero bias and I think it would be of interest to the NeurIPS community.

Typo: the standard notation is $g = \Omega (f)$ if $g \geq c \cdot f$ for some $c>0$. I think you should add a minus in $e^{\Omega (d)}$.

**Time Spent Reviewing:**

3

---

> ### Author Response · Authors · 2021-08-10
> **Response**
>
> We thank the reviewer for the constructive review and will make sure to fix the typos.
>
> Regarding the reviewers’ questions, these are all very interesting questions for future research, although unfortunately are difficult to answer at the moment. In particular, we agree that the results focus on a single neuron with a ReLU activation, with respect to the population loss. We do believe that they can be extended to other activations under certain assumptions  (e.g., Assumption 4.1 from Yehudai and Shamir 2020), and that they might be extendable to an empirical loss using a concentration approach as suggested by the reviewer. However, these are non-trivial goals and are left to future work.

---

### Official Review · Reviewer_ahWe · 2021-07-16

**Rating:** 6
**Confidence:** 4

**Summary:**

The paper studies the problem of learning a single neuron with a bias term in the realizable setting with the ReLU activation using gradient descent. The authors showed the behavior is significantly different from the case without bias both in terms of the optimization landscape and the ability of gradient methods to succeed.

**Limitations And Societal Impact:**

The authors considered learning a single neuron with a bias term using gradient methods. Given the results in Yehudai and Shamir (2020), the theoretical contribution is limited. The authors may consider more complex network architecture.

**Main Review:**

The paper is well written in general and has made concrete contribution to the fundamental problem in deep learning. The authors studied learning a single neuron with a bias term with the ReLU activation via gradient methods, which is of central interest in the theory of deep learning. They provided some failure cases to demonstrate the differences from the non-bias case. They also characterized all the critical points of the objective function and provided positive convergence results under several different assumptions.

My major concerns are as follows: Given the results from the previous work Yehudai and Shamir (2020), the results in this paper are not that surprising and less inspiring. The theoretical contribution is limited in this sense and the analysis can be seen as extension of Yehudai and Shamir (2020). The analysis has less insight compared to the previous work. The authors may analyze more complex network structure, which may give more fundamentally different optimization geometry.

**Time Spent Reviewing:**

7

---

> ### Author Response · Authors · 2021-08-10
> **Response**
>
> We thank the reviewer for the comments.
>
> 1) “The results are not that surprising”: We show that the setting of learning a neuron with bias is significantly different from the bias-less case. In particular, we show that the optimization landscape is much more complex, and give negative results that do not occur in the bias-less setting. In light of these results, we actually found the positive results quite surprising. Namely, even with very mild assumptions (the assumptions in this work are not stronger than the assumptions from Yehudai and Shamir 2020) we were able to obtain convergence guarantees.
>
> 2) “The theoretical contribution is limited”: We emphasize that in this work we developed new theoretical tools which enabled us to obtain significantly stronger results than Yehudai and Shamir 2020. In Yehudai and Shamir the authors were able to show convergence with probability close to 0.5 w.r.t. the initialization (for non spherically-symmetric distributions). In this work (Section 5) we established convergence guarantees with probability close to 1. Thus, our new technique allowed us to strengthen the results of Yehudai and Shamir. We believe that the technique might be also useful for obtaining other new results (e.g., on learning a single neuron in the agnostic setting [Frei et al. 2020], etc.).

---

### Decision · Program_Chairs · 2021-09-27

**Decision:**

Accept (Poster)

**Comment:**

The reviewers generally find the paper to be a valuable addition to the neurips proceeding, despite some concerns on the originality (i.e., the paper seems to be a direct extension of prior works in terms of the problem setup and proof techniques.) The AC recommends the authors to discuss the relationship with prior works (on a both conceptual level and low-level technical level) more prominently.